# How can the First ISLSCP Field Experiment contribute to present-day efforts to evaluate water stress in JULESv5.0?

Karina E Williams[1], Anna B Harper[2], Chris Huntingford[3], Lina M Mercado[3,4], Camilla T Mathison[1], Pete D Falloon[1], Peter M Cox[2], and Joon Kim[5,6]

[1]Met Office, FitzRoy Road, Exeter, Devon, EX1 3PB, UK
[2]College of Engineering, Mathematics and Physical Sciences, University of Exeter, Exeter, EX4 4QF, UK
[3]Center for Ecology and Hydrology, Wallingford, OX10 8BB, UK
[4]Geography Department, College of Life and Environmental Sciences, University of Exeter, Exeter, UK
[5]Dept. of Landscape Architecture & Rural Systems Engineering, Interdisciplinary Program in Agricultural & Forest Meteorology, Research Institute for Agriculture and Life Sciences, Seoul National University, Seoul 08826, Republic of Korea
[6]Institute of GreenBio Science & Technology, Seoul National University, Pyeongchang, 25354 Republic of Korea

**Abstract.** The First ISLSCP Field Experiment (FIFE), Kansas, US, 1987-1989, made important contributions to the understanding of energy and $CO_2$ exchanges between the land-surface and the atmosphere, which heavily influenced the development of numerical land-surface modelling. Thirty years on, we demonstrate how the wealth of data collected at FIFE and its subsequent in-depth analysis in the literature continues to be a valuable resource for the current generation of land-surface models. To illustrate, we use the FIFE dataset to evaluate the representation of water stress on tallgrass prairie vegetation in the Joint UK Land Environment Simulator (JULES) and highlight areas for future development. We show that, while JULES is able to simulate a decrease in net carbon assimilation and evapotranspiration during a dry spell, the shape of the diurnal cycle is not well captured. Evaluating the model parameters and results against this dataset provides a case study on the assumptions in calibrating 'unstressed' vegetation parameters and thresholds for water stress. In particular, the response to low water availability and high temperatures are calibrated separately. We also illustrate the effect of inherent uncertainties in key observables, such as leaf area index, soil moisture and soil properties. Given these valuable lessons, simulations for this site will be a key addition to a compilation of simulations covering a wide range of vegetation types and climate regimes, which will be used to improve the way that water stress is represented within JULES.

# 1 Introduction

Models of the land surface and biosphere, a key component in climate predictions and projections, depend on high quality observational datasets to tune the behaviour of the modelled processes. A significant contribution in this field was produced by the First ISLCP[1] Field Experiment (FIFE), an interdisciplinary collaboration of researchers from remote sensing, atmospheric physics, meteorology and biology. It was based at and around the Konza Prairie Long Term Ecological Research (LTER) site, Kansas, during multiple campaigns, 1987-1989. Its principal objectives were twofold: to improve the understanding of the role of biological processes in controlling atmosphere–surface exchange of heat, water vapour and $CO_2$, and to investigate whether satellite observations can constrain land surface parameters relevant to the climate system (Sellers et al., 1988; Sellers and Hall, 1992).

As part of this experiment, canopy processes were related to leaf-level stomatal conductance, photosynthesis and respiration, including detailed modelling of responses to water availability and atmospheric forcing (Verma et al., 1989; Kim and Verma, 1990b, a, 1991a, b; Verma et al., 1992; Kim et al., 1992; Stewart and Verma, 1992; Norman et al., 1992; Niyogi and Raman, 1997; Cox et al., 1998; Colello et al., 1998). This work has subsequently played an important role in influencing the representation of vegetation in a generation of land-surface models. The parametrisation of water stress in the the Joint UK Land Environment Simulator (JULES) (Best et al., 2011; Clark et al., 2011), for example, originates in a canopy conductance and photosynthesis model presented in Cox et al. (1998), which was developed using FIFE observations. After tuning, the Cox et al. (1998) model gave a very good fit to the data: it explained 91.7% of the variance in net canopy photosynthesis and 89.4% of the variance in canopy conductance, as derived from FIFE flux tower observations. As part of this model, Cox et al. (1998) calculated a piecewise-linear stress factor $\beta$. This factor is zero below the wilting soil moisture and one above a critical soil moisture (Figure 1, solid line), based on the top 1.4m of soil. Crucially, Cox et al. (1998) found that the drop in carbon assimilation in the C4 vegetation as soil water content decreased at FIFE could only be reproduced if the stress factor $\beta$ was applied directly to the net leaf assimilation rate. In their model, soil water stress affected stomatal conductance via the net leaf assimilation rate.

The Cox et al. (1998) stress parameterisation was adopted in early versions of JULES. It was the only implementation of soil moisture stress in JULES until version 4.6 and, to our knowledge, has been used in all published studies to date. The JULES wilting and critical soil moistures are input by users for each soil layer in each gridbox, and are defined as corresponding to absolute matric water potentials of 1.5 MPa and 0.033 MPa respectively (Best et al., 2011). A separate stress factor is calculated for each soil layer, and these are combined into an overall soil moisture stress factor by weighting by the root mass distribution. Other options have been more recently implemented into JULES. These include a 'bucket' approach, in which the stress factor $\beta$ is calculated from the average soil moisture to a specified depth, and the introduction of a new variable $p_0$ which reduces the soil moisture at which a vegetation type first starts to experience water stress (Figure 1, dashed line).

There is currently a community-wide effort to improve the response of JULES to drought conditions. This effort requires a large amount of data to evaluate against, covering a wide variety of climates and vegetation types, in order to give confidence

---

[1]International Satellite Land Surface Climatology Project

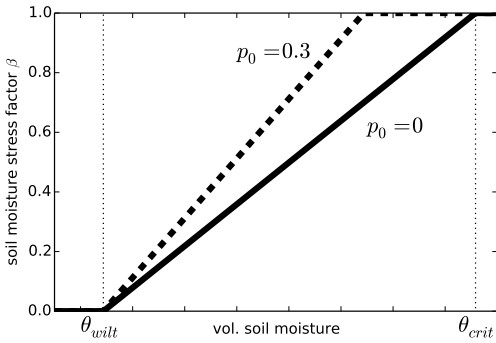

**Figure 1.** JULES soil moisture stress factor $\beta$ with $p_0$=0 (solid line) and $p_0$=0.3 (dashed line). The soil moisture threshold at which the plant becomes completely unstressed ($\beta = 1$) is $\theta_{wilt} + (\theta_{crit} - \theta_{wilt})(1 - p_0)$.

in the underlying representation of this process in the model. This is vital if the model is to be used to simulate global responses to changes in water availability in the future.

Observations taken during the FIFE campaign are still available today, through the Oak Ridge National Laboratory Distributed Active Archive Center (ORNL-DAAC). Given that FIFE observations were fundamental to the development of the
original water stress parametrisation in JULES, we revisit this dataset to determine whether it would make a useful contribution to present-day efforts to improve this process. We aim to demonstrate that there is sufficient data available, and of a sufficient quality, to show that the current version of JULES is unable to capture key features of the impact of water availability on the temperate grassland vegetation at the FIFE site. This can provide a benchmark for this vegetation type, against which future model developments can be assessed. We thus hope to encourage the inclusion of this dataset in comprehensive, multi-site
studies that aim to improve the representation of this process on a global scale.

We first create a simulation that closely reproduces the Cox et al. (1998) study, in order to investigate how this original study was able to provide such a close fit to the observed carbon and water fluxes at FIFE. Our second configuration uses more recent model developments, with parameter values based on the generic C4 grass tile from the global analysis of Harper et al. (2016). These settings are typical for how this vegetation type is usually represented in current-day runs of JULES. We then use FIFE
observations to tune some of these generic C4 grass parameters to more accurately represent tallgrass prairie. The aim here is to allow us to distinguish between model limitations due to approximating this specific vegetation type by generic C4 grass parameters and model limitations due to missing or inadequately represented processes within the model. The model setup for each of these simulations is described in Section 2. In Section 3, we compare the results from the model simulations to net canopy carbon assimilation, derived from $CO_2$ flux measurements, and latent heat energy flux measurements at the FIFE
site. We conclude with a summary of what lessons can be learnt for improving water stress in JULES from FIFE and how this dataset can be useful to the JULES community into the future. Throughout, we refer to the appendices, which give more information about the use of the observations and the alternative datasets considered, in order to assist future modelling work at

this site, both with JULES and other land-surface models. A important component of this study is the provision of a complete JULES setup that can be downloaded and used to run FIFE data through the JULES model, to allow easy inclusion of this site into a comprehensive evaluation framework for JULES.

## 2 Experimental set-up

We will focus on three different configurations of JULES:

- Simulation 1: `repro-cox-1998`. A simplified JULES run which reproduces the original Cox et al. (1998) study as closely as possible. This requires the simple 'big leaf' canopy scheme, prescribes the Leaf Area Index (LAI) and soil moisture from observations, and calculates the soil moisture stress from the average soil moisture in the top 1.4m of soil.

- Simulation 2: `global-C4-grass`. This run uses parameter settings from Harper et al. (2016), which has a generic
representation of C4 grass. It uses many of the 'state-of-art' features of JULES, such as the layered canopy scheme with sunflecks, and calculates soil moisture stress using a weighted sum of the stress factors in each soil layer. LAI and soil moisture are prescribed.

- Simulation 3: `tune-leaf`. As above, but we investigate whether the generic C4 grass leaf parameters can be tuned to site measurements, to give a more accurate representation of the prairie vegetation.

These configurations are described below and summarised in Table 1. All the FIFE datasets used in this study are given in Table A1.

### 2.1 Simulation 1: `repro-cox-1998`

Our first simulation, `repro-cox-1998`, closely reproduces the optimal configuration presented in the Cox et al. (1998) study. Cox et al. (1998) modelled the fluxes for FIFE site 4439 (situated at 39° 03' N, 96° 32' W, 445 m above mean sea level).
This tallgrass prairie site is roughly central within the 15km × 15km FIFE study area. It had been lightly grazed by domestic livestock, but was ungrazed in 1986 and 1987 and was burned on 16th April 1987 (Kim and Verma, 1990a, 1991b). At the flowering stage in 1987, more than 80% of the vegetation was composed of C4 grasses (Kim and Verma, 1990a).

For their analysis, Cox et al. (1998) selected daylight hours that were both after 10 am local time, to exclude dew evaporation, and from days with no rainfall during that day or the preceding day. This minimised the effect of evaporation of rainfall from
the canopy and soil surface and let them focus on modelling transpiration and net canopy assimilation. We will also restrict our analysis to these same time periods. The model was spun up by repeating the entire run ten times, and the output from the eleventh run was analysed.

For driving data, we use a site-averaged product of the FIFE Portable Automatic Meteorological Station (AMS) data at 30 minute resolution (Betts and Ball, 1998). We prescribe both LAI and soil moisture from observations (Stewart and Verma,
1992) rather than calculating these variables internally using the JULES phenology or soil hydrology schemes. We use a

'bucket approach' to calculate the soil moisture stress factor from the average soil moisture in the top 1.4m (this option has been available from JULES 4.6 onwards), again to mimic the Cox et al. (1998) analysis. The wilting soil moisture $\theta_{wilt}$ was set to 0.205 m$^3$ m$^{-3}$ and the critical soil moisture $\theta_{crit}$ was set to 0.387 m$^3$ m$^{-3}$, taken directly from Cox et al. (1998). The resulting stress factor is plotted in Figure 2, and clearly shows the dry period during late July and early August.

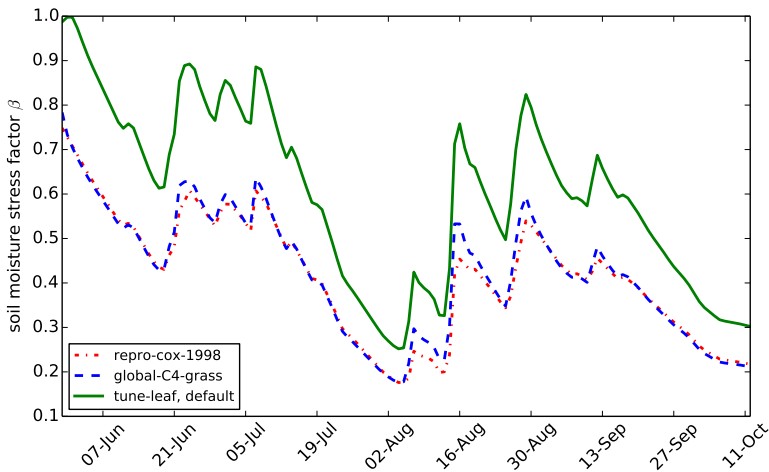

**Figure 2.** Daily mean soil moisture stress factor $\beta$ for each JULES simulation at FIFE site 4439 in 1987.

JULES and the Cox et al. (1998) optimal configuration both use the Collatz et al. (1992) C4 photosynthesis scheme. They also both use the same stomatal conductance parametrisation: Jacobs (1994), which is in turn a simplified version of the Leuning (1995) scheme. We select the 'big leaf' option from the available canopy schemes in JULES, again to mimic Cox et al. (1998).

     In this way, we are able to closely reproduce the Cox et al. (1998) calculation of daytime net canopy carbon assimilation

and daytime canopy conductance with a modern version of JULES. Any remaining differences are minor. For example, in Cox et al. (1998) leaf temperature is calculated from the air temperature and observed sensible heat flux whereas, in JULES, the full energy balance is modelled. There are also differences in the calculation of evaporation from soil and canopy, which are not the focus of this study. The calculation of aerodynamic resistance also differs. For example, in this run, canopy height is prescribed using the data from Verma et al. (1992) for this site in 1987 (see Section A5 for more information), whereas it was

not modelled explicitly as part of the Cox et al. (1998) analysis.

     Many of the key FIFE datasets used in this run have large uncertainties, despite being comprehesively measured by multiple teams. LAI measurements have an error of approximately 75% due to the inherent variability of prairie vegetation. LAI measurements are also affected by leaf curling or folding as the leaves pass through the detector. There are therefore significant differences between datasets (for a more detailed description, see Section A2). For example, at the beginning of August, LAI

measurements vary from 2.5 (Stewart and Verma, 1992) to 0.7 (the FIFE_VEG_BIOP_135 dataset). Soil moisture was also

comprehensively measured across the FIFE area by multiple groups (see Section A3). While these observations are qualitatively consistent, one of the datasets shows a bias in the lower soil levels at site 4439 in 1987 compared to the other datasets. Within-site variability in soil moisture is also large. Soil properties were similarly well studied: there are four different datasets which can be used to calculate the wilting and critical soil moistures, plus the values from two additional published studies (described

in Section A4). However, measurements differ from each other by more than 0.15 m$^3$ m$^{-3}$ in some cases. There also appears to be differences between layers, with the top 10 cm having consistently lower wilting and critical thresholds than soil at a depth of about 30 cm, for example. It is therefore vital that we consider the implications of the spread in observed LAI, soil moisture and soil properties at this site when drawing our conclusions.

## 2.2   **Simulation 2:** `global-C4-grass`

In our second simulation, we use a recent JULES configuration, presented in Harper et al. (2016). This study introduced a trait-based approach to calculating leaf physiology in JULES, and tuned plant parameters to observations in the TRY database (Kattge et al., 2011). Global vegetation was split into 9 plant functional types (PFTs), including one to represent all C4 grasses. The developments introduced in Harper et al. (2016) resulted in improved site-scale and global simulations of plant productivity and global vegetation distributions (Harper et al., 2018). Our `global-C4-grass` configuration is based on the representation

of C4 grasses in Harper et al. (2016) and takes advantage of many of the modern features of JULES. This includes a layered canopy scheme that treats the direct and diffuse components of the incident radiation separately (as in Sellers (1985)) and includes sunflecks (Dai et al., 2004; Mercado et al., 2007, 2009). It also calculates the overall soil moisture stress factor $\beta$ from the sum of the stress factors in each layer, weighted by the root mass distribution. Since we are focussing specifically on the parameterisation of water stress, we continue to prescribe LAI and soil moisture, rather than calculate these parameters

dynamically with the JULES phenology and soil hydrology schemes.

    The driving data was taken from the site-averaged Betts and Ball (1998) product. The diffuse radiation fraction was calculated from shortwave radiation using the method in Weiss and Norman (1985) (see Section A1 for more information). A spherical leaf angle distribution was used, as in Harper et al. (2016). LAI was prescribed using the Stewart and Verma (1992) observations and the vegetation was set to generic C4 grass.

The Stewart and Verma (1992) soil moisture observations were partitioned into the four JULES soil layers (thicknesses 0.1m, 0.25m, 0.75m and 2.0m) using an offline version of the soil hydrology scheme in JULES, assuming the same root distribution as natural C4 grass in Harper et al. (2016). This is described in more detail in Section A3.1. The wilting and critical volumetric soil moistures and the soil albedo were set to the same values as the `repro-cox-1998` run. As Figure 2 shows, the resulting soil moisture stress factor is almost identical to the simulation `repro-cox-1998`. Canopy height

was also prescribed using the same observations as the `repro-cox-1998` configuration, and the run was initialised from the spun up `repro-cox-1998` run.

## 2.3 Simulation 3: `tune-leaf`

For the third configuration, `tune-leaf`, we calibrate the JULES parameters to measurements of the tallgrass prairie vegetation at this particular site. At the flowering stage in 1987, the vegetation at FIFE site 4439 was dominated by three C4 grass species: 27.1% *Andropogon gerardii* (Big bluestem), 22.2% *Sorghastrum nutans* (Indiangrass) and 16.6 % *Panicum virgatum* (Switchgrass) (Kim and Verma, 1990a). Since individual LAI observations for each species (as used in e.g. Kim and Verma (1991b)) were not available, we continue to model this site with a single plant tile. We tune the leaf parameters of this tile to be approximately representative of the dominant species at this site, *A. gerardii*.

### 2.3.1 Leaf properties prior to the application of water stress in the model

As discussed above, JULES uses the Collatz et al. (1992) C4 photosynthesis scheme to calculate the unstressed net leaf photosynthetic carbon uptake and the Jacobs (1994) relation to calculate stomatal conductance. In this section, we calibrate these parameterisations to the available *in situ* observations. A brief description of each of the model parameters fitted in this section is given in Table A2, and they are defined in full in Clark et al. (2011) and Best et al. (2011). Throughout this calibration work, the model points/lines are calculated with the Leaf Simulator package (Williams et al., in prep). This package exactly reproduces the way that JULES calculates leaf carbon uptake and stomatal conductance, but allows leaf-level observations to be used as input.

Knapp (1985) compared leaf-level measurements of *A. gerardii* and *P. virgatum* in burned and unburned ungrazed plots on the Konza Prairie Research Natural Area in 1983, and the response of these two species to different water stress conditions. Their plots were located at 39° 05' N, 96° 35' W, which is within what subsequently became the FIFE study area. The burning occurred in April 1983, to prior to initiation of growth of the warm-season grasses. They found significant differences between vegetation in the burned plot and unburned plots during the May to September period. The particular FIFE site we are modelling in our simulations, site 4439, was also burned prior to the start of the experiment (15th April 1987, Kim and Verma (1990a)), and was ungrazed throughout the FIFE period. Therefore, we use the observations from the burned plot in Knapp (1985) during May-June 1983, when they describe water availability as 'not limiting' (we will investigate this claim in more detail in Section 2.3.2), to constrain our unstressed leaf photosynthesis parameters in the `tune-leaf` configuration. First, we set specific leaf area and the ratio of leaf nitrogen to leaf dry mass for *A. gerardii* and *P. virgatum* to Knapp (1985) observations taken between 25th May and 10th June 1983. Once these parameters are fixed, we then fit the other parameters in the model light response curve by comparison with the light curve presented in Knapp (1985), which was compiled from observations taken May-June 1983 at 35$\pm$2°C (Figure 3).

Knapp (1985) also investigated the temperature dependence of net leaf photosynthesis by artificially altering the temperature of leaves of *A. gerardii* and *P. virgatum*. Their observations showed that the peaks in both species occurred at approximately the same temperatures, but that the peak was significantly broader in *A. gerardii* than *P. virgatum*. In JULES, the temperature dependence of net leaf assimilation for C4 plants is introduced through a temperature-dependent parameterisation of the maximum rate of carboxylation of Rubisco $V_{cmax}$. This enters the calculation of both the gross rate of photosynthesis

and the dark leaf respiration $R_d$ (since model $R_d$ is proportional to model $V_{cmax}$). Therefore, we can use the relation between net leaf assimilation and temperature presented in Knapp (1985) to calibrate the JULES parameters governing the temperature dependence of $V_{cmax}$ in the model. The result is illustrated in Figure 4, alongside the parametrisations used in the `repro-cox-1998` and `global-C4-grass` runs. The lines calibrated to the Knapp (1985) observations peak at

approximately 38°C, whereas the `repro-cox-1998` and `global-C4-grass` parameterisations peak at approximately 32°C and 41°C respectively. This leads to very different model behaviour in the temperature range 32-42°C, where the `repro-cox-1998` parameterisation shows a dramatic decline in $V_{cmax}$, which contrasts sharply with the increase shown in the `global-C4-grass` parameterisation and the more stable lines calibrated to the Knapp (1985) observations. Note also that Polley et al. (1992) found 'no apparent relationship' between leaf temperature and net leaf carbon assimilation in

measurements of *A. gerardii*, *S. nutans* and *P. virgatum*, taken at ambient temperatures between 24.1°C and 47.8°C. They speculate that the difference between their results and the temperature relations found by Knapp (1985) is due to seasonal acclimatisation.

As already stated, for the `tune-leaf` configuration, we use JULES parameters fit to the *A. gerardii* data from Knapp (1985), since *A. geradii* is the dominant species at this site. However, to investigate the uncertainty introduced by the variation

between species, we repeat the runs using parameters fitted to the approximate midpoint of *A. gerardii* and *P. virgatum* light response curves and $V_{cmax}$ temperature relations. We would expect that the best parameter set to lie between these two parameterisations. However, note that Knapp (1985) does not have data for *Sorghastrum nutans*, the second-most dominant plant species at FIFE site 4439, so we were not able to take this species into account in this part of the calibration.

It should also be noted that Knapp (1985) reported a drop in the ratio of leaf nitrogen to leaf dry mass over the course of

the 1982 season of more than 50% in the burned plots. This could be a contributing factor to the drop in leaf assimilation they observed over the course of 1983. We were not able to incorporate a time-varying ratio of leaf nitrogen to leaf dry mass into our simulations, which could lead to an overestimation of leaf assimilation in the senescence period.

There were also gas exchange measurements on individual leaves of *A. gerardii*, *S. nutans* and *P. virgatum* taken as part of the FIFE intensive field campaigns in 1987 (Polley et al., 1992). These observations were taken on upper

canopy leaves perpendicular to the direct beam of the Sun, with varying absorbed PAR and internal $CO_2$ concentrations (FIFE_PHO_LEAF_46). This includes observations taken before, during and after the dry spell. Therefore, if we are to use these observations to calibrate the unstressed model parameters, we have to process them in such as way as to minimise the influence of the parameterisation of water stress in the model.

To achieve this, we identified individual net leaf assimilation ($A_l$) versus leaf internal $CO_2$ concentration ($c_i$) curves from

the FIFE_PHO_LEAF_46 dataset for *A. gerardii* and *P. virgatum* (using the observation time and leaf area). We normalised each $A_l$-$c_i$ curve using the mean $A_l$ for $c_i > 150\mu$ mol $CO_2$ (mol air)$^{-1}$ for that curve. We then selected $A_l$-$c_i$ curves with mean incident radiation greater than 1200 $\mu$mol PAR m$^{-2}$ s$^{-1}$. This procedure minimises the dependence on water stress or individual leaf nitrogen levels, since these factors approximately cancel out in the relations used internally in JULES when they are manipulated in this way. We can then use these normalised curves to calibrate the model $A_l$-$c_i$ response at low $c_i$. For

*A. gerardii* and, to a lesser extent, *P. virgatum*, this leads to a decrease in the initial slope of the $A_l$-$c_i$ curve (Figure 5).

We also attempted to use the $A_l$-$c_i$ curves identified in the FIFE_PHO_LEAF_46 dataset to calibrate the parameters in the JULES relationship between internal leaf $CO_2$ concentration and external $CO_2$ concentration $c_a$. Each individual $A_l$-$c_i$ curve was taken at approximately constant humidity, and $c_a$ is also provided for each point on the curve. JULES uses the Jacobs (1994) parameterisation

$$\frac{c_i - \Gamma}{c_a - \Gamma} = f_0 \left( 1 - \frac{dq}{dq_{crit}} \right), \qquad \qquad (1)$$

where $\Gamma$ is the photorespiration compensation point ($\Gamma = 0$ for C4), $dq$ is specific humidity deficit at the leaf surface. $f_0$ and $dq_{crit}$ are plant-dependent parameters: $f_0$ is a scaling factor on $c_i$ and $dq_{crit}$ governs the strength of humidity dependence of $c_i$. This parameterisation predicts that plotting $c_i$ against $c_a$ at constant humidity would give a straight line, with gradient $f_0 \left( 1 - \frac{dq}{dq_{crit}} \right)$. However, when plotting observations from FIFE_PHO_LEAF_46, we found that the slope of the $c_i$-$c_a$

relationship changed as $c_a$ increased (see Figure S8 in the supplementary material). Therefore, we were unable to calibrate the JULES $c_i$-$c_a$ relationship to this data.

Instead, we use leaf measurements of C4 grass in the Konza prairie, collected in 2008 and published as part of Lin et al. (2015). These were taken at ambient $CO_2$ levels, under unstressed conditions. We can derive the $c_i/c_a$ ratio from the supplied stomatal conductance, net assimilation and internal $CO_2$ observations, and plot this against specific humidity deficit at the

leaf surface, calculated from chamber VPD, neglecting the effect of the leaf boundary layer (Figure 6). We calibrate the Jacobs model parameters $f_0$ and $dq_{crit}$ to this data (green solid line). Given the large scatter of the data and resulting poor fit ($R^2$=0.04), we will also explore the effect of varying $dq_{crit}$ (green dashed lines a,b,c). In each case, $f_0$ is set to best fit this dataset for this $dq_{crit}$ (the parameter values are given in Table A3).

Both Knapp (1985) and Polley et al. (1992) found that leaf stomatal conductance $g_s$ is proportional to the net leaf assimilation

at this site. Their results are approximately consistent with the Lin et al. (2015) observations, given the difference in ambient $CO_2$ levels and the weak dependence on VPD.

As discussed above, in JULES, dark leaf respiration $R_d$ is calculated from model $V_{cmax}$, scaled by a constant. For the tune-leaf simulation, we tune this constant such that the model dark leaf respiration at 30°C matches the dark leaf respiration from Polley et al. (1992) at 30°C (Figure 7). This is roughly double the dark leaf respiration at 30°C in the

repro-cox-1998 and global-C4-grass configurations. The Polley et al. (1992) relation was fitted to observations made at leaf temperatures of approximately 14-46°C. While our tuned model parameterisation of dark leaf respiration compares reasonably well in the range 25-35°C, it rapidly diverges from the Polley et al. (1992) observations beyond this range. This is particularly true for the higher temperature values, where the observations in Polley et al. (1992) show an increase with temperature, whereas the tune-leaf JULES configuration shows a decrease.

Polley et al. (1992) found no significant difference between *A. gerardii*, *S. nutans* and *P. virgatum* for a variety of leaf properties: net leaf assimilation under ambient conditions, maximum assimilation under high light and $CO_2$ saturation, temperature response of net assimilation and relationship between assimilation and stomatal conductance under ambient conditions. This implies that the uncertainty we have introduced by not considering *S. nutans* data throughout most of this calibration is relatively minor.

### 2.3.2 Onset of water stress and relationship between water stress and leaf water potential

In this section, we calibrate the parameter governing the onset of soil water stress in the model, $p_0$. In the `repro-cox-1998` and `global-C4-grass` simulations, $p_0$ is set to 1, meaning that the model vegetation starts to experience soil water stress at a volumetric soil moisture $\theta=\theta_{crit}=0.387$ m$^3$ m$^{-3}$ (Figure 1). This leads to a soil moisture stress factor $\beta$ of 0.75-0.55 during the first 10 days of June 1987, i.e. a reduction of 25-45% compared to the case where model vegetation is not limited by water availability (Figure 2).

We can investigate this in more detail using leaf water potential observations as an indicator of the stress levels of the vegetation. Leaf water potential is affected by both the soil water content and the atmospheric water content, as well as other factors affecting transpiration. Both Polley et al. (1992) and Knapp (1985) found a relationship between leaf water potential and net leaf assimilation in their measurements of grasses in the FIFE study area. Polley et al. (1992) measured leaves of *A. gerardii* and *S. nutans* throughout the 1988 growing season. These observations showed a drop in net leaf carbon assimilation as the leaf water potential declined through the season: leaf water potentials -0.34 to -1.5 MPa were consistent with net leaf carbon assimilate rates of 16.2 to 41.5 $\mu$mol m$^2$ s$^{-1}$ whereas lower leaf water potentials of -1.5 to -2.45MPa were consistent with lower rates of 3.9 to 15.5 $\mu$mol m$^2$ s$^{-1}$ (at internal $CO_2$ concentrations of 200 $\mu$mol mol$^{-1}$ and absorbed PAR of 1600 $\mu$mol absorbed quanta m$^2$ s$^{-1}$)). Knapp (1985) carried out weekly leaf water potential measurements of *A. gerardii* and *P. virgatum* in 1983 for late May to early October, which showed midday leaf water potential dropping from -0.4MPa in late May to less than -6.6MPa (the pressure chamber limit) at the end of July. During this period, net leaf assimilation dropped from approximately 40$\mu$mol m$^2$ s$^{-1}$ to less than 10$\mu$mol m$^2$ s$^{-1}$.

Kim and Verma (1991b) proposed a model which considers the prairie vegetation to be completely unstressed until the leaf water potential drops below -1 MPa. This was partially motivated by the Polley et al. (1992) measurements and evaluated using observations of FIFE site 4439 in 1987, i.e. the same site and time period we use in this study. Kim and Verma (1991a) proposed an alternative water stress model, also based on data in Polley et al. (1992), where both the maximum rate of carboxylation of Rubisco $V_{cmax}$ and the maximum rate of carboxylation allowed by electron transport $J_{max}$ had a dependence on leaf water potential. According to this parameterisation, a leaf water potential of -0.4 MPa introduces a factor of 0.97 into $V_{cmax}$, for example, and a leaf water potential of -0.8 MPa introduces a factor of 0.91.

Midday leaf water potential for *A. gerardii* in the burned plot was approximately -0.4 MPa during the Knapp (1985) 'early season' measurement period. Therefore, according to both the Kim and Verma (1991b) and Kim and Verma (1991a) models, considering this period 'unstressed' is a very good approximation (i.e. $\beta = 1$, to within 3%), and agrees with their statement that "water was not limiting" the vegetation during this period. This validates our use of the Knapp (1985) data set to tune the 'unstressed' JULES parameters in the previous section.

We can now use the same arguments to determine how much water stress the vegetation should be experiencing at the beginning of June in our runs at FIFE site 4439 in 1987. Kim and Verma (1991a) present hourly leaf water potential measurements for *A. gerardii* leaves at this site, for a selection of days in 1987 (Figure 8). On 5th June 1987, they measured a minimum leaf water potential of approximately -0.8 MPa at 2pm local time. According to the Kim and Verma (1991b) model,

vegetation at this leaf water potential would not be water stressed, and according the Kim and Verma (1991a) model, $V_{cmax}$ would be reduced by approximately 9%. This contrasts sharply with the reduction in net assimilation throughout the day of 39%, due to water stress (i.e. $\beta = 0.61$), experienced in both the `repro-cox-1998` and `global-C4-grass` simulations on this day.

For the `tune-leaf` configuration, we therefore reduce the early season water stress, to be more consistent with Kim and Verma (1991a) and Kim and Verma (1991b). This can be achieved by introducing a non-zero $p_0$ value in the stress factor $\beta$. This reduces the soil moisture threshold at which the plant becomes completely unstressed ($\beta = 1$) from $\theta_{crit}$ to $\theta_{wilt} + (\theta_{crit} - \theta_{wilt})(1 - p_0)$, as illustrated in Figure 1. Assuming that the stress factor $\beta$ is 0.9 on 5th June 1987 leads to $p_0$=0.3. The effect of different values of $p_0$ will be shown in more detail in Section 3.

We now examine whether any previous modelling studies at this site support or conflict with this reduction in the soil moisture threshold at which the plant becomes completely unstressed. Crucially, the maximum soil moisture stress factor considered in the original Cox et al. (1998) study was 0.7, therefore a setup with a $p_0$ of 1-0.7=0.3 and parameters re-tuned to give a 30 % reduction in unstressed net leaf assimilation, would have given the same fit to the data. Similarly, a stress function with $p_0$=0.3 fits the plot of the ratio of actual to potential evapotranspiration to available water in Verma et al. (1992) (when corrected for their different soil properties) at least as well as a stress function with $p_0$=0. An increase in $p_0$ can also be considered a proxy for decreasing $\theta_{crit}$ (which, as we have already noted, has a large uncertainty: see Section A4). A $p_0$ of 0.2, for example, can be used to mimic the impact of changing $\theta_{crit}$ from 0.387, as used in this study and in Cox et al. (1998), to 0.348, as used in Verma et al. (1992).

   Kim and Verma (1991a) present hourly water potential measurement of *A. gerardii* leaves at FIFE site 4439 for 3 other days
(in addition to 5th June 1987): 2nd July (peak growth period), 30th July (dry period), 20th August 1987 (early senescence). These show a minimum of -1.2MPa, -2.6MPa and -1.7MPa respectively (Figure 8). Given the relationships between leaf water potential and net leaf assimilation described above, these leaf water potential measurements imply a drop in leaf assimilation during the middle of day in the dry period. In contrast, Polley et al. (1992) found 'no evident seasonal trend' in the maximum leaf assimilation rate or carboxylation efficiency, despite taking observations throughout the day before, during and after the dry
spell in 1987[2]. We were unable to reconcile these results satisfactorily using the associated data in the FIFE_PHO_LEAF_46 dataset (chamber vapour pressure, leaf and chamber $CO_2$ concentrations, leaf and chamber temperatures).

### 2.3.3   Canopy and optical properties

For the `tune-leaf` configuration, we keep the values of leaf reflectance and transmittance from `global-C4-grass`, as they are consistent with those measured by Walter-Shea et al. (1992) in 1988 and 1989 as part of the FIFE experiment.
Walter-Shea et al. (1992) found that leaf optical properties were not dependent on leaf water potential in the range -0.5 to -3.0 MPa. Leaf angle distribution measurements were taken as part of the FIFE campaign (SE-590_Leaf_Data), and tended towards erectophile (Privette, 1996). However, erectophile leaf angle distributions can not currently be set in JULES, so we continue to use a spherical angle distribution, as in the `global-C4-grass` run. Walter-Shea et al. (1992) noted that the

---

[2]Tim Arkebauer, personal communication, and timestamps from the FIFE_PHO_LEAF_46 dataset

leaf angle distribution of grass at FIFE site 4439 was affected by water availability: they concluded that severe water stress in 1988 probably contributed to a more vertical leaf orientation in 1988 than in 1989. The uniformity of the canopy in JULES can be parameterised by a canopy structure factor $a$ ($a = 1$ indicates a completely uniform canopy, $a < 1$ indicates clumping). It is difficult to get a numerical estimate of how uniform the canopy is at FIFE site 4439 because of the large uncertainties

in LAI measurements, which we discuss in Section A2. However, using LAI from Stewart and Verma (1992), together with FIFE observations of the fraction of absorbed photosynthetically active radiation (LB_UNL_42) on a day with mostly diffuse radiation (7th August 1987), gives a rough estimate for a canopy structure factor of 0.8. The structure factor changes the effective LAI seen by the model radiation scheme, and so can be used to investigate the effects of the uncertainty in the LAI dataset.

Leaves of *A. gerardii* roll (fold) in response to water stress, which reduces their sunlit area while still allowing photosynthesis to continue (Knapp, 1985). This dynamic response of the leaves to drought conditions could be an important factor in modelling canopy photosynthesis during dry spells. However, this behaviour is not implemented in the current version of JULES.

### 2.3.4 Summary of `tune-leaf` configuration

The `tune-leaf` configuration contains parameters that are, in theory, more appropriate to the tallgrass prairie vegetation

at this site, by tuning the underlying model processes to leaf and canopy measurements taken in the FIFE study area. The response of leaf photosynthesis to light, $CO_2$ and, particularly, temperature have been fitted to observations. We note that previous studies have indicated a relationship between leaf water potential and net leaf assimilation observations at this site, and that leaf water potential can be considered an indication of the water stress that the vegetation is experiencing. While JULES does not model leaf water potential explicitly, a review of the available leaf water potential observations measurements

indicates the need to delay the onset of model water stress in this tuned configuration, compared to the `repro-cox-1998` and `global-C4-grass` configurations, which we achieve through setting a non-zero $p_0$ parameter. We note that there remains significant uncertainty in the threshold for the onset of water stress, the calculation of internal $CO_2$ concentration and the uniformity of the canopy. There is also an uncertainty introduced by inter-species variation. We note that the comparison with observations has revealed some possible limitations of the model, such as the fixed leaf nitrogen content and leaf orientation

(spherical) through the season and an absence of leaf folding.

### 3 Results and discussion

Figure 9, Figure 10 and Figure 11 show the model output for gross primary productivity (GPP), net canopy assimilation and latent heat flux for eight days during 1987. These dates sample a range of different vegetation states: 5th June is in the early growth stage, 2nd July and 11th July are in the peak growth stage, 23rd July, 30th July and 11th August are in the dry period

and 17th August and 20th August are in the early senescence period (Verma et al., 1992). All of these dates comply with the selection criteria described in Cox et al. (1998) (following Stewart and Verma (1992)). Days with, or directly after, significant rainfall have been avoided, in order to reduce the effect of evaporation from the canopy surface and bare soil. The model latent

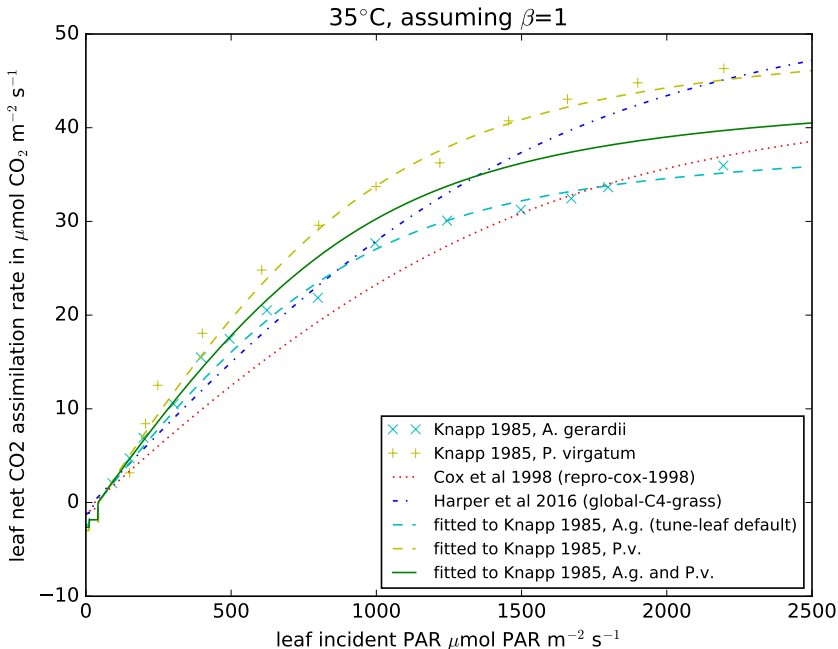

**Figure 3.** Mean observations from Figure 1 in Knapp (1985) from the burned plot, early season (May-June 1983) for *A. gerardii* (cyan diagonal crosses) and *P. virgatum.* (yellow vertical crosses) for net $CO_2$ assimilation rate against incident PAR, at $35\pm2°C$. JULES parameters are fitted to the *A. gerardii* observations (cyan dashed line), *P. virgatum.* (yellow dashed line) and a combination of both (green solid line). Also shown are the relations from the `repro-cox-1998` (red dotted line) and `global-C4-grass` runs (blue dot-dashed line), at $35°C$. Fitted lines assume no water stress (i.e. $\beta = 1$) and $c_i$=200 $\mu$ mol $CO_2$ (mol air)$^{-1}$. Model lines have been created using the Leaf Simulator package, which reproduces the internal JULES calculations.

heat flux is compared to latent heat flux measurements in the FIFE_SF30_ECV_33 dataset. GPP and net canopy assimilation are derived from $CO_2$ flux measurements in FIFE_SF30_ECV_33, using the method in Cox et al. (1998). Further net canopy assimilation estimates have also been read from Kim and Verma (1991a) (see Section A7 for more information).

### 3.1 `repro-cox-1998` and `global-C4-grass` **simulations**

5 GPP in the `repro-cox-1998` simulation after 10am local time compares very well to GPP derived from the flux tower data (Figure 9), for all growth stages. This is expected, given that this simulation is designed to reproduce the model from Cox et al. (1998), which was tuned to this flux dataset. The `global-C4-grass` simulation reproduces the carbon fluxes reasonably well outside the dry period, although GPP is underestimated during the growth stages. For example, GPP is underestimated by approximately 30% during the middle of the day on 5th June. During the dry period, however, the

10 `global-C4-grass` simulation poorly captures the early morning peak and subsequent decline in GPP indicated by the

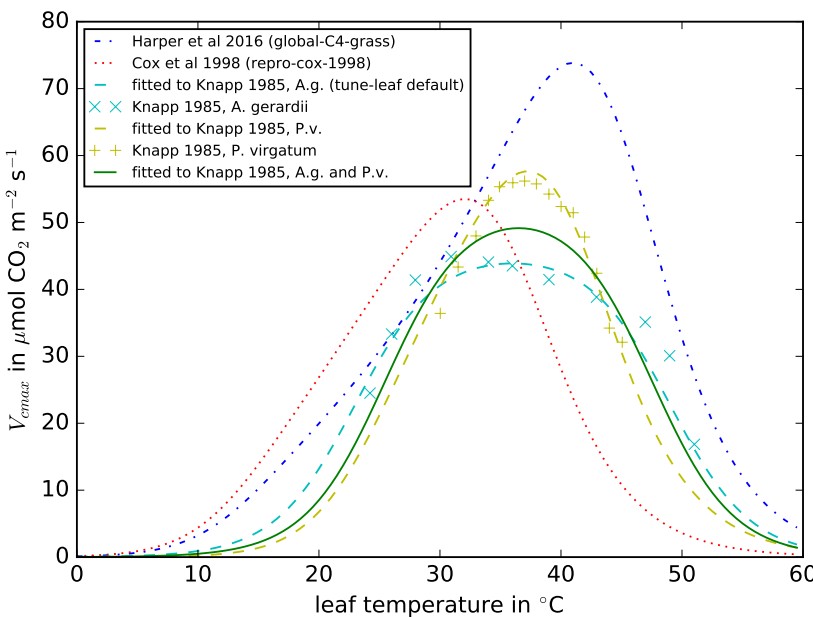

**Figure 4.** $V_{cmax}$ against leaf temperature for *A. gerardii* (cyan diagonal crosses) and *P. virgatum.* (yellow vertical crosses), using the normalised observations from Figure 2 in Knapp (1985), scaled using the fitted light response curves of *A. gerardii* and *P. virgatum* at 35°C shown in Figure 3. JULES parameters are fitted to these derived *A. gerardii* observations (cyan dashed line) and *P. virgatum.* observations (yellow dashed line) and a combination of both (green solid line). Also shown are the relations from the `repro-cox-1998` (red dotted line) and `global-C4-grass` runs (blue dot-dashed line). Model lines have been created using the Leaf Simulator package.

carbon flux observations. The `repro-cox-1998` run captures this behaviour through its response to leaf temperature. The diurnal cycle of air temperature on these days in shown in Figure S5 and modelled leaf temperature in Figure S6. Recall that $V_{cmax}$ in the `repro-cox-1998` simulation declines at leaf temperatures above 32°C. This causes a decline in modelled carbon assimilation during the hottest parts of the day (this is demonstrated explicitly in additional runs in the supplementary

5 material). However, as discussed in Section 2, the temperature response in the `repro-cox-1998` configuration is not supported by observations in Knapp (1985) or Polley et al. (1992). Therefore, it appears that, while the model is successfully capturing the shape of diurnal cycle during the dry period, it is not achieving this with the correct physical process.

Similarly, net canopy assimilation in the `repro-cox-1998` simulation compares well to the time series derived from the flux tower observations, although it has lower leaf respiration, particularly on 23rd July and 30th July Figure 10. As discussed

10 in Section A7, the leaf respiration values assumed when processing the flux measurements were based on observations of leaf respiration in Polley et al. (1992). In Section 2.3, we showed that the `repro-cox-1998` simulation underestimates leaf respiration compared to the Polley et al. (1992) dataset, particularly at the higher temperatures experienced during middle of

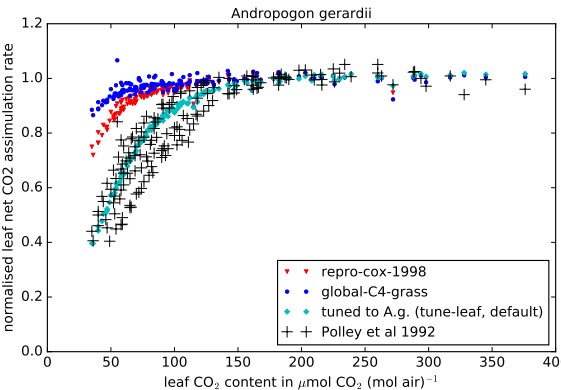 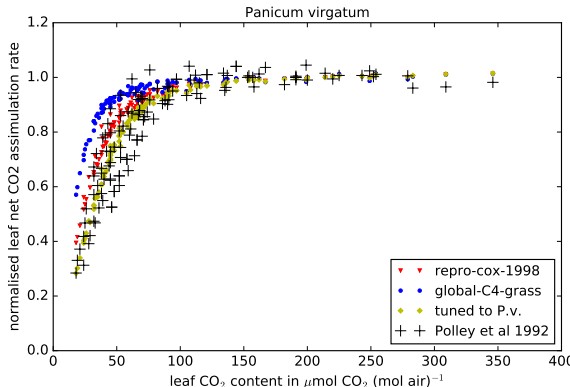

**Figure 5.** Black crosses: $A_l$-$c_i$ curves for *Andropogon gerardii* (left) and *Panicum virgatum* (right) from FIFE_PHO_LEAF_46 (Polley et al., 1992), normalised by the mean $A_l$ of the data points with $c_i > 150\mu$ mol $CO_2$ (mol air)$^{-1}$ in that curve. Only curves with mean incident PAR greater than 1200 $\mu$mol PAR m$^{-2}$ s$^{-1}$ have been used. Coloured points: normalised $A_l$ calculated from observed $c_i$ and incident PAR for each data point in the curve and the mean $T_{leaf}$ observation for each curve, using the JULES relations. The JULES parameters are taken from the `repro-cox-1998` configuration (red triangles), the `global-C4-grass` configuration (blue circles) and fits to A. g. data (`tune-leaf` default configuration, cyan diamonds) and P. v. data (yellow diamonds). Model points have been calculated using the Leaf Simulator package.

the day in the dry period. While the `global-C4-grass` configuration also simulates lower leaf respiration values than seen Polley et al. (1992), a combination of a low bias in the GPP and a peak in $V_{cmax}$ at higher temperatures (compared to the `repro-cox-1998` simulation) reduces the impact on net canopy assimilation.

The latent heat flux is reasonably well modelled in general in both the `repro-cox-1998` and `global-C4-grass` simulations outside the dry period (errors in the peak of the diurnal cycle of less than 20%). However both simulations overestimate the latent heat flux during the dry period (Figure 11). This is expected, given that we have already shown that the canopy carbon assimilation is overestimated, and stomatal conductance is proportional to the net leaf assimilation in the model.

### 3.2 `tune-leaf` **simulations**

The `tune-leaf` configuration generally overestimates both GPP (Figure 9) and net canopy assimilation (Figure 10) compared to the observations and the `repro-cox-1998` and `global-C4-grass` simulations. On days during the dry period, the `tune-leaf` simulation behaves characteristically similarly to the `global-C4-grass` simulation in that it also does not capture the mid-morning peak and subsequent decline in GPP and assimilation. When fitting the `tune-leaf` configuration in Section 2, we highlighted uncertainties in some of the key parameters, and we will now look at the effect of these in turn.

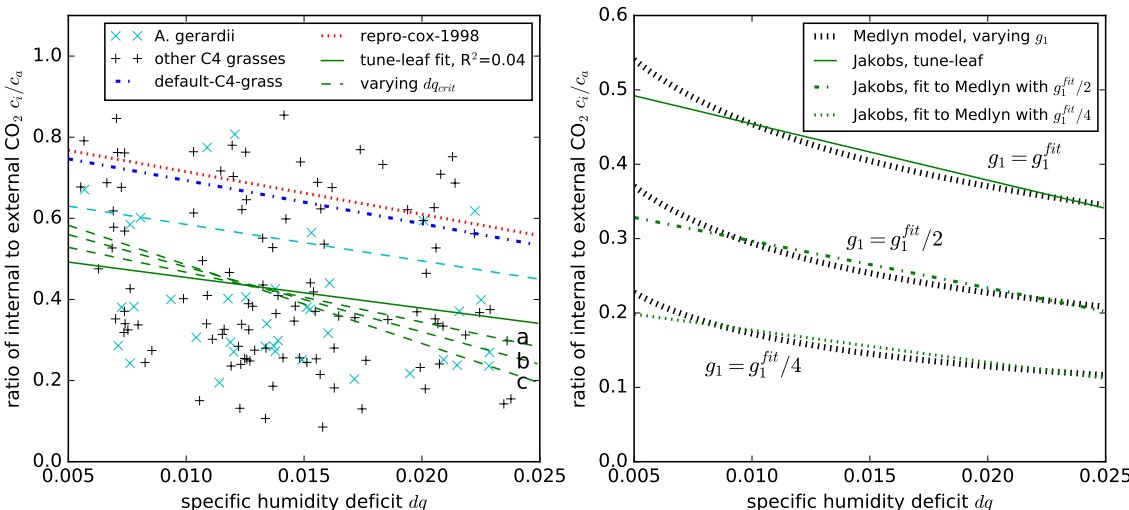

**Figure 6.** Ratio of internal to external $CO_2$ against specific humidity deficit $dq$. Crosses are derived from leaf measurements of *Andropogon gerardii* (cyan) and other C4 grasses (black), taken in the Konza prairie (Jesse Nippert and Troy Ocheltree, published in Lin et al. (2015)). Straight lines show Jacobs model for C4 plants i.e. $c_i/c_a = f_0 \left(1 - \frac{dq}{dq_{crit}}\right)$. Red dotted line: `repro-cox-1998`, blue dot-dashed line: `global-C4-grass`, green solid line: `tune-leaf`. Green dashed lines: varying $dq_{crit}$, and setting $f_0$ to the best fit to the Lin et al. (2015) data for this $dq_{crit}$. Black dotted lines: Medlyn model using $g_1^{fit}$, $g_1^{fit}/2$, $g_1^{fit}/4$, where $g_1^{fit}$ is the value of the Medlyn model parameter $g_1$ fitted in Lin et al. (2015)) to their Konza Prairie C4 grass measurements. The green solid line (the `tune-leaf` configuration) is a good approximation to the Medlyn model with $g_1 = g_1^{fit}$ (because they have both been fit to the same dataset). The green dot-dashed and green dotted lines have been tuned to be close to the Medlyn model lines with $g_1 = g_1^{fit}/2$ and $g_1 = g_1^{fit}/4$ respectively.

Firstly, the `tune-leaf` configuration is based on observations of the dominant grass species at this site, *A. gerardii*. In Section 2, we also fitted parameters to another grass species at this site: *P. virgatum*, and a 'combined' set fitted to both species. Since *A. gerardii* is almost twice as abundant at this site in 1987 as *P. virgatum*, and in the absence of parameter fits to the other grass species at this site, we would estimate that the most representative parameters lie somewhere between these two parameter sets. Using this combined *A.g./P.v.* parameter set increases GPP and net canopy assimilation on the order of roughly 10% compared to using the set fitted solely to *A.g.* (Figure 9, Figure 10), from which we conclude that the error introduced from using the dominant grass species is relatively minor.

A key difference between the `tune-leaf` configuration and the other configurations is the introduction of a non-zero $p_0$. Figure 12 illustrates that varying $p_0$ from 0 (as in the `repro-cox-1998` and `global-C4-grass` simulations) to 0.4 has a strong effect on GPP, as expected. It demonstrates the importance of ensuring that the threshold for water stress is consistent with the 'unstressed' leaf observations we calibrated against. Continuing to use $p_0$=0 with the newly-tuned unstressed parameters would have resulted in much too low GPP during the early growth period. Recall also that changing

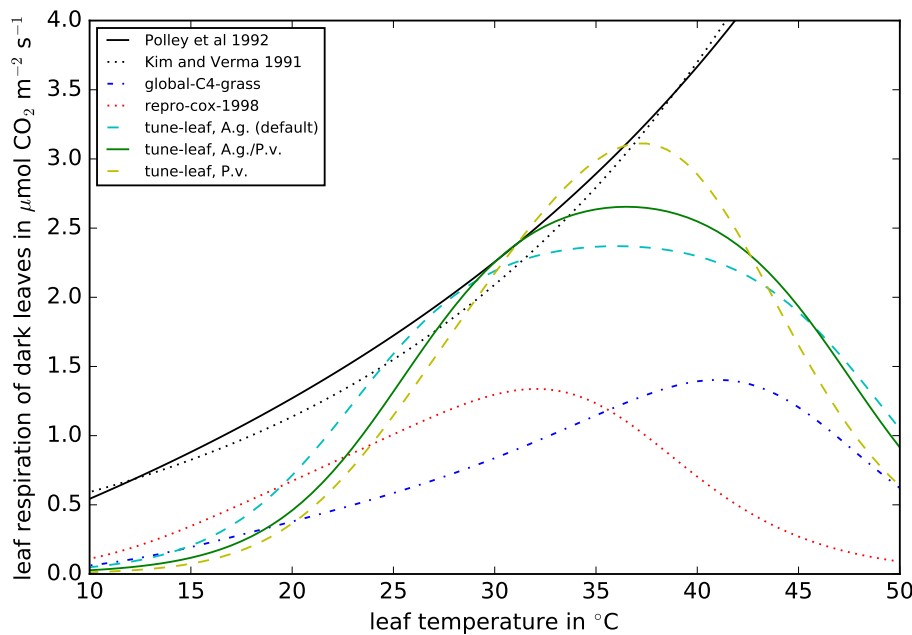

**Figure 7.** Comparison of leaf dark respiration against leaf temperature relations from Polley et al. (1992) (black solid line) Kim and Verma (1991a) (black dotted line), `repro-cox-1998` (red dotted line), `global-C4-grass` (blue dot-dashed line), tuned to A.g. (cyan dashed line), tuned to P.v. (yellow dashed line) and tuned to both A.g. and P.v. (green solid line). All lines assume no light inhibition of respiration. All JULES lines are top of the canopy (TOC) values without water stress. The lines that reproduce JULES configurations have been calculated using the Leaf Simulator package.

$p_0$ can be considered a proxy for changing the critical soil moisture. Therefore these runs also demonstrate the sensitivity to uncertainty in the soil properties.

The effect of varying the canopy structure factor on GPP can be seen in Figure 13. This can also be seen as a proxy for examining the effect of reducing LAI as it changes the effective LAI seen by the model radiation scheme. Varying the canopy

5 structure factor in the range 0.8-1.0 has a negligible effect on GPP on these days. However, reducing the canopy structure factor from 0.8 to 0.3 has a large, negative impact on GPP. As discussed in Section 2, this range is inside the error given in the LAI dataset documentation. The error in LAI for this site therefore has a large impact on the modelled canopy carbon fluxes.

Less straightforward to investigate is the effect of the uncertainty in the calibration of the JULES $c_i$ humidity response. Recall that the observational dataset used in Section 2 had a large spread in $c_i$ compared to its range of specific humidity

10 deficit values. This made it difficult to tune the parameter $dq_{crit}$ separately to the overall scaling factor $f_0$. We therefore take the approach of systematically varying $dq_{crit}$ (while setting $f_0$ to keep the best fit to the observations in Figure 6), to show qualitatively that a different humidity calibration can not improve the agreement with the GPP observations during the dry spell. Figure 14 compares modelled GPP for three different $dq_{crit}$, $f_0$ combinations: $dq_{crit}$ =0.048, $f_0$ =0.59 (upper green dashed

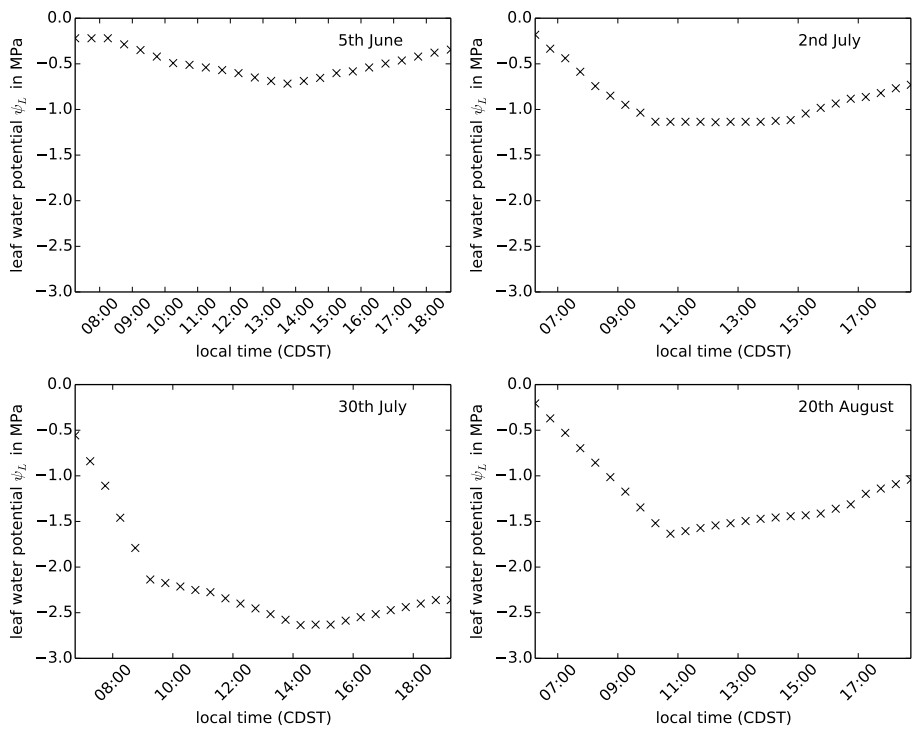

**Figure 8.** Leaf water potential observations for four days taken at FIFE site 4439 in 1987, published in Kim and Verma (1991a).

line), $dq_{crit}$ =0.040, $f_0$ =0.64 (central green dashed line) and $dq_{crit}$ =0.035, $f_0$ =0.68 (lower green dashed line) for four days during the dry spell. Plots of specific humidity deficit on these days are given Figure S7. None of these parameter combinations are able to fit the steady but low rate of GPP during the middle period of the day: they transition between almost no humidity-induced effect on GPP, to a sudden decline. The timing of this decline varies across the four days shown. This demonstrates that, while lower $c_i$ values in these runs during the day in the dry period can reduce GPP, the magnitude of the slope of $c_i/c_a$ against $dq$ is too large. These two effects can not be reconciled while still maintaining consistency with the unstressed observations in Lin et al. (2015). This implies that the Jacobs parameterisation used in JULES, where the relationship between $c_i/c_a$ and specific humidity deficit does not vary over the course of the run, does not have the flexibility needed to capture the behaviour of GPP at this site.

## 3.3  What potential model developments could improve the diurnal cycle of JULES GPP at this site?

As we have seen, the `global-C4-grass` configuration, which is typical of how this site would be modelled in a global JULES run, is unable to capture the diurnal cycle of GPP (and also net canopy assimilation and latent heat flux) at this site during the dry period in 1987. Replacing the generic C4 grass tile parameters with parameters that are calibrated to observations

taken of vegetation at this particular site (the `tune-leaf` configuration) does not improve ability of the model to capture the diurnal cycle in these fluxes. We have demonstrated that this conclusion is robust to uncertainties in LAI, soil moisture, leaf parameters, canopy parameters and soil parameters.

We will now explore a number of possible options for improving the standard representation of the dry period diurnal GPP cycle at this site. Firstly, the model diurnal cycle can be greatly improved via the careful selection of parameters in the existing leaf temperature-dependent calculation of $V_{cmax}$. This was demonstrated in the model runs in Cox et al. (1998), which we have closely reproduced with the `repro-cox-1998` configuration. This method has the advantage that it provides a close fit to data and does not require any changes to the model code. A disadvantage of this method is that the $V_{cmax}$ model parameterisation becomes an effective parameterisation which no longer has a clear biological interpretation. It therefore becomes more difficult to constrain from results in the literature. The numerical success of this method is due to high leaf temperatures acting as a proxy for high atmospheric demand during the middle of the day in the dry period (Figure S6 and Figure S7). While these temperature parameters provide a good approximation at this site in this particular year, it does not follow that these same temperature parameter values would be appropriate for other locations, or at this location under a changing climate.

Secondly, the model could be extended to include a soil moisture effect on the internal leaf $CO_2$ concentration $c_i$. As we demonstrated in Section 3, the current expression for $c_i$ in JULES can not simultaneously fit the unstressed observations and be able to reduce $c_i$ to the required levels to affect GPP during the dry season without also increasing the strength of the response to specific humidity deficit. This results in the humidity-induced stomatal closure occurring too suddenly on days during the dry period. Introducing a soil moisture dependence in $c_i$ would allow $c_i$ to be lower on days where soil water was limiting for all humidity levels, while maintaining the higher values on unstressed days. Zhou et al. (2013) and De Kauwe et al. (2015) both achieve this by adding a soil moisture dependence to the VPD term in the Medlyn conductance model (Medlyn et al., 2011). The Medlyn model is based on the theoretical argument that stomata should act to minimise the amount of water used per unit carbon gained, leading to a stomatal conductance $g_0 + 1.6 \left(1 + \frac{g_1}{\sqrt{D}}\right) \frac{A}{c_a}$, where $g_0$ and $g_1$ are free parameters.

As demonstrated in De Kauwe et al. (2015), the parameters in the Jacobs model ($f_0$, $dq_{crit}$) can be chosen so that the resulting $c_i : c_a$ ratio approximates the Medlyn model, for mid-range VPD values. The unstressed Konza Prairie C4 grass measurements used in Section 2 to calibrate the $c_i : c_a$ ratio in the `tune-leaf` configuration were actually provided in Lin et al. (2015) as part of a comprehensive study to tune the $g_1$ parameter in the Medlyn model for different vegetation types (with $g_0 = 0$). Using the Medlyn model with their calibrated $g_1$ value ($g_1^{fit} = 1.04$) does indeed give a similar $c_i/c_a$ to our `tune-leaf` configuration (Figure 6, solid green line).

Therefore, to investigate the effect of a soil moisture-dependent $g_1$ on GPP, we can set the JULES $c_i/c_a$ relation to mimic a lower $g_1$, and try this out on days with low soil moisture. For this test, we choose JULES parameter values that provide a rough approximation to the Medlyn model with $g_1 = g_1^{fit}/2$ and $g_1 = g_1^{fit}/4$ (Figure 6, dot-dashed and dotted green lines). These reductions in $g_1$ are well within the range observed in Zhou et al. (2013) for a range of different vegetation types under water-limited conditions. The resulting JULES parameter values are given in Table A3. Figure 15 demonstrates that lowering $c_i/c_a$ in this way is able to qualitatively reproduce the shape of the diurnal cycle of GPP in the dry period. The run mimicking

$g_1 = g_1^{fit}/2$, in particular, is a very good match to the observations. This shows the potential value of extending JULES to allow interaction between the plant response to soil moisture dependence and VPD.

Another way to implement this interaction in JULES would be to add a dependence on leaf water potential, since leaf water potential is affected by both soil moisture (water supply) and VPD (atmospheric water demand). As discussed in Section 2.3, there is an observed relationship between leaf water potential and leaf assimilation in grass species at this site.

Previous studies have demonstrated that models with an explicit dependence on leaf water potential can successfully capture the dry period diurnal cycle at this site. Kim and Verma (1991a) were able to qualitatively capture the mid-morning peak and subsequent decline in net canopy photosynthesis on 30th July at this site, using a model in which both $V_{cmax}$ and $J_{max}$ had a dependence on their leaf water potential measurements. Furthermore, Kim and Verma (1991b) were able to reproduce this behaviour in canopy conductance at this site on 30th July and 11th August 1987 using a model that included an explicit dependence on observed leaf water potential, in addition to a direct dependence on VPD.

Leaf water potential is not currently modelled explicitly within JULES. Typically, in plant hydraulic models, leaf water potential is calculated assuming a steady-state water balance, using the soil water potential, transpiration, and leaf-to-root and root-to-soil resistance terms (as in, e.g. Newman (1969)). Adding this to the JULES code is technically non-trivial as water stress is currently applied to leaf-level processes before transpiration is calculated. Also, modelling the plant resistances would require additional input parameters, which would need to be constrained from observations.

Stress parameterisations involving leaf water potential come in a range of complexities. The simplest involve inserting a leaf water potential-dependent stress factor into an existing part of the model e.g. the limiting photosynthesis rates as in Kim and Verma (1991a), or stomatal conductance, as in Kim and Verma (1991b) and Tuzet et al. (2003). More sophisticated models include the plant hydraulics as part of schemes incorporating risk-benefit analysis (e.g. Sperry et al. (2017); Eller et al. (2018)) and/or chemical signalling (e.g. Tardieu and Davies (1992); Dewar (2002); Huntingford et al. (2015)).

Finally, another way to improve the diurnal cycle of GPP in the dry period would be to incorporate a parameterisation of leaf rolling. For example, effective leaf area available to the radiation scheme could be decreased during hot, dry weather. Kim and Verma (1991a) attribute the residual overestimation of net canopy carbon assimilation on days during the dry period of their leaf water potential-based model to this effect. It would therefore be interesting to investigate the contribution that leaf rolling makes to the overall plant water use strategy. However, while the occurrence of leaf rolling/folding at the FIFE site has been recorded, the effect has not been quantified. This would be a necessary first step for modelling this process at this site.

## 3.4 Can the FIFE dataset make a useful contribution to current-day JULES evaluation and development work?

A global land-surface model such as JULES needs to perform well for a wide range of climate regimes, time scales, spatial scales and vegetation types. Model evaluation or development work needs to represent this variety. The availability of comprehensive databases, such as FLUXNET (Baldocchi et al., 2001) and TRY (Kattge et al., 2011), have revolutionised land-surface science by giving easy access to observations from a wide variety of sources, in a common format. Given this context, why would a modeller consider also using the FIFE dataset?

Firstly, FIFE provides an ideal case study for improving the model representation of water stress on carbon and water fluxes in JULES in tallgrass prairie. While, at one time, tallgrass prairie extended over 10% of the contiguous United States (Fierer et al., 2013), it has declined 82-99% since the 1830s due to agricultural use (Sampson and Knopf, 1994; Blair et al., 2014a). However, grasslands in general (including other grass- and graminoid-dominated habitats, such as savanna, open and closed shrubland, tundra) cover more terrestrial area than any other single biome type (up to 40 % of Earth's land surface (Blair et al., 2014a)). It is therefore important to include lots of examples of grasslands in any global analyses of vegetation responses to changing conditions. The Konza Prairie LTER site, where FIFE was based, has been used extensively to investigate the dynamics and trajectories of change in temperate grassland ecosystems, including drivers such as fire, grazing, climate, nutrient enrichment (see Blair et al. (2014b) for a review).

FIFE looked at the processes for representing water stress in detail, and intensively studied the relevant factors. This has led to a wide variety of complementary observations, and literature specifically focussing on how this data can be used to inform models. LAI is a good illustration of this advantage. As we have discussed, LAI is an important parameter for modelling canopy water and carbon fluxes. LAI was measured by multiple groups at FIFE, directly and indirectly, and the large differences found between the different attempts was fully explored at the time. We can use their results to inform our own use of these datasets.

When adding a new process to a global land-surface model, it is important to tune new parameters to a comprehesive range of datasets. For example, as mentioned in Section 3.3, Lin et al. (2015) use data for 314 species from 56 sites across the world to tune the new $g_1$ parameter introduced in the Medlyn model of stomatal conductance for key plant functional types. This breadth of sites and vegetation types is essential. Each site contributed leaf gas exchange observations taken under similar protocols to allow a carefully controlled common analysis.

Access to individual experiments, which have investigated the combined effect of a wide range of processes, such as FIFE, can play a complementary role in land-surface evaluation and development. For example, FIFE provides cases where improving an individual process in isolation degrades overall model performance. As we have shown, calibrating unstressed model $V_{cmax}$(T=25°C) from leaf observations without also calibrating when the model is considering the vegetation to be unstressed significantly underestimates early-season GPP. Similarly, tuning the model parameters to improve the fit to canopy GPP and evapotranspiration can result in an unrealistic temperature dependence of $V_{cmax}$. Looking at sites in a holistic way can also highlight complications or influences that might not *a priori* have been considered, such as leaf rolling in our case.

There are two main disadvantages to the use of FIFE in evaluation and model development studies. The first is the limited time period: observations are available for a period of up to three years, with some key measurements only undertaken during the intensive field campaigns. Where long term effects are being studied, alternative datasets would need to be used.

The second disadvantage is that it is relatively more time consuming to add FIFE to an evaluation study, compared to adding an extra site from one of the large, standardised databases such as FLUXNET. This is partly because FIFE provides a choice of different datasets to use for forcing, calibrating parameters and evaluation, which takes time to investigate. It is also partly because, although the data is easily downloadable, well documented and in common file formats, is still needs to be manipulated into a format that can be used in JULES runs. We aim to address this issue by providing a suite that can be

used to pre-process the FIFE data and run JULES with the configurations described in this manuscript (see the 'code and data availability' section).

This aim is central to the provision of this manuscript. FIFE is the first 'JULES golden site', a concept was launched at the annual JULES meeting 2018. A JULES golden site is a site targeted by the JULES community because it can help address one of the key science questions facing JULES and has high-quality observational data that can be used to drive JULES and evaluate the output. It creates a network of researchers within the JULES community with experience of how this site can be exploited for JULES development, with input from site investigators. A key component is the provision of shared runs and evaluation datasets, which can be gradually expanded and improved.

In our study, we have focussed on the contribution that FIFE can make to the development of water stress in JULES. This has governed the choices we have made when setting up our configurations, e.g. choosing to prescribe LAI and soil moisture. However, we note here that FIFE could also be used to investigate other processes, such as plant and soil respiration (Section A7), the seasonal decline in leaf nitrogen (Knapp, 1985) and the modelled energy balance (Kim and Verma, 1990a; Colello et al., 1998).

| | repro-cox-1998 | global-C4-grass | tune-leaf |
|---|---|---|---|
| Radiation | Site averaged product Betts and Ball (1998). No diffuse radiation needed. | Site averaged product Betts and Ball (1998). Diffuse radiation from shortwave radiation using method in Weiss and Norman (1985). | Site averaged product Betts and Ball (1998). Diffuse radiation from shortwave radiation using method in Weiss and Norman (1985). |
| Other met. data | Site averaged product Betts and Ball (1998), apart from air pressure, which is set to a constant. | Site averaged product Betts and Ball (1998). | Site averaged product Betts and Ball (1998). |
| Leaf Area Index | Prescribed using obs. from Stewart and Verma (1992) | Prescribed using obs. from Stewart and Verma (1992) | Prescribed using obs. from Stewart and Verma (1992) |
| Canopy height | Prescribed using obs. from Verma et al. (1992) | Prescribed using obs. from Verma et al. (1992) | Prescribed using obs. from Verma et al. (1992) |
| Soil layers | 0.1m, 0.25m, 0.75m and 2.0m | 0.1m, 0.25m, 0.75m and 2.0m | 0.1m, 0.25m, 0.75m and 2.0m |
| Soil moisture | Prescribed using obs. from Stewart and Verma (1992), no variation with depth. | Prescribed using obs. from Stewart and Verma (1992), variation with depth obtained from pre-processing with an offline version of the JULES hydrology code. | Prescribed using obs. from Stewart and Verma (1992), variation with depth obtained from pre-processing with an offline version of the JULES hydrology code. |
| Wilting and critical vol. soil moisture | Cox et al. (1998). | Cox et al. (1998). | Cox et al. (1998). |
| Soil moisture stress | One stress factor calculated ('bucket approach'). $p_0 = 0$. | Stress factor for each soil layer weighted by root distribution. $p_0 = 0$. | Stress factor for each soil layer weighted by root distribution. $p_0 = 0.3$. |
| Canopy scheme | 'Big leaf' approximation. | Layered canopy, direct and diffuse beams, sunflecks. | Layered canopy, direct and diffuse beams, sunflecks. |
| PFT parameters (see Table A2) | Cox et al. (1998) | Harper et al. (2016) | Some tuning to site observations, as described in Section 2. |

**Table 1.** Model settings for the runs at FIFE site 4439 for 1987. Further descriptions of the model setup can be found in Section 2 and the choice of FIFE observations in Section A.

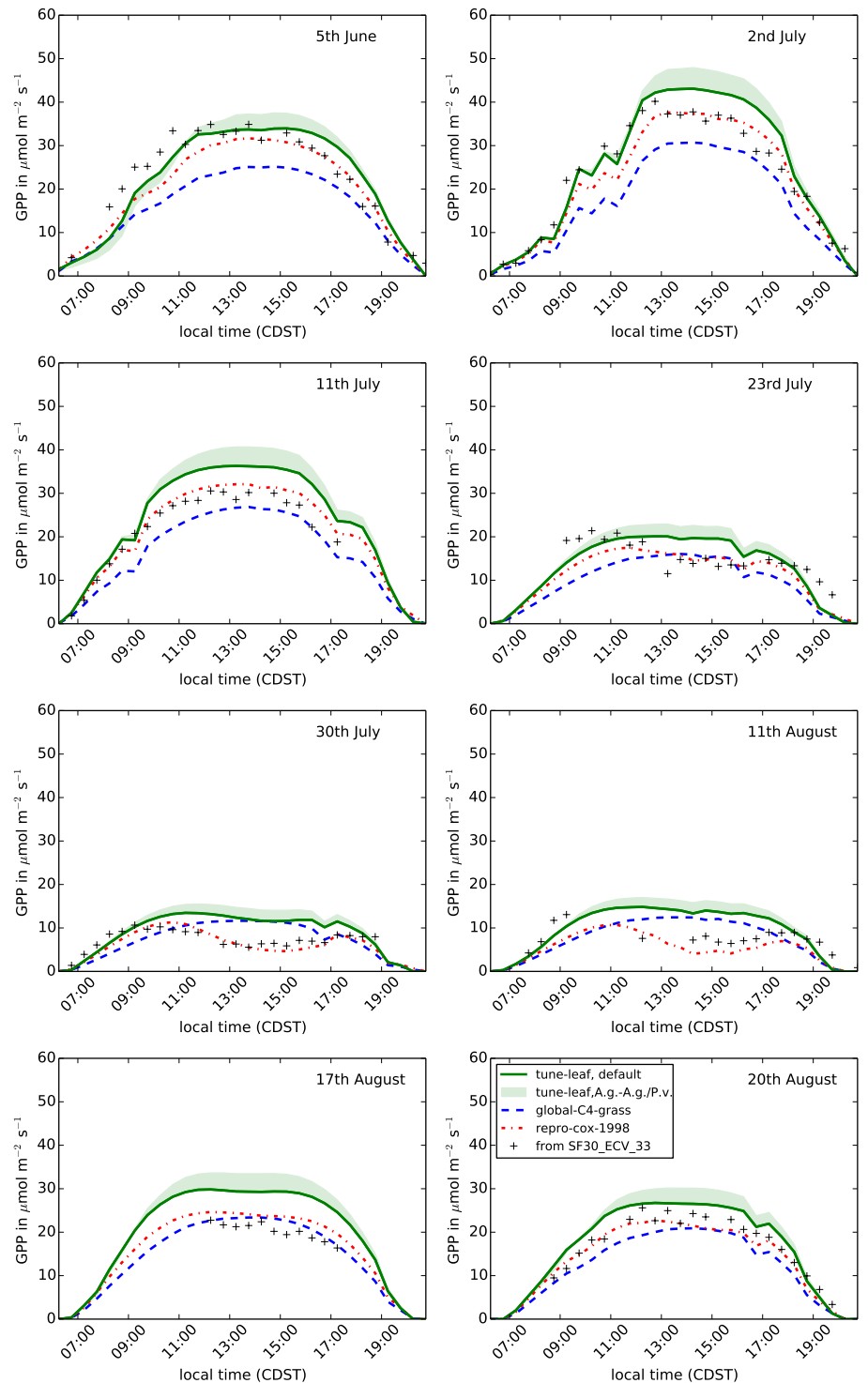

**Figure 9.** The diurnal cycle of GPP at site 4439 in the FIFE area for 8 days in 1987: 5th June (early growth), 2nd July and 11th July (peak growth), 23rd July, 30th July and 11th August (dry period) and 17th August and 20th August (early senescence). Green band show uncertainty from fitting plant parameters to *A. gerardii* compared to fitting to both *A. gerardii* and *P. virgatum*.

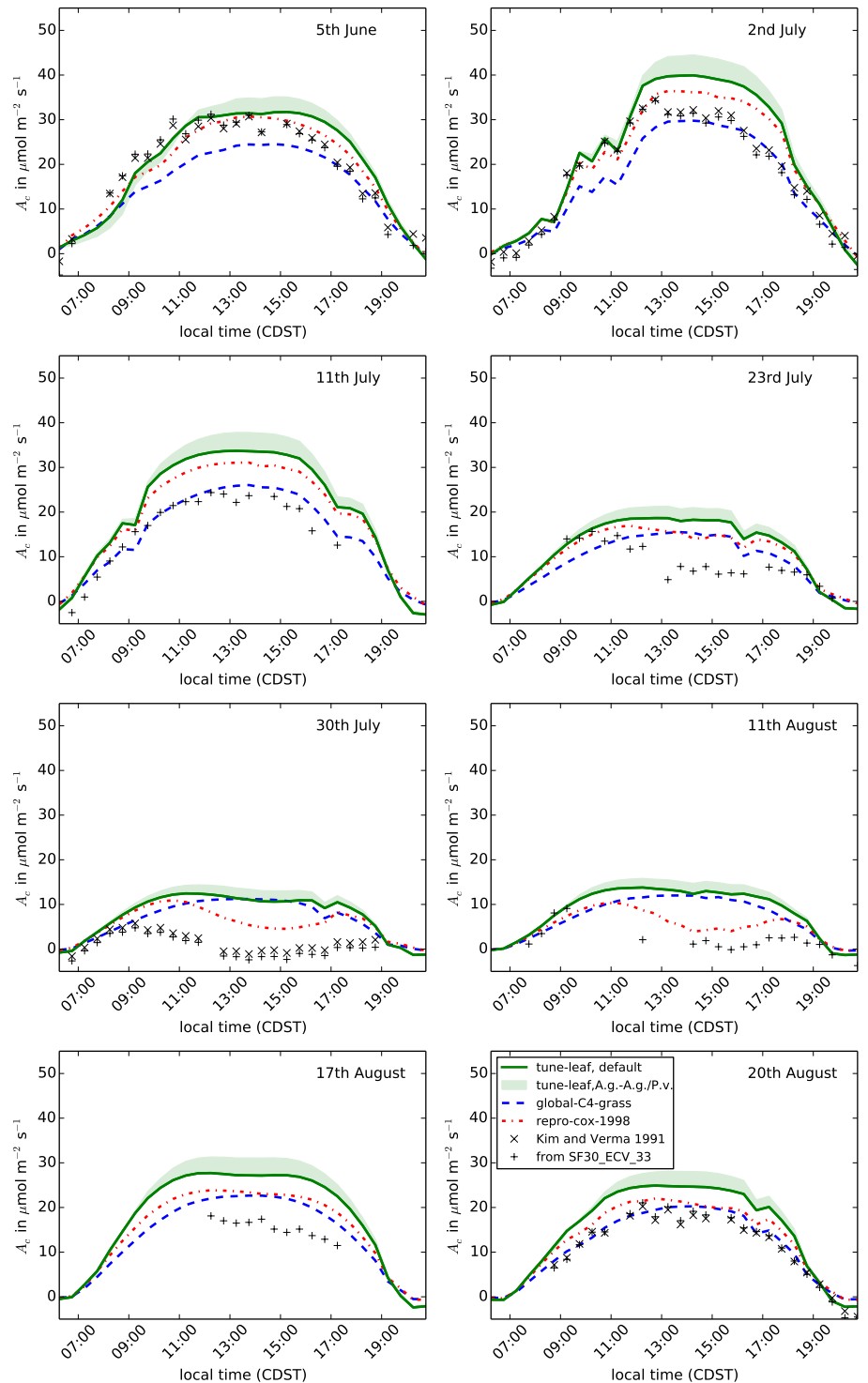

**Figure 10.** The diurnal cycle of net canopy assimilation $A_c$ at site 4439 in the FIFE area for 8 days in 1987: 5th June (early growth), 2nd July and 11th July (peak growth), 23rd July, 30th July and 11th August (dry period) and 17th August and 20th August (early senescence). Green band show uncertainty from fitting plant parameters to *A. gerardii* compared to fitting to both *A. gerardii* and *P. virgatum*.

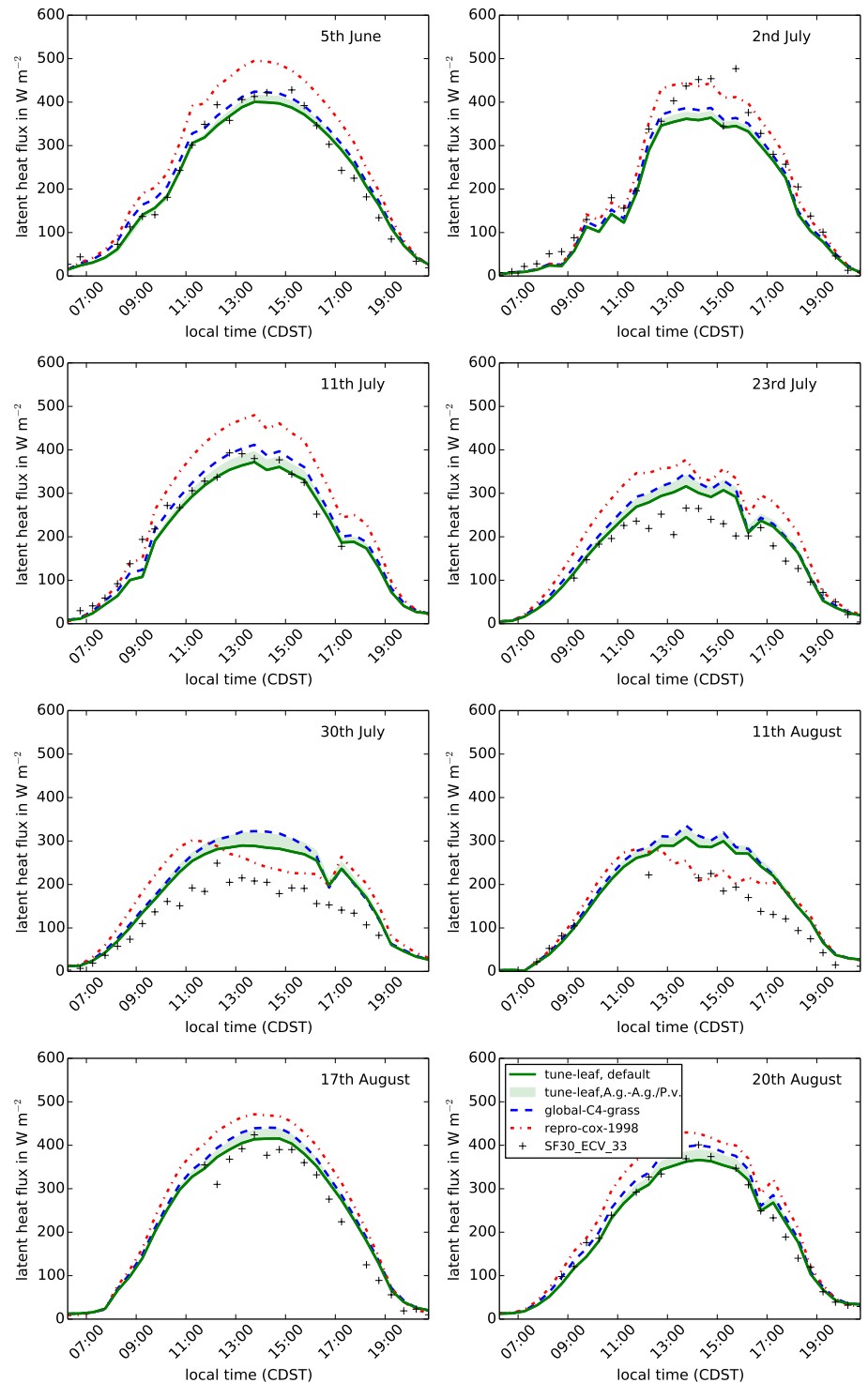

**Figure 11.** The diurnal cycle of latent heat flux at site 4439 in the FIFE area for 8 days in 1987: 5th June (early growth), 2nd July and 11th July (peak growth), 23rd July, 30th July and 11th August (dry period) and 17th August and 20th August (early senescence). Green band show uncertainty from fitting plant parameters to *A. gerardii* compared to fitting to both *A. gerardii* and *P. virgatum* (upper limit corresponds to the combined *A. g., P. v.* fit, lower limit to the *A.g.* fit (i.e. the default `tune-leaf` configuration).

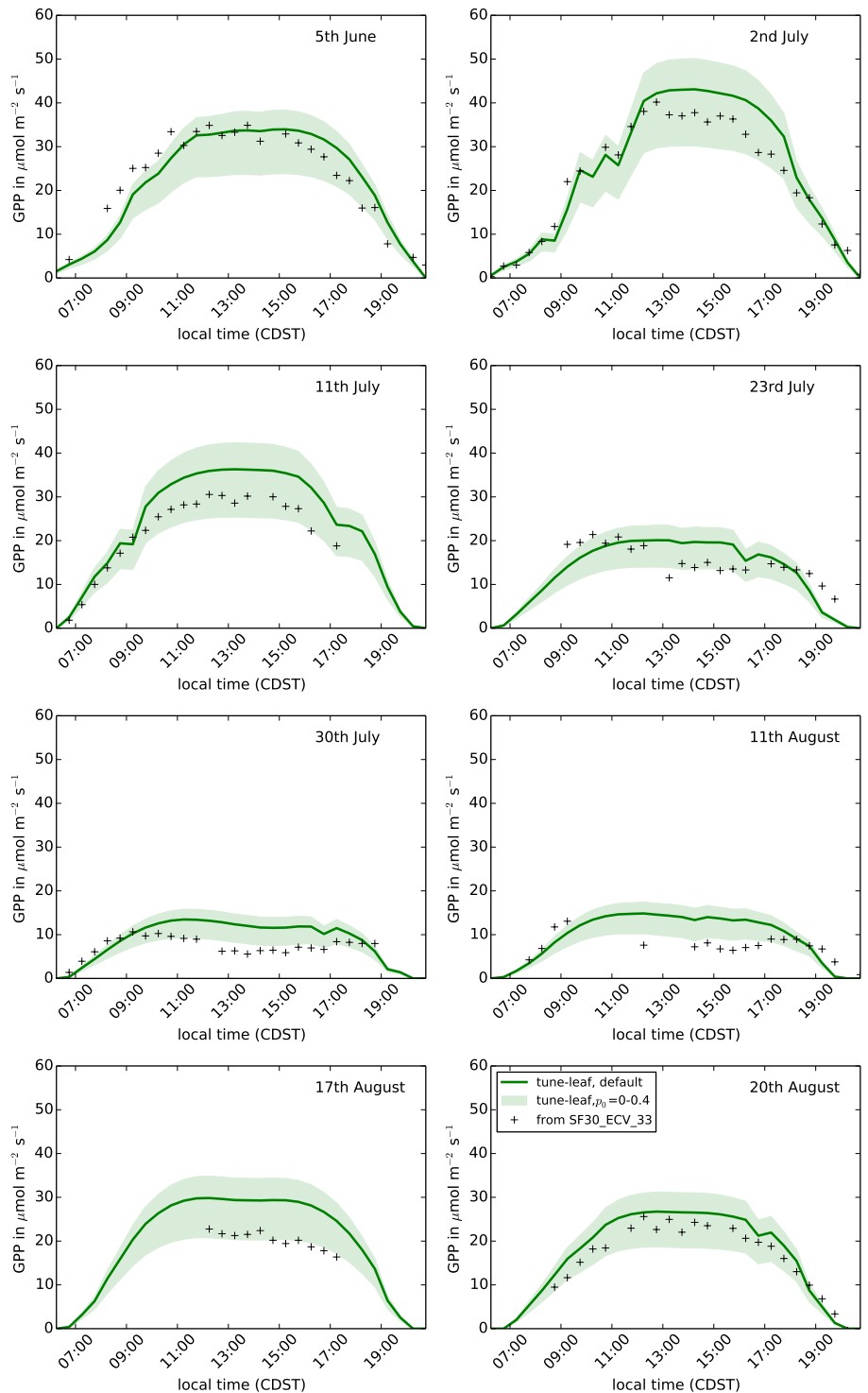

**Figure 12.** The diurnal cycle of GPP at site 4439 in the FIFE area for 8 days in 1987: 5th June (early growth), 2nd July and 11th July (peak growth), 23rd July, 30th July and 11th August (dry period) and 17th August and 20th August (early senescence). Green band shows how `tune-leaf` simulation would vary for $p_0$ in the range 0 to 0.4 (lower limit corresponds to $p_0$=0, upper limit to $p_0$=0.4).

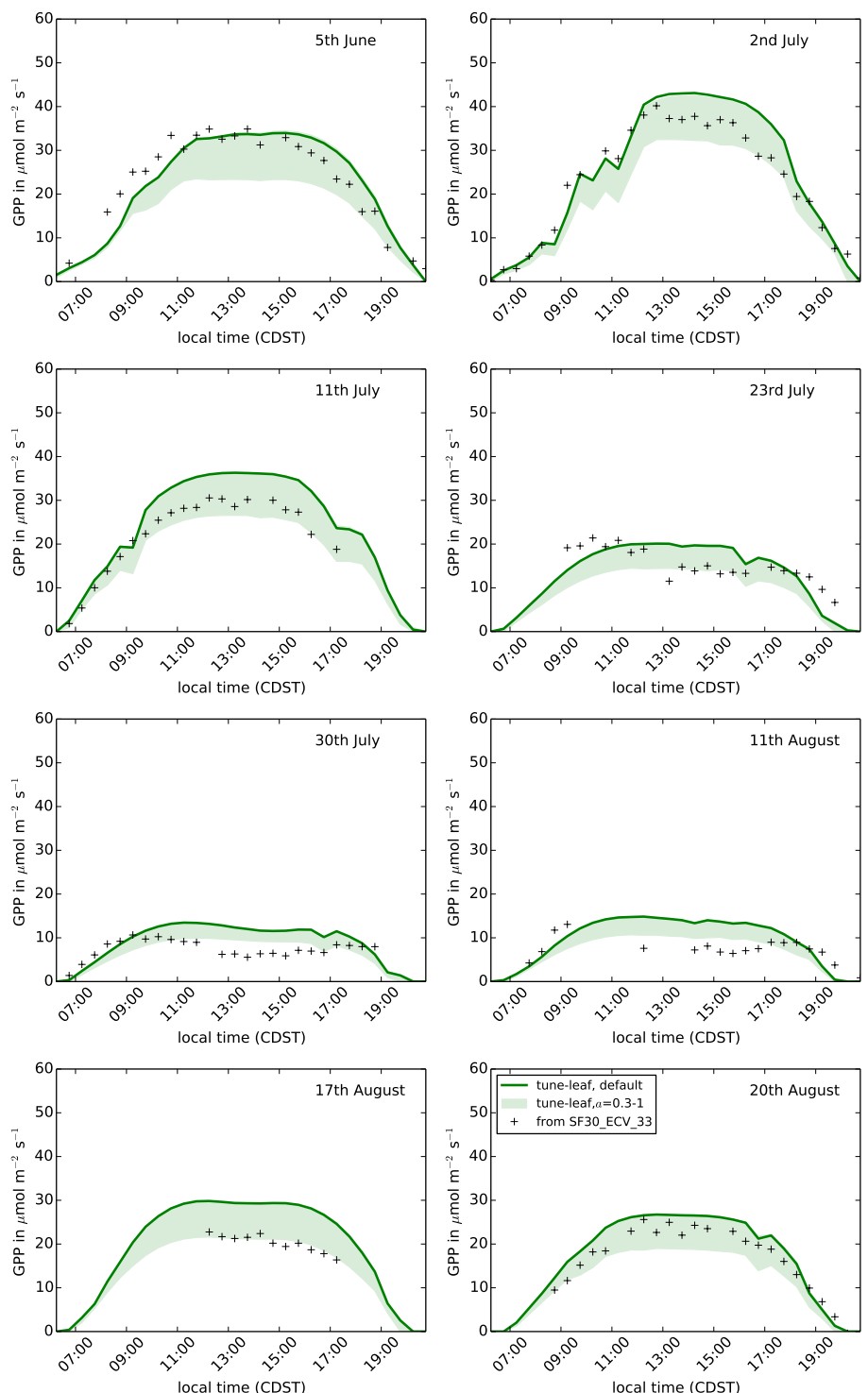

**Figure 13.** The diurnal cycle of GPP at site 4439 in the FIFE area for 8 days in 1987: 5th June (early growth), 2nd July and 11th July (peak growth), 23rd July, 30th July and 11th August (dry period) and 17th August and 20th August (early senescence). Green band shows how `tune-leaf` simulation would vary for a canopy structure factor $a$ in the range 0.3 to 1 (upper limit corresponds to $a$=1, lower limit to $a$=0.3).

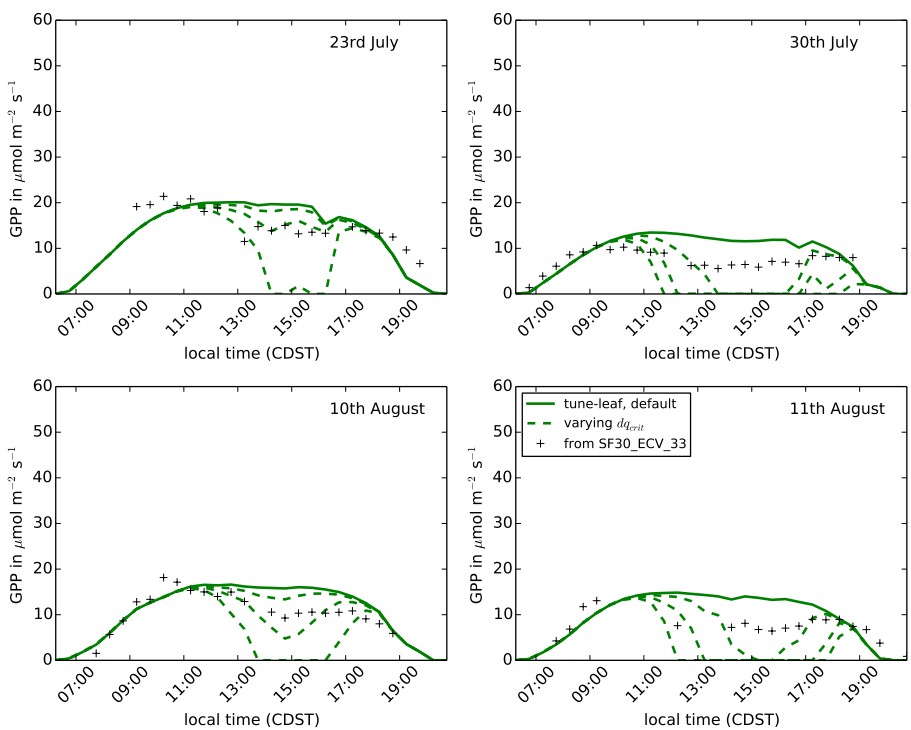

**Figure 14.** The diurnal cycle of GPP at site 4439 in the FIFE area for 4 days in during the dry period of 1987. Solid green lines uses the `tune-leaf` configuration. Dashed dashed green lines show how GPP varies if $dq_{crit}$ is increased, while $f_0$ is changed to maintain the best fit to the Konza prairie C4 grass observations in Lin et al. (2015) (Upper, middle and lower dashed lines correspond to parameter combinations a, b, c respectively, as defined in Table A3).

## 4 Conclusions

In their closing remarks, Sellers and Hall (1992) state that "FIFE created an environment for the discussion of all aspects of the land surface component of Earth remote sensing and Earth system modeling and provided a data set which has been and continues to be used to test models and algorithms." Our study demonstrates that this is still the case, twenty-five years after this remark, and thirty years since the experiment itself. There is a wealth of available data and extensive analysis in the literature, particularly on the response of vegetation carbon and water fluxes to periods of low water availability.

Historically, FIFE observations were used to derive the original soil moisture stress parametrisation in JULES. This early model was extremely successful in fitting the canopy net assimilation and water fluxes, during both dry and wet periods (Cox et al., 1998). However, a typical modern-day configuration of JULES, from Harper et al. (2016), which models the FIFE vegetation with generic C4 grass parameters, could not reproduce the observed diurnal cycle of carbon and water fluxes during

the period of low water availability. Calibrating the plant parameters to site observations did not solve this problem, nor could it be explained by the large observational uncertainties in leaf area index, soil moisture, and soil properties. Reproducing the original configuration in Cox et al. (1998) illustrated that the temperature dependence of the maximum rate of carboxylation of Rubisco $V_{cmax}$ in the model was key for reducing modelled photosynthesis rates during the hottest parts of the day in the dry period, since model $V_{cmax}$ declined steeply at the leaf temperatures experienced on these days. However, this temperature response was not supported by the available leaf-level gas exchange observations. With a more realistic temperature response, this configuration was no longer able to capture the reduction of photosynthesis during the middle of the day in the dry period either.

FIFE therefore provides a robust example of how the current processes that govern the way that vegetation in JULES responds to water availability do not behave realistically during dry spells for this type of grassland. This deficiency could be addressed by allowing the effect of soil moisture availability and vapour pressure deficit on stomatal conductance to interact, for example, via leaf water potential. FIFE is thus a useful site to consider when evaluating the benefits of new water stress parameterisations to JULES, particularly those with an explicit representation of plant hydraulics.

FIFE can play a role in JULES evaluation and development only as one small component of a comprehensive range of datasets, covering different climate regimes, time scales, spatial scales and vegetation types. FIFE is valuable partly due to the concentration of overlapping datasets. Key observables such as leaf area index, soil moisture, and soil properties, from independent investigations during FIFE, have been intensively analysed and yet still show a wide spread. This illustrates the intrinsic variability of these parameters, which must be carefully considered when scaling up to gridded, global runs. FIFE also provides clear examples of how calibrating one process to observations can reduce the overall model performance, due to compensating biases (such as calibrating the unstressed parameters without also checking the time period during which the model considers the vegetation to be unstressed). Confidence that the model is capturing key processes is necessary if the model is being run into new regimes, such as when forced with climate projections. This ability to disentangle and evaluate individual processes emphasises the value that intensive experiments such as FIFE have towards the larger modelling community evaluation efforts. In order to facilitate the inclusion of FIFE data in comprehensive model evaluations, this manuscript is accompanied by a release of the full set of data processing and configuration files needed to reproduce these model simulations. It is intended that this suite of files will continue to develop in the future as additional parts of the model are evaluated against the FIFE dataset, so that the JULES community can build up a comprehensive body of knowledge of data and model runs at this site.

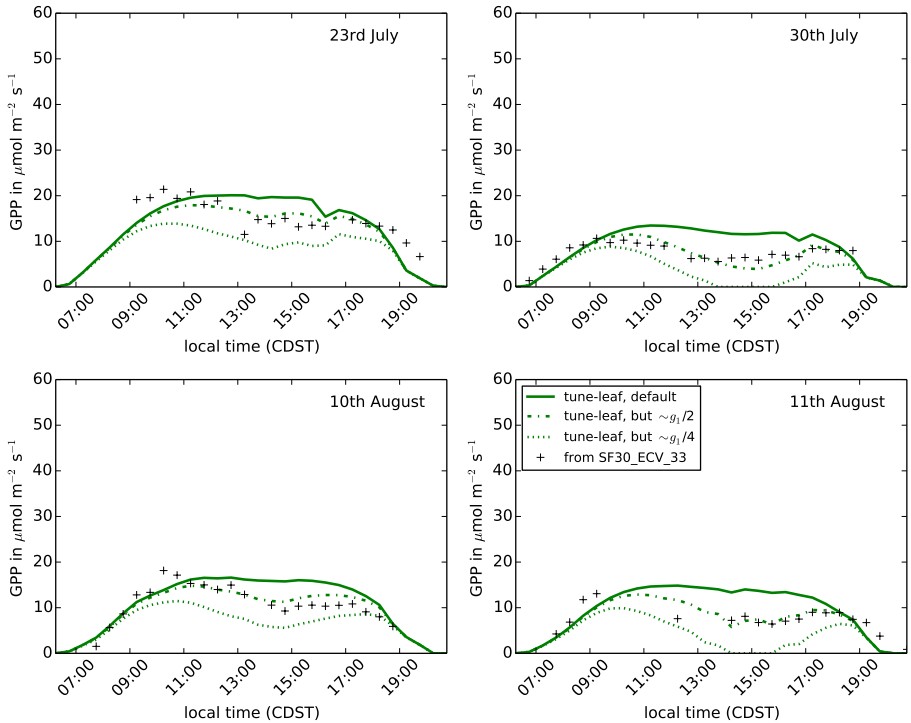

**Figure 15.** The diurnal cycle of GPP at site 4439 in the FIFE area for 4 days in during the dry period of 1987. Solid green lines uses the `tune-leaf` configuration. The $c_i$ to $c_a$ ratio in this configuration closely corresponds to the $c_i$ to $c_a$ ratio for C4 grasses in the Konza prairie in Lin et al. (2015), using the Medlyn model and fitting the Medlyn model parameter $g_1$ to measurements taken in 2008 ($g_1 = g_1^{fit}$=1.04 kPa$^{-0.5}$). The dot-dashed lines and dotted lines show the results from fitting the JULES parameters $dq_{crit}$ and $f_0$ to approximate the Medlyn model when $g_1 = g_1^{fit}/2$ and $g_1 = g_1^{fit}/4$ (the parameter values are given in Table A3).

*Code and data availability.* JULES can be downloaded from the JULES FCM repository on the Met Office Science Repository Service at https://code.metoffice.gov.uk/trac/jules (registration required). We use JULES version 5.0 (tag 'vn5.0'), which corresponds to revision 9522. The Leaf Simulator can be downloaded from https://code.metoffice.gov.uk/trac/utils. Where data points have been read directly from published plots, this was done with the EasyNData tool (Uwer, 2007). The three JULES simulations described in this study can
5   be reproduced using the rose suite u-bb181, available at https://code.metoffice.gov.uk/trac/roses-u/browser/b/b/1/8/1/trunk. This suite also contains instructions for downloading the driving data from ORNL-DAAC and a script to pre-process the driving data, including calculating the diffuse radiation fraction.

## Appendix A: FIFE observations

This section discusses the use of the observations and the alternative datasets considered. All of these datasets are available either in the published literature or available for download from the Oak Ridge National Laboratory (ORNL) Distributed Active Archive Center (DAAC). A list of all the ORNL-DAAC datasets referred to in this manuscript is given in Table A1.

### A1   Driving data

This study used a 30 minute resolution combined data product (FIFE_FFOAMS87_88) from observations from Portable Automatic Meteorological Stations (AMS) across the FIFE area, described in Betts and Ball (1998). Descriptions and references to all the FIFE datasets available from Oak Ridge National Laboratory Distributed Active Archive Center, are given in Table A1. Extensive manual processing was undertaken to clean the station data before it was combined into the site-averaged data product (Betts and Ball, 1998).

The fraction of diffuse radiation is an important driving variable when the full layered canopy scheme is used in JULES Mercado et al. (2007), although it is frequently not available and so set to a constant. For our study, we calculate diffuse radiation from shortwave radiation using the method in Weiss and Norman (1985). This method was used successfully at the FIFE site by Kim and Verma (1991a) and Kim and Verma (1991b). We also investigated using the hourly cloud observations of Marshall AAF, KS, approximately 12 km west of the FIFE site, which were included as part of the FIFE_FFOAMS87_88 dataset, which we converted to diffuse radiation fraction using the linear relationship given in Butt et al. (2010). This relationship was derived for two sites in the Amazon, but we confirmed that this was approximately consistent with observations of sites in the Southern Great Plains region of Oklahoma and Kansas in Still et al. (2009). However, we found that the cloud cover observations were not sufficiently consistent with the shortwave radiation used to drive the model runs. There are also total cloud cover observations from the FIFE area available in FIFE_FFOAMS87_88, but this had a period of missing data between the end of August and the middle of September. It would be interesting to compare these results to the approximation for diffuse radiation used by Gu et al. (2002) for a tallgrass prairie site in Oklahoma.

Colello et al. (1998) also carried out model runs driven by the site-averaged product FIFE_FFOAMS87_88, and applied corrections to shortwave downward radiation, longwave downward radiation and wind speed using observations from site 4439. In our study, we do not apply local corrections to the site-averaged meteorological data. However, this may be useful to consider in the future.

### A2   Leaf area index

The green Leaf Area Index values used in this paper are destructive measurements for FIFE site 4439, read from Figure 1 of Stewart and Verma (1992), which were taken roughly once a fortnight between 26th May and 11th October 1987. These observations are plotted in Figure A1. They correspond closely to the green LAI observations from Verma et al. (1992) and are similar to the green LAI observations for this site given in Sellers et al. (1992) for the intensive field campaigns. The LAI values used in the Cox et al. (1998) modelling study are very similar to these datasets. Destructive LAI measurements for

grass LAI, non-grass LAI and total LAI are available as part of the FIFE_VEG_BIOP_135 dataset. However, the total LAI in FIFE_VEG_BIOP_135 is substantially different from the measurements in Stewart and Verma (1992), Verma et al. (1992) and Sellers et al. (1992). This was investigated in detail at the time (Kim et al., 1989). The FIFE_VEG_BIOP_135 dataset documentation estimates that there is standard error of the mean LAI in their data of around 75% due to the inherent variability of prairie vegetation and a variation of about 25% can be attributed to leaf curling or folding as the leaves passed over the detector, particularly an issue for drought-stressed leaves. Foliage Area Index measurements (i.e. includes green leaves, dead leaves, stems) are available in FIFE_LB_UNL_42 for site 4439 in 1987, and plotted in Figure A2. FIFE_LIGHTWND_43 and FIFE_LB_KSU_41 also have Foliage Area Index measurements for site 4439, but these were taken in 1988-9, not 1987.

We also experimented with the internal phenology scheme in JULES. Calculating LAI dynamically with the phenology scheme would remove the need to prescribe LAI. However, we found that this scheme did not have the flexibility to reproduce the observed seasonal cycle of LAI.

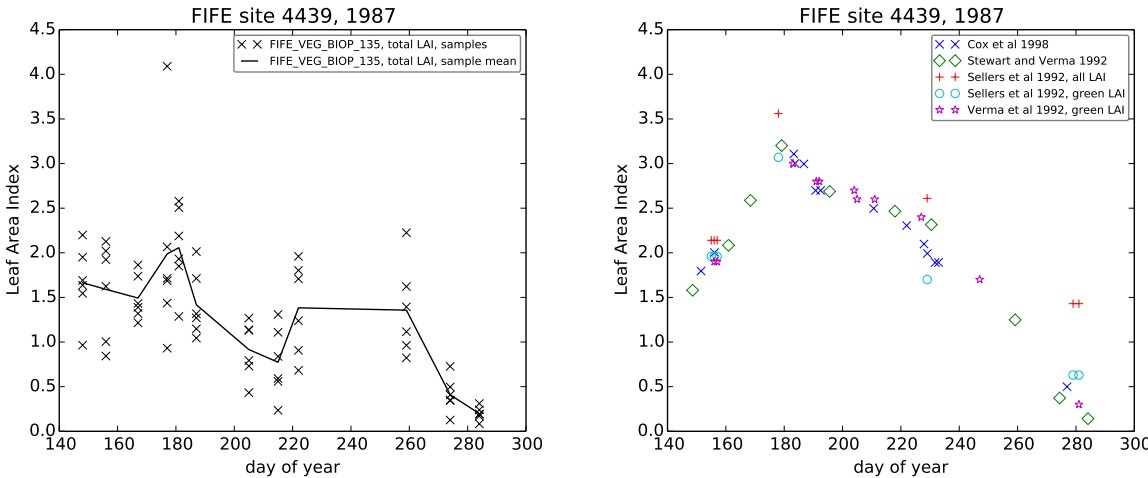

**Figure A1.** Leaf area index observations for site 4439 for 1987. Left: data from FIFE_VEG_BIOP_135. Right: literature values. Plot includes data extracted from Stewart and Verma (1992) Figure 1 and Cox et al. (1998) Figure 1, total LAI and green LAI from Sellers et al. (1992) for the intensive field campaigns and green LAI data from Table 4 in Verma et al. (1992).

## A3   Soil moisture

The soil moisture data for site 4439 presented in Figure 1 of Stewart and Verma (1992) were created from a combination gravimetric measurements and neutron probe measurements. The gravimetric measurements were taken in the top 0.1m soil daily during the FIFE intensive field campaigns and weekly between campaigns. The neutron probe measurements were taken at different depths on 15 dates, at approximately weekly intervals between the end of May and the beginning of September 2017. These measurements were interpolated in Stewart and Verma (1992) using daily precipitation and evaporation measurements

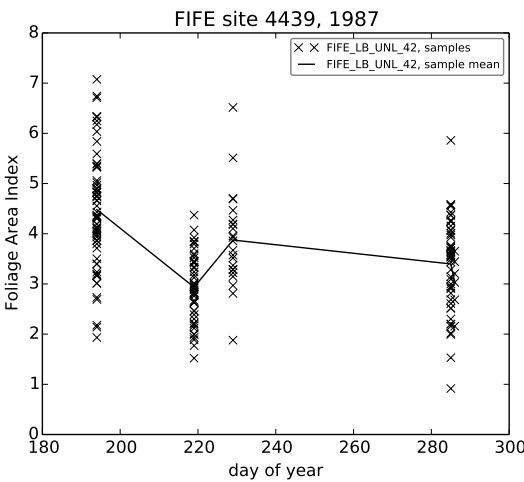

**Figure A2.** Foliage Area Index observations from FIFE_LB_UNL_42 for site 4439 in 1987.

to get a daily soil moisture values for the 0-1.1m soil layer. Stewart and Verma (1992) also observed 'virtually no seasonal variation' in soil moisture below 1.1m. The data from Stewart and Verma (1992) for the top 1.1m of soil corresponds very closely to the 0-1.6m soil moisture values used in Cox et al. (1998) on their selected days, as illustrated in Figure A3. Stewart and Verma (1992) also presents data for an ungrazed site in the FIFE area, and state that, while the ungrazed and grazed sites

5  received very similar season totals of precipitation, individual storms resulted in differences in soil moisture (which gives a possible motivation for using site 4439 precipitation measurements over the site-averaged data product we use here).

ORNL-DAAC contains two main datasets of soil moisture observations on levels that can be considered for site 4439 for 1987: FIFE_SM_NEUT_111, which contains measurements carried out at site 4439 and FIFE_FFONEU87_100, which is a site-averaged product for the FIFE area (Betts and Ball, 1998). These are plotted in Figure A4 for 1987. It can be seen that,

10  at lower depths, the site 4439 measurements are considerably lower than the site-averaged product. For 1988, however, the site-averaged product is mostly within or near the edge of the spread of observations at site 4439, up to approximately 120cm. Neither of these datasets are consistent with the Stewart and Verma (1992) site 4439 dataset when summed over the top 1.1m. The FIFE_SM_NEUT_111 for site 8639, on the other hand, is consistent with the Stewart and Verma (1992) site 8739 dataset. The documentation for FIFE_FFONEU87_100 also cautions that the 20cm neutron probe data is 'suspect' as the range of

15  the probe exceeds 20cm in dry soil and says that it is 'inconsistent' with the rest of the profile in 1987. It has been linearly interpolated between observation dates. Plots of observed soil profiles for 9th July and 31st July 1987 are presented in Kim and Verma (1990a). Soil profiles for individual days are also presented in Colello et al. (1998), which are consistent with the neutron probe measurements in FIFE_SM_NEUT_111, but not the gravimetric measurements. Given these inconsistencies, we chose not to use the soil moisture observations for individual levels to directly drive our simulations.

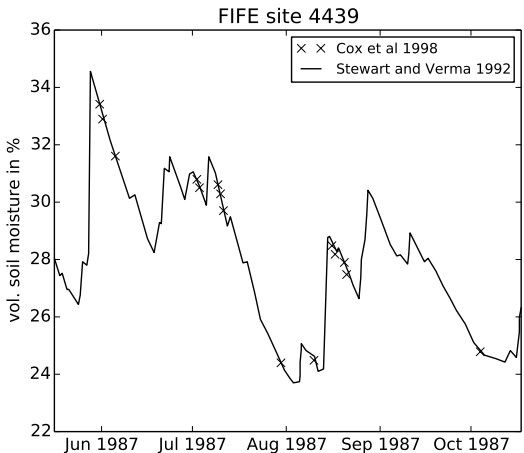

**Figure A3.** Soil moisture data from Cox et al. (1998), compared to the derived time series of top 1.1m soil moisture in Figure 1 of Stewart and Verma (1992). Both datasets are for FIFE site 4439 in 1987.

| Folder name | Dataset Reference | Description |
|---|---|---|
| FIFE_FFOAMS87_88 | Betts (1994a) | Site Averaged AMS Data, published in Betts and Ball (1998) |
| FIFE_PHO_LEAF_46 | Norman (1994a) | Leaf Photosynthesis Rates, published in Polley et al. (1992) |
| FIFE_VEG_BIOP_135 | Nelson et al. (1994) | Biophysical properties of the vegetation at the FIFE study area collected for FIFE by the staff of the Evapotranspiration Laboratory at Kansas State University under the direction of E.T. Kanemasu, and by the staff of the University of Nebraska, Lincoln under the direction of B. Blad. The dedicated efforts of A. Nelson, J. Killeen, L. Ballou, T. Shah, and C. Hays in collecting and preparing these data is particularly appreciated. |
| FIFE_LEAF_H2O_126 | Blad and Walter-Shea (1994b) | Total Leaf Tissue Water Potential data collected by B.L. Blad, E.A. Walter-Shea, C.J. Hays, and M.A. Mesarch of the University of Nebraska. |
| FIFE_LB_UNL_42 | Blad and Walter-Shea (1994a) | Leaf Area Index and PAR Determined from the UNL Light Bar Data collected under the direction of B.L. Blad and E.A. Walter-Shea at the University of Nebraska. The dedicated efforts of C.J. Hays and M.A. Mesarch in the collection and preparation of these data is particularly appreciated. |
| FIFE_LIGHTWND_43 | Shah and Kanemasu (1994b) | Indirect Leaf Area Index Obtained from the KSU Light Wand data collected by staff of Kansas State University under the direction of T. Shaw and E.T. Kanemasu. The contribution of these data is appreciated. |
| FIFE_LB_KSU_41 | Shah and Kanemasu (1994a) | Leaf Area Index and PAR Determined from KSU Light Bar Measurements collected as part of the KSU staff science effort directed by Dr. E.T. Kanemasu. |
| FIFE_SM_NEUT_111 | Kanemasu (1994a) | Neutron Probe Soil Moisture data collected for FIFE by the staff and students of the Evapotranspiration Laboratory at Kansas State University under the direction of Dr. Edward Kanemasu. The dedicated effort of A. Nelson, T. Shah and G. Harbers in the collection and preparation of these data is particularly appreciated. |
| FIFE_FFONEU87_100 | Betts (1994b) | Site Averaged Neutron Soil Moisture (Betts and Ball, 1998) |
| FIFE_SF30_ECV_33 | Verma (1994) | Eddy Correlation Surface Flux Observations (UNL) collected by Dr. Shashi B. Verma. |
| FIFE_SOIL_CO2_105 | Norman (1994b) | Soil $CO_2$ flux data, published in Norman et al. (1992). |
| FIFE_PHO_BOX_27 | Asrar and Sellers (1994) | Canopy Photosynthesis data collected by G. Asrar of NASA HQ and P.J. Sellers of NASA Goddard Space Flight Center. |
| FIFE_SOILSURV_115 | Huemmrich and Levine (1994) | Soil survey data obtained by the FIFE Information System staff from the US Dept. of Agriculture, Soil Conservation Service (USDA-SCS). Thanks are due to Dr. Elissa Levine who was instrumental in acquiring, interpreting, and preparing these data. |
| FIFE_SOILDERV_117 | Sellers and Huemmrich (1994) | Soil Water Properties data set produced by the FIFE Information System staff using data collected by the FIFE staff science team. |
| FIFE_SOIL_REL_112 | Hope and Peck (1994) | FIFE Level-3 Example Gridded Soil Moisture data provided by Drs. A. Hope and E. Peck. The assistance of Dr. James Wang, NASA, in furnishing PBMR soil moisture data was sincerely appreciated. Thanks to the FIS staff, especially Fred Huemmrich, Diana van Elburg-Obler and Jeff Newcomer, for providing information in such usable form. Thanks also to Eric Wood, Princeton University, for providing soil moisture data in digital form for the Kings Creek Basin. |
| FIFE_SOILHYDC_107 | Kanemasu (1994b) | Soil Hydraulic Conductivity data collected for FIFE by the staff of the Evapotranspiration Lab at Kansas State University. |
| FIFE_SOILREFL_114 | Huemmrich (1994) | Soil Reflectance Reference data obtained by the FIFE Information System staff from Stoner et al., 1980. The permission of Stoner et. al. to use these data is greatly appreciated. |

**Table A1.** List of FIFE datasets from ORNL-DAAC referenced in this document. Each dataset is referred to by its folder name.

### A3.1  Derived soil moisture

In order to create a daily soil moisture time series on levels, which could be used to drive the `global-C4-grass` and `tune-leaf` runs, we used a python implementation of the JULES hydrology scheme. The soil layer thicknesses used were the same as in Harper et al. (2016), apart from the third soil layer, which was extended by 10cm. This meant that the total depth

of the top three layers was 1.1m, which meant that we could constrain the sum of the soil moisture in the top three levels in our runs to be equal to the daily 0-1.1m soil moisture values from Stewart and Verma (1992). We assumed that positive changes in the 0-1.1m soil moisture were due to rainfall (with runoff, canopy evaporation and soil evaporation from that day already subtracted) and therefore added it to the top layer, while negative changes in the 0-1.1 m soil moisture were assumed to be due to transpiration (corrected for the transpiration flux from the lowest level and the flux between the lowest and second-to lowest

layer), which was taken from the soil layers according to an exponential root distribution with efold depth $d_r$ =0.5m. This $d_r$ depth is the same as natural C4 grass in Harper et al. (2016). We used the same soil hydrological parameters as in our JULES simulations (described in Section A4).

The resulting derived soil moisture timeseries are shown in Figure A5 (left). As expected, the upper levels show more variability than the lower levels, which is consistent with the sitegrid 4439 and site-averaged soil moisture time series on levels

(see Section A3) and approximately with the statement in Stewart and Verma (1992) that there was 'virtually no seasonal variation' below 1.1m. Figure A5 (right) compares the derived time series for soil moisture in the top soil level (10cm thickness) to the gravimetric soil moisture data for 2.5cm and 7.5cm from FIFE_SM_NEUT_111. While the fit is reasonable, given the spread in observations, it appears to indicate that the variability in the top level soil moisture is still underestimated. This could be due to the assumed root distribution (a lower $d_r$ would lead to more water extracted from the upper layer), or the

approximation that soil evaporation can be neglected on days without rainfall, or approximations made by Stewart and Verma (1992) when deriving the 1.1m soil moisture timeseries.

We also attempted two other methods for deriving a soil moisture time series on levels from Stewart and Verma (1992): using the transpiration from the `repro-cox-1998` run and editing the `repro-cox-1998` run so that soil moisture was no longer prescribed. The first method did not perform well, possibly due to the transpiration and soil moisture time series

not quite being in step with each other. The second method worked well if the canopy capacity at zero LAI was reduced (in JULES, the canopy capacity is a linear function of LAI) and the PFT infiltration enhancement factor increased. Interestingly, Colello et al. (1998) concluded that they needed to change the infiltration and canopy interception capacity for this site. There was an issue capturing one of the peaks in the surface soil moisture in the spring, which was probably due to missing data in the rainfall dataset: the local day maximum in `FIFE_FFOAMS87_88` from day 130 to day 150 was 42.71mm, which occurred

on day 147, which had 9 missing timesteps. In contrast, the local day maximum from for this interval in Stewart and Verma (1992) was much higher, at around 70mm.

## A4 Soil properties

This section discusses and compares the available measurements of the hydraulic, thermal and optical soil properties, which can be used as ancillary data for runs at FIFE site 4439. Soil in the FIFE area was extensively studied. At site 4439, the soil was classified as predominantly Dwight silty clay loam (Typic Natrustolls) (Verma et al., 1992). Colello et al. (1998) describes the soil column as being "about 140cm in depth, changing from silty-clay-loam to clay to gravel to impermeable bedrock".

In our simulations, each soil ancillary variable was set to be constant throughout the soil column. The two most important soil parameters are the 'wilting' soil moisture $\theta_{wilt}$ and 'critical' soil moisture $\theta_{crit}$, which we define as the volumetric soil moisture at -0.033MPa and -1.5MPa respectively (following Cox et al. (1998) and Best et al. (2011)). These soil parameters enter directly in to the soil moisture stress calculation. In all of our simulations, $\theta_{wilt}$ was set to 0.205 and $\theta_{crit}$ was set to 0.387, taken from Cox et al. (1998) (which quotes Stewart and Verma (1992), although these values do not appear in this paper explicitly). In contrast, Verma et al. (1989) states that the surface (0 to 0.05m) wilting and critical soil moistures were approximately 15.0% and 39.4% respectively. It is also possible to obtain the wilting and critical soil moistures used in Verma et al. (1992), from comparing their extractable water values to volumetric soil moisture measurements from individual days in Cox et al. (1998). This leads to wilting and critical soil moistures of 20.1% and 34.8% respectively.

We used the Brooks and Corey (1964) relation between soil water content $\theta$ and absolute matric potential $\Psi$

$$\frac{\theta}{\theta_S} = \left( \frac{\Psi}{\Psi_S} \right)^{-1/b}, \tag{A1}$$

where $S$ denotes values at saturation, to obtain the Brooks-Corey parameter $b$ and the soil water suction at saturation $\Psi_S$ from the Cox et al. (1998) values of $\theta_{wilt}$ and $\theta_{crit}$. The other hydraulic and thermal soil ancillary variables were calculated from the fraction of sand, silt and clay given for Dwight soil in FIFE_SOILSURV_115, averaged over 0-122cm, using the relations from Cosby et al. (1984). The soil albedo (0.162) was calculated from the Munsell color value for dry Dwight soil given in FIFE_SOILSURV_115, averaged over 0-122cm, using the relation in Post et al. (2000). This was consistent with the reflectance data for Dwight soil in FIFE_SOILREFL_114 (which had mean 0.153, standard deviation 0.055 and was taken at a range of wavelengths).

There are also measurements available at specified depths. FIFE_SOILSURV_115 contains observations for clay, silt, sand and organic carbon content, bulk density, wilting and critical soil moistures for Dwight soil at different depths (this data is from site 2731, but it states that this data can also be used for site 4439, because the two sites have similar soil series). The relations in Cosby et al. (1984) can be used to convert the clay, sand, silt fractions to the soil hydraulic and thermal parameters needed by JULES. These can be corrected for organic content using Dankers et al. (2011) and Chadburn et al. (2015). The FIFE_SOIL_REL_112 dataset contains site 4439 bulk density and soil water potentials at different volumetric soil contents (including the wilting and critical soil moistures). FIFE_SOILDERV_117 has soil porosity, saturated water potential and the $b$ parameter from Eq. A1 for site 4439. Water retention curves plotted using this data are consistent with the data in FIFE_SOIL_REL_112 (not shown). Hydraulic conductivity for site 4439 is provided in FIFE_SOILHYDC_107. Bulk density can be converted to saturation volumetric soil moisture using the relation given in the FIFE_SOILDERV_117 documentation.

The resulting soil hydraulic and thermal parameters from these different methods are plotted in Figure A6, and shows that there are considerable differences between the different datasets. The large spread in the wilting and critical soil moistures is particularly important to note, since, as we have discussed, they both enter the soil moisture stress factor $\beta$ explicitly, and therefore plant GPP and transpiration are very sensitive to variations in these parameters. The thermal and optical soil properties and the remaining hydraulic properties have a comparatively minor effect on GPP and evapotranspiration.

## A5  Canopy height

In this study, we used the canopy height observations presented in Table 2 of Verma et al. (1992): 0.4-0.6m, 0.6-0.75m, 0.75-0.9m for days 120-179, 180-239, 240-300 respectively for site 16 in 1987. Another available dataset for canopy height at this site is FIFE_VEG_BIOP_135, which is plotted in Figure A7, and shows considerable differences with the Verma et al. (1992) data, particularly in the 240-300 day period. As discussed in Section A2, the non-uniformity of the vegetation at this site is a significant source of error in these measurements.

## A6  Canopy dark respiration

Polley et al. (1992) shows leaf dark respiration as a function of leaf temperature for observations of *A. gerardii*, *S. nutans* and *P. virgatum* taken in the FIFE area in 1987 and fits the following relationship:

$$R_{dl} = \frac{0.0496T_l - 0.0157}{1 - 0.01158T_l}. \tag{A2}$$

When this relation was used in Cox et al. (1998), it was scaled up to the canopy level by multiplying by LAI, i.e. dark respiration was assumed to be constant on leaves through the canopy. In contrast, in the model presented in Kim and Verma (1991a), leaf respiration was calculated from

$$R_d = R_{d,25} \exp\left[45000(T_l - 25)/(298R(T_l + 273))\right], \tag{A3}$$

where $R_{d,25}$=1.55 $\mu$mol m$^{-2}$ s$^{-1}$, $R$=8.314 J K$^{-1}$ mol$^{-1}$ is the gas constant and $T_l$ is the leaf temperature in °C and leaf dark respiration was suppressed by 50% when the absorbed PAR was greater than 20 $\mu$mol quanta m$^{-2}$ s$^{-1}$, to account for the light dependency of mitochondrial respiration. Air temperature near the top of the canopy was used to approximate leaf temperature. Kim and Verma (1991a) scaled this leaf respiration up to the canopy level by considering the sunlit and shaded portions of the leaf separately.

In JULES, dark respiration decreases through the canopy in the same way as $V_{cmax}$ and it is multiplied by the soil moisture stress parameter $\beta$. In the 'big leaf' approximation used in the `repro-cox-1998` run, $V_{cmax}$ decreases through the canopy with light. In the layered canopy model with sunflecks used in the `global-C4-grass` and `tune-leaf` runs, the decrease of $V_{cmax}$ through the canopy is set by an input parameter $k_{nl}$, and the leaf dark respiration is reduced by a factor of 30% above a light threshold.

## A7   Net canopy assimilation

In this study, we compared the net canopy carbon assimilation from the model (for Gross Primary Productivity (GPP) minus respiration from leaves) to two different datasets. The first dataset was read from Figures 1-4 in Kim and Verma (1991a), for 5th June, 2nd July, 30th July and 20th August 1987, which was obtained from eddy correlations of atmospheric $CO_2$, measured above the canopy. Leaf respiration was calculated from Eq. A3, as described in Section A6. The leaf respiration over the entire canopy was subtracted from the night-time $CO_2$ flux from the night following or proceeding the day under consideration, to calculate the other sources of respiration (soil, root), which were adjusted to daytime soil temperatures using a $Q_{10}$ factor of 2.

The second net canopy carbon assimilation dataset was created from FIFE_SF30_ECV_33 observations of $CO_2$ flux from eddy correlation techniques using the procedure in Cox et al. (1998). The total respiration $F_s$ in Cox et al. (1998) was fitted to the functional form proposed by Norman et al. (1992) for use when LAI measurements were not available, evaluated with FIFE data:

$$F_s = s_1 \left( \frac{\theta - s_2}{0.4 - s_2} \right) e^{s_3(T_{s,10} - 25)}, \tag{A4}$$

where $T_{s,10}$ is the 10cm soil temperature in $^\circ$C and $s_1$, $s_2$ and $s_3$ are fitted parameters. Using air temperature in the place of the soil temperature, Cox et al. (1998) found that using this expression with the parameter values $s_1$=17.8$\mu$ mol $CO_2$ m$^{-2}$ s$^{-1}$, $s_2$=0.2, $s_3$=0.062 $^\circ$C$^{-1}$ explained 50.7% of the variance in night-time $CO^2$ flux measurements at FIFE. Leaf-level dark respiration was calculated using Eq. A2, scaling from leaf-level to canopy level by multiplying by LAI, as described in Section A6, assuming that the leaf temperature and the air temperature were the same (we used the air temperatures in FIFE_SF30_ECV_33).

Canopy measurements taken in a Plexiglas chamber (FIFE_PHO_BOX_27) at 4 sites, including 4439, could possibly be used as an additional source of net canopy assimilation for comparison with the model. It would also be interesting to extend the analysis to include an evaluation of the modelled soil respiration. The model could be compared directly to the fitted expressions for soil respiration (with and without a LAI dependence) from Norman et al. (1992) or, alternatively, to the soil $CO_2$ flux measurements available in FIFE_SOIL_CO2_105.

*Competing interests.*   The authors declare that they have no conflict of interest.

*Acknowledgements.*   Firstly, we wish to express our respect and gratitude to the FIFE investigators. FIFE was a monumental scientific achievement and its legacy will be long-lasting. We would like to thank Tim Arkebauer for information about the FIFE leaf gas exchange observations from Polley et al. (1992). KW and PF acknowledge funding from the IMPREX research project supported by the European Commission under the Horizon 2020 Framework programme with grant no. 641811. KW, CM and PF were supported by the Met Office Hadley Centre Climate Programme funded by BEIS and Defra. CH grateful acknowledges the NERC/CEH National Capability Fund. L.M

was supported by the UK Natural Environment Research Council through The UK Earth System Modelling Project (UKESM, Grant No. NE/N017951/1).

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

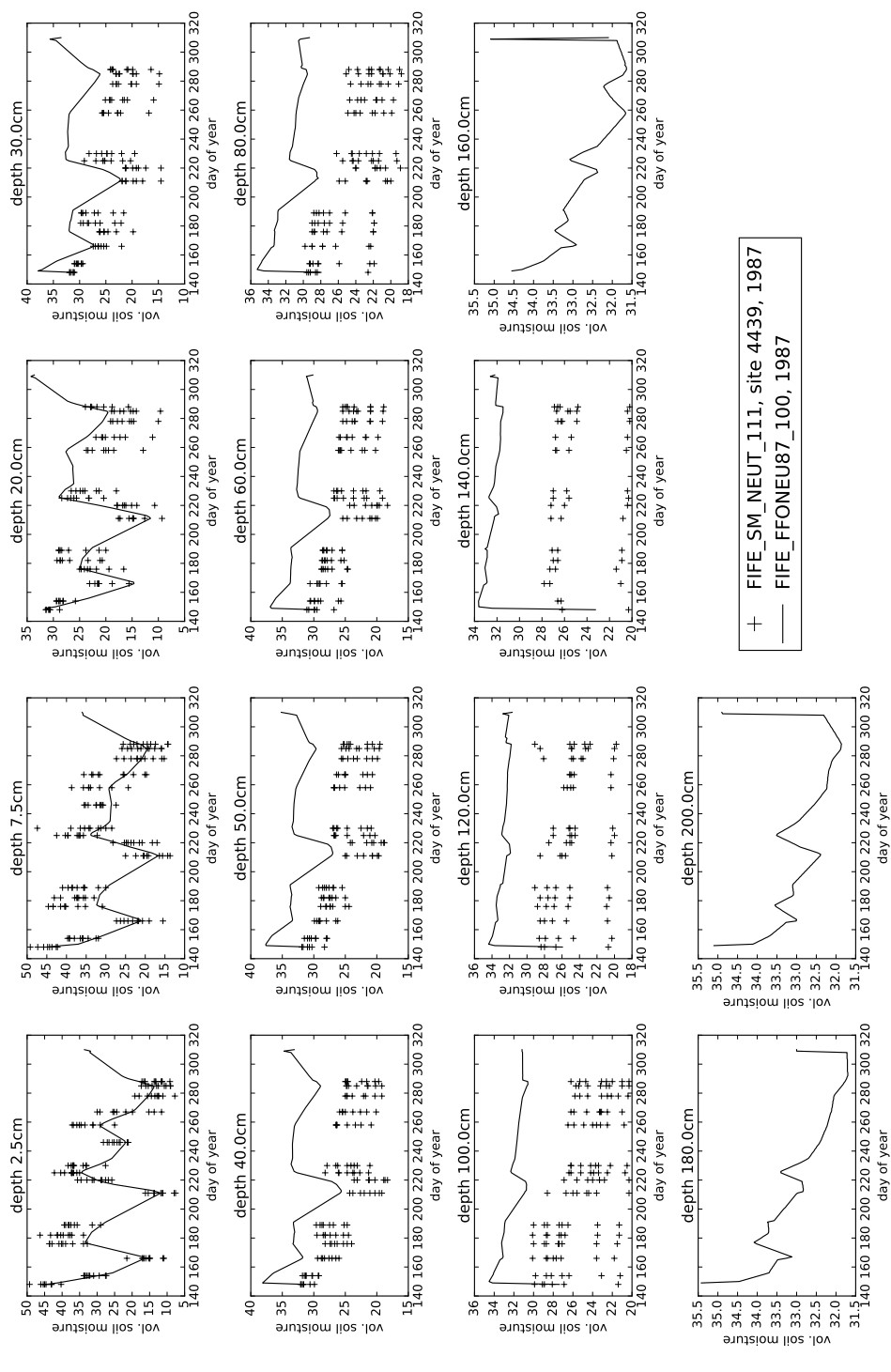

**Figure A4.** Site-averaged soil moisture on levels from FIFE_FFONEU87_100 for 1987 (line) and individual observations of site 4439 in 1987 from FIFE_SM_NEUT_111 (points).

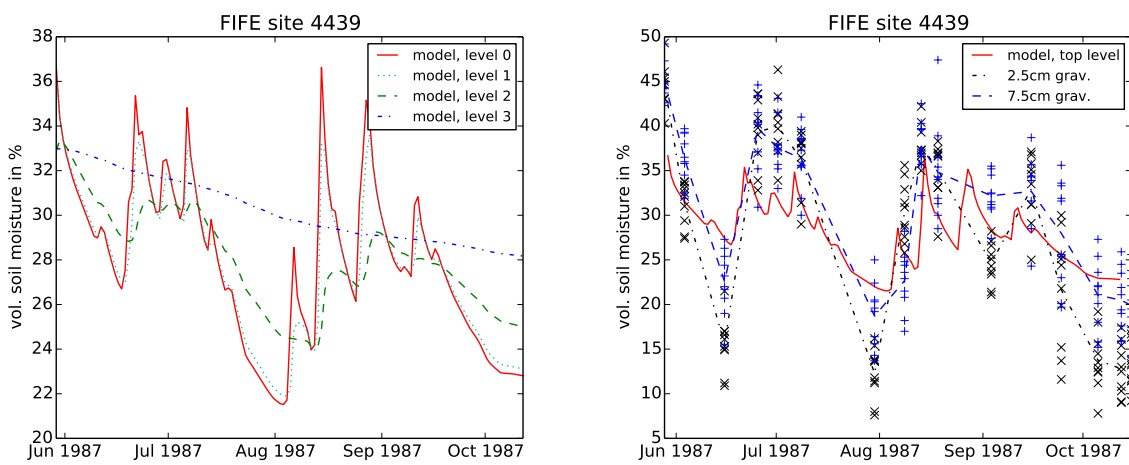

**Figure A5.** Left: Derived soil moisture dataset, on model soil levels. Right: Derived soil moisture in the top layer, compared to the gravimetric soil moisture measurements for 2.5cm and 7.5cm from FIFE_SM_NEUT_111.

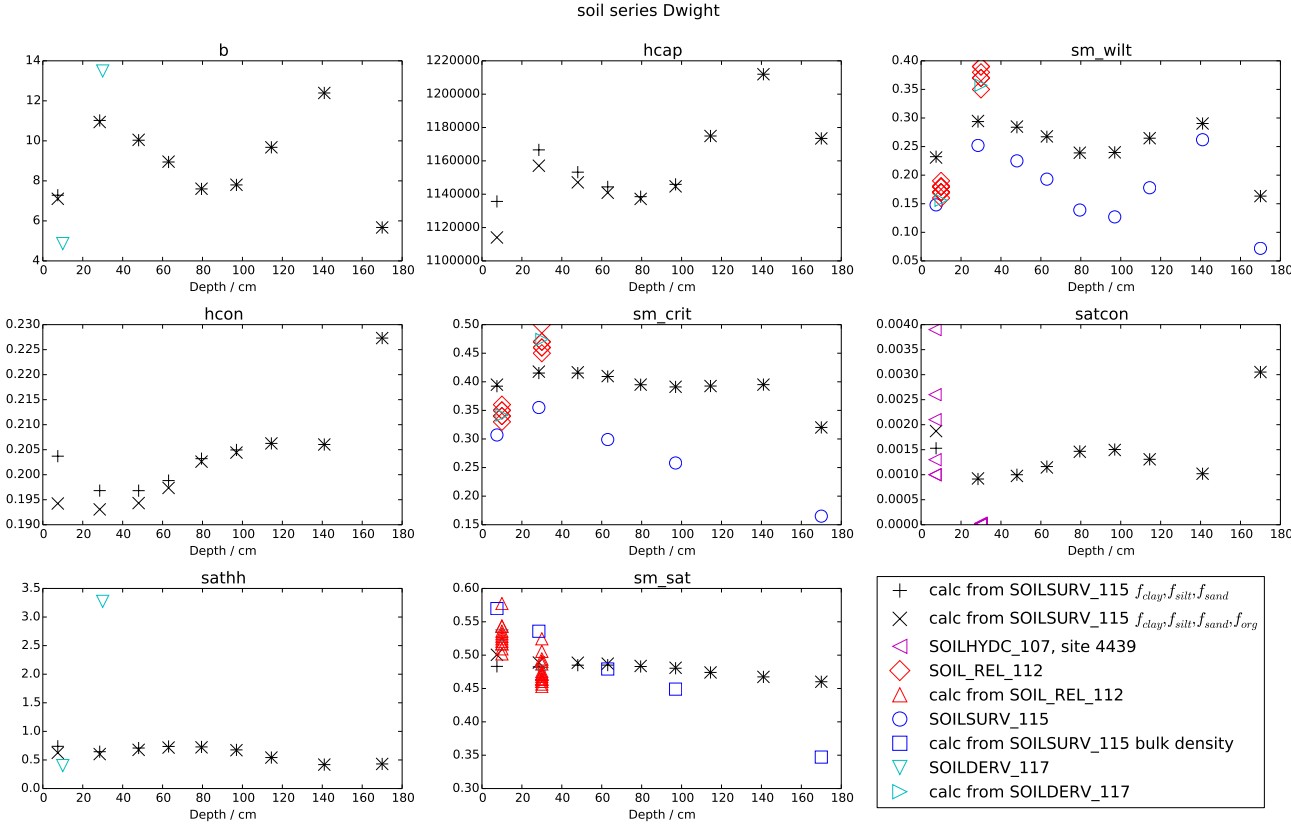

**Figure A6.** Soil ancillary variables needed by JULES, using the notation from the JULES namelists. When JULES is set to use soil hydraulic characteristics from Brooks and Corey (1964), these are b (exponent in soil hydraulic characteristics i.e. $b$ in Eq. A1), hcap (dry heat capacity in J m$^{-3}$ K$^{-1}$), sm_wilt (volumetric soil moisture content at -1.5MPa, $\theta_{wilt}$), hcon (dry thermal conductivity in W m$^{-1}$ K$^{-1}$), sm_crit (volumetric soil moisture content at -1/30MPa, $\theta_{crit}$), satcon (hydraulic conductivity at saturation in kg m$^{-2}$ s$^{-1}$), sathh (absolute value of the soil matric suction at saturation $\Psi_S$ in m) and sm_sat (volumetric soil moisture content at saturation $\theta_S$).

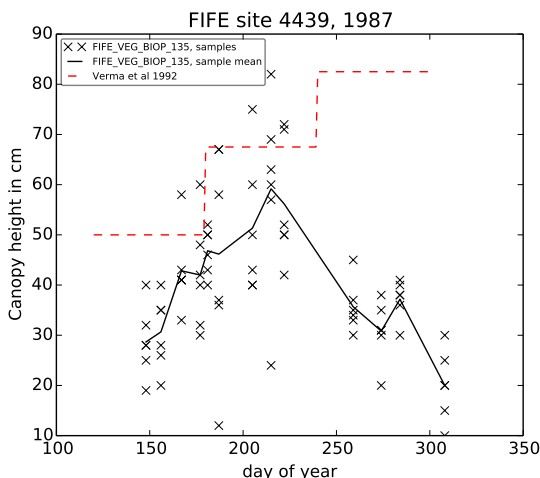

**Figure A7.** Canopy height in cm for site 4439 for 1987 from FIFE_VEG_BIOP_135.

| JULES notation | repro- cox-1998 | global- C4-grass | tune-leaf A.g. (default) | tune-leaf A.g./P.v. | Description |
|---|---|---|---|---|---|
| can_rad_mod | 1 | 6 | 6 | 6 | Flag to select canopy radiation scheme (-). |
| fd_io | 0.025 | 0.019 | 0.054 | 0.054 | Scale factor for dark respiration (-). |
| nmass_io | 0.015326 | 0.0113 | 0.025 | 0.02455 | Top leaf nitrogen content per unit mass (kg N (kg leaf)$^{-1}$). |
| vsl_io | 20.48 | 20.48 | 30.0 | 37.5 | Slope in the linear regression between $V_{cmax}$ and nitrogen per leaf area ($\mu$mol $CO_2$ g N$^{-1}$ s$^{-1}$). |
| lma_io | 0.137 | 0.137 | 0.0609 | 0.0574 | Leaf mass per unit area (kg leaf m$^{-2}$) |
| tlow_io | 13.0 | 13.0 | 23.0 | 25.5 | Lower temperature parameter in the $V_{cmax}$ calculation (°C). |
| tupp_io | 36.0 | 45.0 | 49.0 | 47.5 | Upper temperature parameter in the $V_{cmax}$ calculation (°C). |
| q10_leaf_io | 2.0 | 2.0 | 1.0 | 1.0 | $Q_{10}$ factor in $V_{cmax}$ calculation (-). |
| dq_crit_io | 0.078 | 0.075 | 0.070 | 0.070 | Critical humidity deficit $dq_{crit}$ (kg $H_2O$ per kg air) |
| f0_io | 0.82 | 0.8 | 0.53 | 0.53 | Ratio of internal to external $CO_2$ pressure when canopy level specific humidity deficit is zero $f_0$ (-). |
| fwe_c4 | 2.0E4 | 2.0E4 | 1.0E4 | 1.25E4 | |
| fsmc_mod_io | 1 | 0 | 0 | 0 | Integer indicating weighting of soil layers in water stress factor. |
| fsmc_p0_io | 0.0 | | 0.3 | 0.3 | Scaling factor $p_0$ in water stress factor calculation . |
| can_struct_a_io | 1.0 | 1.0 | 0.8 | 0.8 | Canopy clumping factor $a$. |
| rootd_ft_io | 1.4 | 0.5 | 0.5 | 0.5 | Parameter determining the root depth (m). |
| alpha_io | 0.034 | 0.04 | 0.048 | 0.053 | Quantum efficiency (mol $CO_2$ (mol PAR photons)$^{-1}$). |
| omega_io | 0.001 | 0.16 | 0.16 | 0.16 | Leaf scattering coefficient for PAR (-). |
| alpar_io | 0.0005 | 0.1 | 0.1 | 0.1 | Leaf reflection coefficient for PAR (-). |
| beta2 | 0.9 | 0.93 | 0.93 | 0.93 | Coupling coefficient for co-limitation in photosynthesis model (-). |
| co2_mmr | 0.0005 | 0.00053 | 0.00053 | 0.00053 | Concentration of atmospheric $CO_2$, expressed as a mass mixing ratio. |

**Table A2.** JULES parameters used to represent vegetation at FIFE site 4439, which vary across runs. These parameters are all specified in the JULES_PFTPARM namelist apart from can_rad_mod (JULES_VEGETATION), co2_mmr (JULES_CO2) and beta2 (JULES_SURFACE).

| JULES | tune-leaf | varying $dq_{crit}$ | | | fit to Medyn model | |
|---|---|---|---|---|---|---|
| notation | A.g. (default) | a | b | c | $g_1 = g_1^{fit}/2$ | $g_1 = g_1^{fit}/4$ |
| dq_crit_io | 0.070 | 0.048 | 0.040 | 0.035 | 0.057 | 0.051 |
| f0_io | 0.53 | 0.59 | 0.64 | 0.68 | 0.36 | 0.22 |

**Table A3.** Parameter combinations used for the $f_0$, $dq_c rit$ sensitivity studies.