# Peer review of "How can the First ISLSCP Field Experiment contribute to present-day efforts to evaluate water stress in JULESv5.0?"

_Geoscientific Model Development, 2018_

## Referee Comment (RC1) · Anonymous Referee #1 · 29 Oct 2018

Williams et al. explored parameterising three different configurations of JULES to capture the diurnal cycle of GPP, net canopy assimilation, and latent heat flux during the dry spell at FIFE site 4439, Kansas, US, in 1987. They chose this site because it is of historical importance to the JULES community, having been used to develop the parameterisation of water stress in the model, but also because of the wealth of data collected there during 1987-1989. Out of the three configurations they tested, the authors found that the repro-cox-1998 was most successful at capturing the site fluxes, i.e. the closest approximation to the original Cox et al. 1998 model. Despite incorporating physical processes which are not supported by observations at the site (e.g. Vcmax declining at leaf temperatures above 32°C), this configuration is heavily tuned to the site data and mimics the model historically used to derive the JULES' parameterisation

of water stress (i.e. Cox et al. 1998). The two less successful configurations both embed more sophisticated and mechanistic representations of the canopy, soil, and radiation modules; however, they are not run with the same PFT-specific parameters that were used in repro-cox-1998.

Overall, the authors' results seem to highlight the need for: (i) more coherent / less error-ridden site forcing data, (ii) more thorough evaluation at different stages of model development with regard to the assumptions in calibrating vegetation parameters. As presented, it is unclear what novel advances to the literature these broad conclusions brings. Nevertheless, there are interesting elements within the study, such as the author's effort to test three different configurations of a single model, representing different levels of complexity, with a variety of data for a specific PFT. For the value of those elements to clearly appear to the reader and for this manuscript to be ready for publication, I believe major revisions are necessary.

It is especially important in revision that the authors reorganise their manuscript to more clearly demonstrate their findings. It is likely that separating the result and discussion sections will help the authors to more clearly present the paper's findings. In particular, thinking beyond the JULES community may help them articulate their findings - why would a developer of another LSM care about what is in this manuscript? Could more process-level interpretation arise from the simulations? But also, what are the advances for the JULES model community? If this is meant as a benchmarking type of effort, where is the performance evaluation? The latest more sophisticated configurations appear to perform "worse" than repro-cox-1998, so should JULES swap back to repro-cox-1998 for C4 grasses?

———————————————— Main comments ————————————————

This paper focuses on how well a model can simulate a C4 tallgrass prairie's response to water stress. So, generally, what are the valuable lessons? Why does the model fail to capture the dry-down response (what mechanism)? What have the authors tested to capture the missing mechanism? Even if simply empirically? Why not also run the global-C4-grass and the tune-leaf simulations with the JULES parameters used in repro-cox-1998 to highlight where the process based differences play a role? Unless I have missed this analysis in the paper, I think the respective parameterisations are different enough to hinder a mechanistic understanding of why differences occur. Where they simply assert: "inherent uncertainties in key observables, such as leaf area index, soil moisture and soil properties", could the authors attempt to constrain one of these, e.g. LAI? Otherwise, I fail to see the point if we simply end up concluding these data are too uncertain to evaluate against. Or, given one of the paper's aims to "demonstrate how the wealth of data collected at FIFE and its subsequent in-depth analysis in the literature continues to be a valuable resource for the current generation of land-surface models", what are the immediate next steps the authors intend to make to exploit these data to improve the JULES model, without data related uncertainty hampering model development?

Far more evidence is required to substantiate some of the points/arguments made to explain why the model is failing. As currently presented, they are purely speculative. For example, the authors speculate that an empirical link between leaf water potential and Vcmax/Jmax could improve models simulations. It would be an advance to the literature to actually show such a model (given they are relatively trivial to implement, e.g. Zhou et al. 2013, AFM; Kim and Verma 1991a, AFM), or at the very least, link more explicitly to literature that has done this (e.g. Tuzet et al. 2003, PCE; Zhou et al. 2013, AFM). Further, it would be useful to discuss the mechanism behind a direct limitation

of leaf water potential on Vcmax and/or Jmax. It is unclear why the authors feel like the influence of VPD is negligible in the existing approach in JULES (lines 6-8, p.11), given that increasing VPD would drive a reduction in Ci? Indeed, in the last paragraph of Section 3.2, they show that for tune-leaf VPD influences GPP via this mechanism on both the 30th July and the 11th August. They also show that this mechanism alone doesn't have the flexibility to reproduce the observations. For all the days presented in Figure 13, it would be interesting to also plot Ac / E (the transpiration can simply be derived from the latent heat) relative to the declining soil moisture (or/and time), to see where the relative constraint is greater (on A more than E or vice-versa?) which might help further understand why the model is failing for the more extreme dqcrit parameterisation.

It is clear that FIFE site 4439 has historical value for the JULES community and that a lot of data is available. I am uncertain, though, as to how representative of the C4 grass PFT or the tallgrass prairie vegetation in general it is? Could the authors elaborate on this point, perhaps in the discussion? Why is this site a good proxy to calibrate the model for this PFT, in particular considering the variability of the site data? And is there any indication that it behaves like any other C4 grass site would during a dry spell? If this cannot be shown, I would encourage the authors to reword statements like "... FIFE observations were used to derive the original soil moisture stress parameterisation that was incorporated into JULES. This therefore makes FIFE an ideal test case for evaluating and improving this process.".

The authors could also more clearly demonstrate the impact of key assumptions. The following is an important point concerning the physical representation of the response of Vcmax to leaf temperatures above 32°C in Cox et al. (1998): "However, as discussed in Section 2, this temperature response is not supported by observations in Knapp (1985) or Polley et al. (1992). Therefore, it appears that, while the model is successfully capturing the shape of the diurnal cycle during the dry period, it is not achieving this

with the correct physical process." It would be easy for the authors to test this, simply by swapping the temperature response function and determining if this statement is true or false.

Finally, I would like to thank the authors for making all of the code and data available. Their careful description of the steps taken to set up the simulations is also appreciated.

——————————— Minor comments ———————————

Lines 28-30, p.2: the precision that "changing $p0$ can be considered a proxy for changing the critical soil moisture" which appears lines 8-9, p.16, could probably appear here, or at least in Figure 1's caption

Lines 1-2, p.5: even though the differences in evaporative schemes aren't the focus here, can the authors estimate how those might influence their conclusions?

Section 2.3: how is the tune-leaf configuration calibrated exactly? What is matched for in the calibration process? What does "approximately representative of the dominant species" mean? How so?

Lines 9-10, p.7: it is unclear to me why the burned plot was not water-limited. Could the authors please elaborate?

Lines 31, p.7: why do the authors assume that the best parameter set is a composite of two species' parameter sets, given the non-linear response of photosynthesis to plant traits?

Line 10, p.8: I don't understand which mean the Al-Ci curves were normalised against

Lines 11-13, p.8: can the authors demonstrate this claim or refer to studies that do?

Lines 30-34, p.8: does this mean the dark respiration at Tleaf different to 30°C is then still scaled according to the temperature dependency in JULES (if so, can the authors justify this approach)? Or does the scaling follow Polley et al. (1992)?

Lines 4-14, p.9: that whole paragraph could be moved to the beginning of section 2.3.1, thus the text that follows might be less confusing for the reader

Line 22, p.9: missing the work "water" after "vegetation. Leaf"

Lines 22-23, p.9: maybe specify what other factors can affect leaf water potential

Lines 1-2, p.15: is this observed during the dry period as well? Does this mean that it is constantly proportional through time?

Lines 6-8, p.16: please add the missing words in the sentence

Line 3, p.17: missing "of" before "the humidity response"

Lines 4-6, p.18: leaf rolling/folding implementation feasibility in a global model should at least be discussed in view of the existing literature and considering the author's

previous statement that "this behaviour cannot be modelled in the current version of JULES" (line 28, p.11); the same goes for including leaf nitrogen

Line 9, p.25: the authors should specify "for C4 grasses" or something equivalent

Line 14, p.26: I believe this is the first time senescence is mentioned. Do the authors propose to do this via the leaf water potential parameterisation? Or do they envision it might somehow relate to a LAI phenology?

Line 20, p.32: the approximation that soil evaporation can be neglected for days without rainfall seems rather big to me; have the authors considered including soil evaporation (though it isn't the focus of the study) to reduce the uncertainty?

Figure A4: the depth unit should appear somewhere in the plot

————————————— Suggested references —————————————

Tuzet, A., Perrier, A., Leuning, R. (2003). A coupled model of stomatal conductance, photosynthesis and transpiration. Plant, Cell Environment, 26(7), 1097-1116.

Zhou, S., Duursma, R. A., Medlyn, B. E., Kelly, J. W., Prentice, I. C. (2013). How should we model plant responses to drought? An analysis of stomatal and non-stomatal responses to water stress. Agricultural and Forest Meteorology, 182, 204-214.

———————————————————————

---

## Referee Comment (RC2) · Anonymous Referee #2 · 29 Oct 2018

This manuscript describes an evaluation of the JULES land model using field data from the FIFE dataset. Three simulations are presented: a replication of a simulation from an earlier model version, a standard simulation using the current version, and a tuned simulation using site specific parameters rather than global parameters.

Numerous model intercomparison projects (MIPs) have been published in recent years, and one of the criticisms of these studies is the lack of adequate control in the experimental design. For example, models participating in a study may differ in forcing data, structure, and parameters, which makes attribution of differences in the results to these model characteristics difficult if not impossible. This study can be thought of as a three model MIP, and it features the same difficulty. Although the three models are all versions of JULES, they differ in multiple ways, making the interpretation of the results

unclear.

For example, simulations 2 and 3 differ due to the tuning of simulation 3 of multiple parameters for stomatal conductance and C4 photosynthesis, such as: SLA, leaf nitrogen, light response parameters, Vcmax temperature dependence, A-ci response at low ci, ci-ca relationship, dark leaf respiration - Vcmax ratio, p0 (water stress onset parameter), and canopy structure (uniformity). Figures 8-10 show that the site specific tuning exercise leads to large differences between simulations 2 and 3; in fact, these differences are larger under *unstressed* conditions than under stressed conditions. Moreover, while the tuned parameters should by design lead to better agreement with the observations against which they were calibrated (figures 3-7), they do not in general improve the simulation of GPP or An under unstressed conditions. I recommend to the authors to give the reader a better understanding of the individual impacts of this tuning exercise in the context of their effect on GPP and An. Which parameter changes improved the comparison, and which degraded the comparison? This is important to understand, more so because the agreement differs for the different days presented in the analysis. While it is subjective, I did not agree with the authors' statement that the model was "...able to successfully reproduce the net canopy assimilation and latent heat energy flux reasonably well through the season". Given this perhaps unsatisfactory starting point for unstressed conditions, I recommend to the authors to focus first on obtaining more credible results under unstressed conditions before addressing the model's response to drought.

The title indicates that the goal of the manuscript is the evaluation of water stress in the current version of JULES, thus my expectation was that the experimental design would isolate the behavior of this parameterisation; however, this is not really the case. Only figures 11-13 show single factor analyses, and of those, only figure 11 directly examines the water stress parameterisation used in the model.

Figure 11 shows the results of varying the p0 parameter that determines the initial soil moisture value at which vegetation experiences stress. I recommend that the authors

show a few actual lines rather than the spread of p0 = [0,0.4] to enable the reader to easily see whether increasing p0 increases or decreases GPP. One might expect that water stress based on soil moisture would not exhibit large diurnal variation, and this is confirmed by these plots. Figures 8-10 show that the diurnal variation in GPP and An can be simulated (in this case by simulation 1) if a predictor having stronger diurnal variation (such as temperature) is used. However, the authors note that this type of parameterisation is not well supported ("The repro-cox-1998 simulation is more successful, but this response is mediated by a temperature dependence in leaf carbon assimilation which is not supported by observations") as shown by figure 4.

At this point the analysis is basically complete, with no improvement in the diurnal cycle of carbon flux. This seems not to support the authors' conclusions regarding outcome of the study (e.g. "FIFE provides an ideal case study for improving the model representation of water stress on carbon and water fluxes on a tallgrass prairie site") as no significant improvement was made aside from tuning the p0 parameter based on unstressed soil moisture conditions. The authors note that leaf water potential was used by authors of previous studies to simulate the diurnal cycle of GPP under dry conditions, leading them to conclude "JULES is not currently able to capture the diurnal cycle of net canopy photosynthesis at this C4 grass site, due to the lack of a strong dependence on the canopy vapour pressure deficit (indirectly or directly)", but this is largely conjecture and not actually tested by the authors of this paper.

In summary, this manuscripts describes many of the issues that one encounters when attempting to constrain a model to field observations, such as uncertainty in measurements and spatial heterogenaeity. It highlights the fact that greater model complexity does not guarantee greater model fidelity. It also shows that site-specific model parameters may give significantly different results relative to global parameters. This is valuable information and worth presenting. However, that is not the stated focus of the manuscript, which is water stress and its improvement in JULES. To that end, I recommend to the authors to revise their title and to shift the focus of their discussion towards

the actual content of the paper.

---

## Referee Comment (RC3) · Anonymous Referee #3 · 7 Nov 2018

The authors present three simulations of FIFE Site 4439 (Konza Praire, Kansas) utilizing three configurations of JULES, from Cox et al. 1998, Harper et al. 2016, and a third developed for this manuscript.

The parameterizations are compared for sensitivity to changes in root-zone soil water, light availability, leaf temperature as well as for the relationship between intercellular $CO_2$ and net assimilation. The results show that the two literature-based configurations may not capture observed relationships in site-level data (e.g. Polley et al. 1992, Knapp 1985).

The simulations are next confronted with observation-based data of GPP, net assimilation, and latent heat flux from 8 days during 1987. Generally speaking, the Cox et al. simulation output tends to provide the best fit to these data. Lastly the authors present

parameter sensitivity results (of GPP) for four "tune-leaf" parameters.

Strengths: (1) The authors very clearly explain the process by which they developed the tune-leaf parameterization and its differences with the Harper and Cox parameterizations. This includes Figures 3-5,7, which effectively demonstrate the fit provided to leaf-level data by the three parameterizations. (2) The authors show, across a small subset of observational data, that utilizing field-derived paramater values tend to degrade model performance (Figs8-10). This is an important (and not unexpected) finding that will merit further study.

Weaknesses: (1) Conclusions are not strongly based in the specific work conducted within this study. See comments. (2) The scope of the study does not match the ambitions of the introduction. Page 2, Line 31 states: "This effort requires a large amount of data to evaluate against, covering a wide variety of climate and vegetation conditions". Yet within this study only one site is analyzed. The authors might consider better contextualizing the current status of progress towards such a dataset. (3) The authors do not acknowledge the mismatch between the leaf level scale and the eddy covariance / ESM gridcells. A short discussion of the challenges of reconciling leaf level parameters with the effective large scale parameters would be merited. (4) Observational data record is extremely short (8 days). Are there other data available (remote sensing or flux tower) that could be used to evaluate the different model configurations? Why is only one year of the FIFE data used? (5) Many statements based on literature review are presented without citation in the results and conclusions sections. The relevant citations should be repeated, or the reader should be referred to the specific section where the citations can be found.

Specific Comments:

P15: Consider moving figure 6 to supplementary

Figures 8-10: would be interesing to see summary statistics on the goodness of fit for the various parameterizations.

[Figure]

P2.L31: There is a significant disconnect between the first five sentences of this paragraph and the single-site nature of this study.

P17.L24: "Other studies have argued that the dry period diurnal cycle at this site can by captured via an explicit dependence on leaf water potential". I understand this is covered in Section 2.3.2, but I think you should either repeat the citations or refer back to Section 2.3.2.

P26.L2-3: "JULES is not currently able to capture the diurnal cycle of net canopy photosynthesis at this C4 grass site", unclear which model configuration you are referring to.

P26.L2-3: "JULES is not currently able to capture the diurnal cycle of net canopy photosynthesis at this C4 grass site, due to the lack of a strong dependence on the canopy vapour pressure deficit (indirectly or directly)." Second clause needs citation. Not shown within this study.

P26.L4-5: "The temperature response of Vcmax can be tuned to compensate for this, but it is more desirable for the model to respond to high temperature stress and high water stress individually." This is not very well contextualized, perhaps instead you could more specifically say that repro-cox still manages a reasonable diurnal cycle, but likely due to compensating errors, and then refer back to the section where you discuss that in greater detail.

P26.L5-6: "These runs also showed how the default water stress parameterisation can result in large reductions in photosynthesis during periods that are not considered water-limiting at the site." I am not sure which figure you are referring to or which time period specifically.

P26.L10-11: "These have been extensively studied at FIFE in independent investigations and yet still show a wide spread, leading to large modelling uncertainties" needs citation.

P26.L12: "The FIFE data also indicates ..." How exactly does the FIFE data indicate that JULES should represent leaf rolling and senescence?

---

## Author Comment (AC1) · 25 Feb 2019

**Response to reviewers: gmd-2018-210**

**1 General notes to all reviewers**

We thank all the reviewers for their thorough reading of the manuscript and insightful and helpful comments.

One common theme in all three responses is that we have not described the aim of the paper with sufficient clarity. We address this serious deficiency in the version of the manuscript submitted with this response, and thank the reviewers for drawing our attention to it.

Our overarching aim is to illustrate why it is useful to include FIFE in any collection of evaluation work on soil moisture stress in JULES. The FIFE dataset provides a clear example of where the current model parameterisation is unable to capture the diurnal cycle of GPP and transpiration during dry periods in this vegetation type, which is useful for testing model extensions. However, we did not mean to imply that any model improvements could be based on this dataset alone.

The availability of large, standardised data collections such as FLUXNET have revolutionised the evaluation of land-surface models against site data. Models and proposed extensions can be tested for a wide variety of climate regimes and vegetation types, and this is vital for making sure that they perform well on the global scale. The more sites that are considered in any particular study, the less detail can be covered on any one particular site in that study. Some single site experiments, such as FIFE, have an important contribution to make, although with the need for a proper understanding of the limitations of the dataset (for example, any conclusions drawn from this dataset should be robust to the large LAI uncertainty). Therefore, more detailed analyses of data from one site, such as ours, can help inform the more comprehensive multi-site analyses which are vital for model development, and GMD is the ideal place for both kinds of studies.

There are two studies underway so far which use the contents of this preprint, which provide good illustrations of how this paper can benefit future work. One of these studies is a comprehensive evaluation of JULES across many biomes involving 40+ authors. The other study evaluates an extension of JULES which incorporates optimisation arguments into the treatment of soil water stress in JULES, considering a wide variety of sites. In both cases, it was only practical to include FIFE because they could use the results from our study - given how many datasets they both include, they could not have devoted the time (or space in their manuscript) for going into this much detail on one site.

**Suggested change to manuscript:**

We have substantially redrafted the manuscript to make the aim of this study more clear, particularly the conclusions section. We have replaced the previous title *Revisiting the First ISLSCP Field Experiment to evaluate water stress in JULESv5.0* with

*How can the First ISLSCP Field Experiment contribute to present-day efforts to evaluate water stress in JULESv5.0?* , which makes the point that this paper is not aiming to be a comprehensive model evaluation on its own. We have also added a new section: ***Can the FIFE dataset make a useful contribution to current-day JULES evaluation and development work?*** , where we explicitly the describe the contribution the FIFE can make, while stressing that it should be used in conjunction with a large number of other datasets.

**Summary of the new plots added, to address the reviewers' comments:**

- Leaf water potential observations on four days, from Kim and Verma 1991.

- Calibrating the parameters in the ci calculation (f0, dqcrit) with a new observational dataset, from Lin et al 2015.

- Sensitivity of GPP to systematic variations in f0, dqcrit.

- Effect on GPP of imitating a 50% and 75% reduction in the g1 parameter in the Medlyn et al 2011 model (also shown is a plot showing how well the parameterisation used in JULES is able to fit the Medlyn parameterisation).

- How GPP, net canopy assimilation and latent heat flux change when replacing the Vcmax temperature distribution in the repro-cox-1998 configuration (we also show a plot of the Vcmax temperature distributions being tried out)

- Diurnal cycles of air temperature, leaf temperature and vapour pressure deficit

- Diurnal cycles of model transpiration and model evaporation (from canopy and surface/soil moisture stores).

**2  Response to Reviewer 1**

Williams et al. explored parameterising three different configurations of JULES to capture the diurnal cycle of GPP, net canopy

assimilation, and latent heat flux during the dry spell at FIFE site 4439, Kansas, US, in 1987. They chose this site because it is of historical importance to the JULES community, having been used to develop the parameterisation of water stress in the model, but also because of the wealth of data collected there during 1987-1989. Out of the three configurations they tested, the authors found that the repro-cox-1998 was most successful at capturing the site fluxes, i.e. the closest approximation to the original Cox et al. 1998 model. Despite incorporating physical processes which are not supported by observations at the site (e.g. Vcmax declining at leaf temperatures above 32°C), this configuration is heavily tuned to the site data and mimics the model historically used to derive the JULES' parameterisation of water stress (i.e. Cox et al. 1998). The two less successful configurations both embed more sophisticated and mechanistic representations of the canopy, soil, and radiation modules; however, they are not run with the same PFT-specific parameters that were used in repro-cox-1998.

Overall, the authors' results seem to highlight the need for: (i) more coherent / less error-ridden site forcing data, (ii) more thorough evaluation at different stages of model development with regard to the assumptions in calibrating vegetation parameters. As presented, it is unclear what novel advances to the literature these broad conclusions brings. Nevertheless, there are interesting elements within the study, such as the author's effort to test three different configurations of a single model, representing different levels of complexity, with a variety of data for a specific PFT. For the value of those elements to clearly appear to the reader and for this manuscript to be ready for publication, I believe major revisions are necessary. It is especially important in revision that the authors reorganise their manuscript to more clearly demonstrate their findings. It is likely that separating the result and discussion sections will help the authors to more clearly present the paper's findings. In particular, thinking beyond the JULES community may help them articulate their findings - why would a developer of another LSM care about what is in this manuscript? Could more process-level interpretation arise from the simulations? But also, what are the advances for the JULES model community? If this is meant as a benchmarking type of effort, where is the performance evaluation? The latest more sophisticated configurations appear to perform "worse" than repro-cox-1998, so should JULES swap back to repro-cox-1998 for C4 grasses?

We thank the reviewer for this feedback, and have revised and reorganised the manuscript to address these points. In particular, we have redrafted the section in the introduction that outlines the aim for the study. We have also separated the results and discussions section into more sections, and rewritten the conclusions in order to answer these requests.

The reason we have presented three different configurations is not because we are looking at the effect of different levels of complexity or because we are looking at which is 'better' or 'worse'. Rather, we are using these configurations to explore whether this dataset would be a useful addition to the collection of datasets being used to evaluate the soil moisture stress representation within JULES and whether it needs improving. We realise that this was not made clear in the original version of the manuscript and we have rewritten the relevant sections (particularly the second half of the introduction and the conclusions).

It is unsurprising that the repro-cox-1998 configuration fits the net assimilation and evapotranspiration measurements at FIFE best, since it was tuned to these particular measurements. In contrast, the global-C4-grass configuration was designed (in Harper et al) to give good performance for C4 grass across the world. For this reason, it is generally inadvisable to adjust a PFT parameterisation that is believed to be broadly valid at the global scale, by instead one that is tightly tuned to a very specific

location. We have described this point more explicitly in the new section *Can the FIFE dataset make a useful contribution to current-day JULES evaluation and development work?* , where we stress that model parameterisations need to work globally, which means that they need to be calibrated against a wide variety of datasets. We give an example of good practise (Lin et al 2015) to further emphasise this point.

5    The importance of the repro-cox-1998 configuration in this study is the fact that it appears to correctly simulate a temporary drop in productivity during the middle of the day in the dry spell. Showing that this is for the wrong reasons (an unrealistic temperature dependence in Vcmax) is an important step in showing that the current version of JULES cannot correctly capture the diurnal water stress processes at this site. We have redrafted this discussion, and made sure that the reason for reproducing the Cox et al 1998 configuration is stated clearly and unambiguously in both the introduction and the conclusion sections.

10    We think that JULES community can benefit from careful use of this dataset, with full knowledge of its strengths and weaknesses (i.e. a more pragmatic approach then seeking "more coherent / less error-ridden site forcing data"). Our opinion is that the FIFE dataset is a good example of where complementary measurements have been taken (e.g. LAI), which has enabled a more thorough investigation of the uncertainty in these measurements (e.g. Kim et al 1989) than is available for many other datasets. We have edited the manuscript so that we discuss this in the new *Can the FIFE dataset make a useful contribution*
15    *to current-day JULES evaluation and development work?* section and the conclusions section, and we have also added a comment about this to the model setup section.

While the main focus of this paper is to determine how this particular dataset can contribute to evaluating and improving a specific process in JULES, we think that other land surface models might also consider using this dataset their own evaluation work. All the arguments in the first half of the new *Can the FIFE dataset make a useful contribution to current-day JULES*
20    *evaluation and development work?* section apply to all land-surface models. However, the model suite we provide to users to pre-process and run with the FIFE dataset is specific to the JULES.

**2.1    Response to main comments from Reviewer 1**

This paper focuses on how well a model can simulate a C4 tallgrass prairie's response to water stress. So, generally, what are the valuable lessons? Why does the model fail to capture the dry-down response (what mechanism)? What have the authors tested to capture the missing mechanism? Even if simply empirically? Why not also run the global-C4-grass and the tune-leaf simulations with the JULES parameters used in repro-cox-1998 to highlight where the process based differences play a role? Unless I have missed this analysis in the paper, I think the respective parameterisations are different enough to hinder a mechanistic understanding of why differences occur.

This point is also mentioned by reviewer 2, and has led us to rethink the collection of runs that we have presented this analysis. This study involved hundreds of runs, with different parameter combinations to really probe the relative contribution
25    of different mechanisms. We therefore had to think of how to pick a limited set for the manuscript to illustrate the main points. We picked three main configurations: (1) repro-cox-1998 to investigate the results published in Cox et al 1998, (2) global-c4-grass to show what the current 'best' parameterisation of this vegetation type in global JULES runs would capture, and (3) tune-leaf, to check that model deficiencies we are seeing in the global-c4-grass simulation can not be remedied by making the

vegetation parameters more appropriate for the vegetation at this site using site measurements of leaf properties etc. We then adjusted the tune-leaf configuration to show (a) the effect of the choice of vegetation to tune to (by changing from parameters tuned to A.g. obs to parameters that lay between the A.g. and P.v. obs), (b) varying the soil moisture at which vegetation started to become stressed, governed by $p_0$ (which could also be seen as a proxy for uncertainty in one of the soil properties, $\theta_{crit}$) and

5   (c) the canopy clumping factor $a$ (which could also be seen as a proxy for uncertainty in the LAI). We also showed a version of tune-leaf where the sensitivity of stomatal conductance to VPD had been over-exaggerated, well beyond realistic leaf-level values. This aimed to demonstrate that even in this extreme (and unrealistic) case, the VPD dependence of canopy GPP was not sufficient to model fluxes in the middle of the day during the dry season. This set of 9 runs was designed to cover the main points we wished to illustrate in the manuscript.

10     However, we also state in the manuscript that the reason that the repro-cox-1998 has a more realistic diurnal cycle during the dry spell than the other configurations is because of its unrealistic Vcmax temperature dependence. We did test this explicitly in runs with varying Vcmax temperature dependences, but did not include the plot in the previous version of the manuscript. We have now added these extra runs to the manuscript, which are based on repro-cox-1998 but show different vcmax temperature relations. We hope that showing this explicitly strengthens the paper, as the claim that the repro-cox-1998 configuration gets

15   the dry spell diurnal cycle right for the the wrong reasons is an important step in verifying that the current representation of water stress in JULES cannot correctly simulate the dry spell diurnal cycle.

**Suggested change to manuscript:**

- We have redrafted the sections discussing the motivation for each of the parameter combinations we illustrate.

20   - We have added new plots showing runs which demonstrate the impact of three Vcmax(T) functions, while keeping the other parameters the same as repro-cox-1998. This is to provide evidence for our claim that the temporary drop in the fluxes in the middle of the day in the dry spell in the repro-cox-1998 goes away when Vcmax(T) is replaced with versions that are more high temperature tolerant.

- We have also added 5 new runs exploring more combinations of f0, and dqcrit, which are the parameters that go into the

25     JULES ci/ca calculation.

Where they simply assert: 'inherent uncertainties in key observables, such as leaf area index, soil moisture and soil properties', could the authors attempt to constrain one of these, e.g. LAI? Otherwise, I fail to see the point if we simply end up concluding these data are too uncertain to evaluate against.

We agree that the uncertainty in LAI has an important effect on the confidence we have in the model water and carbon fluxes. As described above, varying the canopy clumping parameter $a$ is a proxy for the effect of varying LAI on model GPP, as it has a similar effect on the canopy radiation scheme. This is currently mentioned when the canopy clumping factor is first

introduced and when the results from the runs varying the clumping factor are described. Similarly, varying the parameter p0 can be considered as a proxy for varying the critical soil moisture.

**Suggested change to manuscript:**

5    We have clarified the paragraph describing the runs which vary the clumping factor, including stating explicitly that these results mean that 'The error in LAI for this site therefore has a large impact on the modelled canopy carbon fluxes.'

Or, given one of the paper's aims to 'demonstrate how the wealth of data collected at FIFE and its subsequent in-depth analysis in the literature continues to be a valuable resource for the current generation of land-surface models', what are the immediate next steps the authors intend to make to exploit these data to improve the JULES model, without data related uncertainty hampering model development?

In this paper, we aimed to show that the model cannot currently capture processes governing the dry spell diurnal cycle in GPP and ET at this site, and that this conclusion is robust to the sources of uncertainty in both the data used to force the model (e.g. LAI) and the model parameters (e.g. Vcmax temperature dependence). The next step is to make this analysis easy

10   to repeat , as we have done by providing the rose suite that accompanies this paper (rose is the scheduling software package typically used with JULES). As described above, this is currently being used in a comprehensive evaluation of soil moisture stress in JULES, which is based on a wider variety of site datasets. It is also being used (again in conjunction with lots of other site datasets) to evaluate a model extension, which includes a more sophisticated VPD dependence.

**Suggested change to manuscript:**

We have included a description how this work will be built on in practise to the new ***Can the FIFE dataset make a useful contribution to current-day JULES evaluation and development work?*** section:

  ...although the data is easily downloadable, well documented and in common file formats, is still needs to be manipulated into
  a format that can be used in JULES runs. We aim to address this issue by providing a suite that can be used to pre-process the
20   FIFE data and run JULES with the configurations described in this manuscript (see the 'code and data availability' section).
  This aim is central to the provision of this manuscript. FIFE is the first 'JULES golden site', a concept was launched at the
  annual JULES meeting 2018. A JULES golden site is a site targeted by the JULES community because it can help address
  one of the key science questions facing JULES and has high-quality observational data that can be used to drive JULES and
  evaluate the output. It creates a network of researchers within the JULES community with experience of how this site can be
25   exploited for JULES development, with input from site investigators. A key component is the provision of shared runs and
  evaluation datasets, which can be gradually expanded and improved.

Far more evidence is required to substantiate some of the points/arguments made to explain why the model is failing. As currently presented, they are purely speculative. For example, the authors speculate that an empirical link between leaf water potential and Vcmax/Jmax could improve models simulations. It would be an advance to the literature to actually show such a model (given they are relatively trivial to implement, e.g. Zhou et al. 2013, AFM; Kim and Verma 1991a, AFM), or at the very least, link more explicitly to literature that has done this (e.g. Tuzet et al. 2003, PCE; Zhou et al. 2013, AFM). Further, it would be useful to discuss the mechanism behind a direct limitation of leaf water potential on Vcmax and/or Jmax.

**Suggested change to manuscript:**

We have added a new section to address these points: ***What potential model developments could improve the diurnal cycle of JULES GPP at this site?*** . This goes through much more systematically the possible ways that JULES could be changed to capture the diurnal cycle of GPP during the dry spell than we had in the manuscript previously, and more clearly distinguishes between what we can conclude from this analysis, what is implied by this analysis and what is speculation.

It includes a discussion of how Zhou et al. 2013 and De Kauwe et al. (2015) allow the g1 parameter in the Medlyn et al 2011 stomatal conductance model to vary with soil moisture. We then present two new model runs, which show the approximate effect that multiplying the unstressed g1 by 0.5 and 0.25 would have on these JULES runs, if the Medlyn stomatal conductance model was used in JULES. This demonstrates that allowing $c_i$ to depend on soil moisture would be one way to improve the diurnal cycle of JULES GPP at this site. We then go on to discuss other possibilities, including adding an explicit leaf water potential dependence (including Kim and Verma (1991a), Kim and Verma (1991b), Tuzet et al. (2003), Sperry et al. (2017); Eller et al. (2018), Tardieu and Davies (1992); Dewar (2002); Huntingford et al. (2015)) and leaf rolling. We also mention the technical issues around modelling leaf water potential explicitly in JULES.

We attempted to implement the Kim and Verma 1991b model for A. gerardii and P. virgatum by applying their factor to net leaf assimilation, in place of the JULES beta function, using leaf water potential observations from Kim and Verma 1991a. The result is shown in Figure R1 below. We are only able to do this for the four days with leaf water potential observations. We have not included this in the manuscript because we feel this does not add an extra information beyond what was presented in Kim and Verma 1991b, which we have already summarised in this section.

[Figure]

**Figure R1.** Net canopy assimilation for days where leaf water potential observations were given in Kim and Verma 1991a.

**Suggested change to manuscript:**

New section ***What potential model developments could improve the diurnal cycle of JULES GPP at this site?*** which contains some material from the previous version of the manuscript (but better explained) and also

5      – plots of diurnal cycle of air temperature, leaf temperature and specific humidity deficit, to refer to when arguing that the temperature response in the repro-cox-1998 runs is successful because it acts as a proxy for the specific humidity response.

     – brief discussion of why including an interaction between soil moisture and specific humidity deficit could help the model reproduce the diurnal GPP cycle

10      – plots using 2 new model runs, which mimic the behaviour of decreasing g1 (g1 decreases in Zhou et al. 2013 when soil moisture is limiting)

     – brief discussion of why leaf water potential is another way to combine the effects of soil moisture and specific humidity deficit

     – brief mention of some models which include a leaf water potential dependence

> It is unclear why the authors feel like the influence of VPD is negligible in the existing approach in JULES (lines 6-8, p.11), given that increasing VPD would drive a reduction in Ci? Indeed, in the last paragraph of Section 3.2, they show that for tune-leaf VPD influences GPP via this mechanism on both the 30th July and the 11th August. They also show that this mechanism alone doesn't have the flexibility to reproduce the observations. For all the days presented in Figure 13, it would be interesting to also plot Ac / E (the transpiration can simply be derived from the latent heat) relative to the declining soil moisture (or/and time), to see where the relative constraint is greater (on A more than E or vice-versa?) which might help further understand why the model is failing for the more extreme dqcrit

15      The dependence of GPP on ci in these runs is pretty low anyway, even after tuning to observations, as shown in Figure 5 (much lower than, for example, C3 grass from the global C3 grass configuration in Harper et al, due to the different implementation of C3 and C4 photosynthesis in JULES).

     This extreme dqcrit shown in the previous version of the manuscript is not realistic, and is only shown in Figure 13 (old numbering) to demonstrate that even in this unrealistic regime of parameter space, there is almost no effect on GPP in three of 20 the four days shown, and on the other day (30th July), there is a threshold behaviour, where GPP collapses to zero for a time, and is almost not affected apart from this. This can be understood from examining the ci equation in JULES (i.e. the suggested Ac/ E analysis is not the best way to illustrate this).

     However, we appreciate that we have not done a good job of communicating the main result (i.e. that JULES can't capture the temporary drop in GPP in the middle of the day, regardless of what values of f0, dqcrit are used in the calculation of ci if 25 they are also consistent with leaf-level observations in unstressed conditions).

Therefore, we have re-thought the way that we perform this part of the analysis. We (a) use another, better, dataset of measurements of unstressed C4 grass in the Konza prairie, from Lin et al 2015 which enables us to better constrain the parameters in the default tune-leaf configuration and (b) frame the analysis as a sensitivity test, where we look at the behaviour of GPP for three different combinations of f0 and dqcrit, which vary more systematically, rather than just showing what happens for an extreme dqcrit and (c) include two more combinations of f0, dqcrit, which mimic the behaviour of g1 decreasing under water stress (as described above).

**Suggested change to manuscript:**

- Used a new dataset to constrain the parameters that go into the JULES ci/ca calculation - figure 6 (new numbering)

- Systematically varied dqcrit, while staying consistent with this dataset, to show that a different f0, dqcrit combination would not improve the fit to the canopy fluxes - figure 14 (new numbering)

- Varied f0, dqcrit to mimic the behaviour in Zhou et al 2013, to show that if these parameters varied over the course of the run (because of, say, soil moisture) then this would provide a much better fit to the canopy fluxes - figure 15 (new numbering)

It is clear that FIFE site 4439 has historical value for the JULES community and that a lot of data is available. I am uncertain, though, as to how representative of the C4 grass PFT or the tallgrass prairie vegetation in general it is? Could the authors elaborate on this point, perhaps in the discussion?

This is a very good point. Part of this would be addressed by including this site as one of a large dataset, as that protects against overtuning to one particular site (as discussed above, we have stressed this point more strongly in the newer version of the manuscript). So it should only be used as one of many examples of a C4 grass PFT. FIFE site 4439 was representative of tallgrass prairie vegetation in terms of its species distribution and having been recently burned (see section 2.3) but not in terms of the fact that it was fenced off and hence ungrazed. Also, tallgrass prairie is has evolved in regions where it has had to cope with strong seasonal water limitation - this makes it a very interesting of example to study, particularly as we are interested in the interaction of soil moisture stress and atmospheric VPD controls.

**Suggested change to manuscript:**

We have added a discussion of why adding a tallgrass prairie site to multi-site evaluation dataset is useful:

While, at one time, tallgrass prairie extended over 10% of the contiguous United States Fierer et al. (2013), it has declined 82-99% since the 1830s due to agricultural use (Sampson and Knopf, 1994; Blair et al., 2014a). However, grasslands in general (including other grass- and graminoid-dominated habitats, such as savanna, open and closed shrubland, tundra) cover

more terrestrial area than any other single biome type (up to 40% of Earth's land surface (Blair et al., 2014a)). It is therefore important to include lots of examples of grasslands in any global analyses of vegetation responses to changing conditions. The Konza Prairie LTER site, where FIFE was based, has been used extensively to investigate the dynamics and trajectories of change in temperate grassland ecosystems, including drivers such as fire, grazing, climate, nutrient enrichment (see Blair et al.

5    (2014b) for a review).

> Why is this site a good proxy to calibrate the model for this PFT, in particular considering the variability of the site data?

In global JULES runs, vegetation at the FIFE site would be modelled as 'natural C4 grass'. We would not recommend that the JULES 'natural C4 grass' PFT parameters get calibrated to this site - they are chosen to get good performance globally for this PFT, particularly in terms of global distributions of this PFT when the dynamic vegetation model within JULES is switched on. Since we do not consider the dynamic vegetation model at all, a discussion of how to improve the parameters in

10   the C4 grass PFT would be difficult to add to this manuscript without adding a lot of extra explanation which would not be relevant in the rest of the paper.

**Suggested change to manuscript:**

As described above, we have clarified the aims in the paper so that it is clearer that we are not proposing any changes to the

15   global C4 grass parameters.

> And is there any indication that it behaves like any other C4 grass site would during a dry spell? If this cannot be shown, I would encourage the authors to reword statements like '... FIFE observations were used to derive the original soil moisture stress parameterisation that was incorporated into JULES. This therefore makes FIFE an ideal test case for evaluating and improving this process.'.

**Suggested change to manuscript:**

We address the question of why FIFE is an ideal test case for evaluating improving water stress in JULES in the new section

***Can the FIFE dataset make a useful contribution to current-day JULES evaluation and development work?*** but we stress

20   that this is only alongside a comprehensive range of other datasets. For example,

> A global land-surface model such as JULES needs to perform well for a wide range of climate regimes, time scales, spatial scales and vegetation types. Model evaluation or development work needs to represent this variety. The availability of comprehensive databases, such as FLUXNET (Baldocchi et al., 2001) and the TRYKattge et al. (2011), have revolutionised land-surface science by giving easy access to observations from a wide variety of sources, in a common format. ... When adding
>
25   > a new process to a global land-surface model, it is important to tune new parameters to a comprehensive range of datasets. For

example, as mentioned in Section 3.3, Lin et al. (2015) use data for 314 species from 56 sites across the world to tune the new $g_1$ parameter introduced in the Medlyn model of stomatal conductance for key plant functional types. This breadth of sites and vegetation types is essential.

and in the new version of the conclusion,

5  FIFE can play a role in JULES evaluation and development only as one small component of a comprehensive range of datasets, covering different climate regimes, time scales, spatial scales and vegetation types... Confidence that the model is capturing key processes is necessary if the model is being run into new regimes, such as when forced with climate projections. This ability to disentangle and evaluate individual processes emphasises the value that intensive experiments such as FIFE have towards the larger modelling community evaluation efforts.

The authors could also more clearly demonstrate the impact of key assumptions. The following is an important point concerning the physical representation of the response of Vcmax to leaf temperatures above 32°C in Cox et al. (1998): 'However, as discussed in Section 2, this temperature response is not supported by observations in Knapp (1985) or Polley et al. (1992). Therefore, it appears that, while the model is successfully capturing the shape of the diurnal cycle during the dry period, it is not achieving this with the correct physical process.' It would be easy for the authors to test this, simply by swapping the temperature response function and determining if this statement is true or false.

10  **Suggested change to manuscript:**

As described above, we have added plots showing 3 runs that swap the Vcmax temperature response curve in the repro-cox-1998 configuration with other temperature distributions (these runs had formed part of the original work, which is why we were able to state that the repro-cox-1998 relied on its temperature dependence to reproduce the temporary drop in canopy 15  assimilation during the day in the dry period, but we had not backed up this claim by showing the plots in the previous version of the manuscript).

Finally, I would like to thank the authors for making all of the code and data available. Interactive Their careful description of the steps taken to set up the simulations is also appreciated.

Thanks, we very much appreciate that feedback, and we hope that the code and data will continue to be reused in other studies.

**2.2 Response to minor comments from Reviewer 1**

Lines 28-30, p.2: the precision that 'changing p0 can be considered a proxy for chang- ing the critical soil moisture' which appears lines 8-9, p.16, could probably appear here, or at least in Figure 1's caption

Varying this parameter does act as a useful proxy for varying the critical soil moisture theta_crit in our runs. However, it is not designed to do this (it is actually meant to represent differences between different vegetation types when grown on the same soil), so adding this in to this theoretical description of the JULES soil moisture parameterisation at this stage would not assist clarity. theta_crit is a hydraulic parameter of the soil (defined through soil potential). As we discuss, varying p0 is interesting in its own right, as a way to parameterise the response of the FIFE vegetation to soil moisture. We discuss the additional use of varying this parameter as a way of getting a handle on the effect of uncertainty in theta_crit in section 2, when we are discussing the uncertainty in the soil parameters and in section 3 when we discuss what we can conclude from varying p0. This works because we are just running the model at this one location, and have one vegetation type - the interpretation of anything else would start to get convoluted.

Varying the canopy clumping parameter $a$ is a similar case: this parameter has a physical interpretation, and could not be well constrained by the available observations. Therefore it is useful to look at the effect of the uncertainty in its own right. In addition, it can also give us insight into how GPP would differ if a different LAI was used, since more canopy clumping has a similar effect in the radiation scheme to less LAI (although the effect on leaf respiration differs).

Lines 1-2, p.5: even though the differences in evaporative schemes aren't the focus here, can the authors estimate how those might influence their conclusions?

We have added plots which split the model evapotranspiration into the transpiration and evaporation components. While this does not influence our conclusions, evaporation would be interesting to investigate at FIFE in a future study. In this case, we would use all time steps, rather than those selected in this analysis.

**Suggested change to manuscript:**

New plots added with model transpiration and evaporation.

Section 2.3: how is the tune-leaf configuration calibrated exactly? What is matched for in the calibration process? What does 'approximately representative of the dominant species' mean? How so?

The calibration of the tune-leaf configuration against site observations involves tuning the JULES parameters given in table A2. We outline how each of these parameters is calibrated in section 2.3. As described, some parameters can be calibrated directly to parameters given in literature, e.g. specific leaf area and the ratio of leaf nitrogen to leaf dry mass. However, many need to be calibrated in combination with eachother: plots 3,4,5, where we are calibrating to the equivalent quantity in JULES.

**Suggested change to manuscript:**

We have improved the calibration of the ci parameters f0 and dqcrit by using a better observational dataset. This has allowed us to greatly simplify the description in the manuscript, which we hope has assisted in the overall clarity. We have also clarified the description of figure 5 in the text and caption.

Lines 9-10, p.7: it is unclear to me why the burned plot was not water-limited. Could the authors please elaborate?

'not limiting' is a quote from their paper, from their experiment description (note that this just applies to data they took in the period May-June). We looked at whether this statement is consistent with their leaf potential observations in section 2.3.2.

**Suggested change to manuscript:**

We have edited the text to make it clear that 'not limiting' is a quote from Knapp et al:

Therefore, we use the observations from the burned plot in Knapp et al (1985) during May-June 1983, when they describe water availability as 'not limiting' (we will investigate this claim in more detail in Section 2.3.2)

Lines 31, p.7: why do the authors assume that the best parameter set is a composite of GMDD two species' parameter sets, given the non-linear response of photosynthesis to plant traits?

Our default tune-leaf configuration is based on A.g., so does not suffer from this issue. However, we do use a composite A.g./P.v. configuration to investigate the effect that considering other species might have on our conclusions. We absolutely agree that the interpretation of this composite configuration is clouded by the non-linearity of the response of photosynthesis to plant traits. However, we feel it is sufficient for our purposes, just to give an idea of the sensitivity.

Also, note that, as described above, we do not consider either of these two configurations to be the 'best'. Global-C4-grass continues to be the appropriate configuration to use for this site in global studies.

Line 10, p.8: I don't understand which mean the Al-Ci curves were normalised against Lines 11-13, p.8: can the authors demonstrate this claim or refer to studies that do?

The normalisation for the observations is the mean $A_l$ at high ci (i.e. >150) for each individual curve. We have clarified the identification and normalisation of these curves, both in the text and in the caption to figure 5.

The water stress factor $\beta$ cancels in the model Al when normalised in this way, because Al in the model is linearly dependent on $\beta$ (assuming that water stress is constant over the short period the curve was taken at - a very good approximation). Leaf nitrogen approximately cancels in the model Al when normalised in this way because the model Rubisco-limited and the model PEPCarboxylase limitation rate are both proportional to leaf nitrogen content. The light limited rate is not, which only pick points above a light threshold for this part of the analysis, so that the importance of the light-limiting rate is minimised

(although not zero) in these model points. Each curve corresponds to measurements taken on the same leaf, so the leaf nitrogen content is constant for that curve. The strength of this approach can be seen in the low spread of the model points, which makes it easier to calibrate to the observations. (most of the remaining spread is due to the change in leaf temperature while the measurements for the curve are being taken)

**Suggested change to manuscript:**

The text description and caption now read

... if we are to use these observations to calibrate the unstressed model parameters, we have to process them in such as way as to minimise the influence of the parameterisation of water stress in the model.

To achieve this, we identified individual net leaf assimilation ($A_l$) versus leaf internal $CO_2$ concentration ($c_i$) curves from the FIFE_PHO_LEAF_46 dataset for *A. gerardii* and *P. virgatum* (using the observation time and leaf area). We normalised each $A_l$-$c_i$ curve using the mean $A_l$ for $c_i > 150\mu$ mol $CO_2$ (mol air)$^{-1}$ for that curve. We then selected $A_l$-$c_i$ curves with mean incident radiation greater than 1200 $\mu$mol PAR m$^{-2}$ s$^{-1}$. This procedure minimises the dependence on water stress or individual leaf nitrogen levels, since these factors approximately cancel out in the relations used internally in JULES when they are manipulated in this way. We can then use these normalised curves to calibrate the model $A_l$-$c_i$ response at low $c_i$. For *A. gerardii* and, to a lesser extent, *P. virgatum*, this leads to a decrease in the initial slope of the $A_l$-$c_i$ curve (Figure 5).

and

Black crosses: $A_l$-$c_i$ curves for *Andropogon gerardii* (left) and *Panicum virgatum* (right) from FIFE_PHO_LEAF_46 (Polley et al 1992), normalised by the mean $A_l$ of the data points with $c_i > 150\mu$ mol $CO_2$ (mol air)$^{-1}$ in that curve. Only curves with mean incident PAR greater than 1200 $\mu$mol PAR m$^{-2}$ s$^{-1}$ have been used. Coloured points: normalised $A_l$ calculated from observed $c_i$ and incident PAR for each data point in the curve and the mean $T_{leaf}$ observation for each curve, using the JULES relations. The JULES parameters are taken from the `repro-cox-1998` configuration (red triangles), the `global-C4-grass` configuration (blue circles) and fits to A. g. data (`tune-leaf` default configuration, cyan diamonds) and P. v. data (yellow diamonds). Model points have been calculated using the Leaf Simulator package.

Lines 30-34, p.8: does this mean the dark respiration at Tleaf different to 30°C is then still scaled according to the temperature dependency in JULES (if so, can the authors justify this approach)? Or does the scaling follow Polley et al. (1992)?

In JULES, dark leaf respiration is proportional to $V_{cmax}$. We calibrated the temperature dependence of model $V_{cmax}$ to the Knapp net assimilation observations, taken at different temperatures. Therefore, the temperature dependence of dark leaf respiration in JULES is different to the temperature dependence in Polley 1992 (a rectangular hyperbola). This is illustrated in Figure 7 (old numbering) (note also that JULES has light-inhibited respiration - see section A6 for a more detailed description). This difference between the JULES leaf respiration at higher temperatures and the available observations does introduce an uncertainty, and is one of the reasons that we consider both GPP and net canopy assimilation in the results section. However,

this uncertainty is relatively minor (also note that (a) leaf temperatures in these runs only reach a maximum 42°C and (b) the Polley analysis has just 2 data points which are above 39°C ). This uncertainty does not affect our conclusions.

Lines 4-14, p.9: that whole paragraph could be moved to the beginning of section 2.3.1, thus the text that follows might be less confusing for the reader

We were able to move the second part of the paragraph to earlier in the section, but we couldn't get the rest of the text to fit in earlier, since it deals with the effect of not including S. nutans in the calibration lots of different parameters, according to the Polley 1992 study, all of which get discussed at different places in this section. Therefore moving this section would require lots of repetition which would reduce clarity.

**Suggested change to manuscript:**

We have moved the text

Polley et al. (1992) also found that there was 'no apparent relationship' between leaf temperature and net leaf carbon assimilation in their measurements of A. gerardii, S. nutans and P. virgatum, taken at ambient temperatures between 24.1°C and 47.8°C . They speculate that the difference between their results and the temperature relations found by Knapp (1985) is due to seasonal acclimatisation. On the one hand, this supports the change from using the rapidly varying Vcmax with temperature in this regime in both the repro-cox-1998 and global-C4-grass simulations to using the relatively more stable tune-leaf parameterisation. On the other hand, it implies that an even more stable parameterisation would be desirable. We will revisit this issue in Section 3.

to earlier in the section (and also redrafted it to get it to fit in).

Line 22, p.9: missing the work 'water' after 'vegetation. Leaf'

Thanks for pointing out this mistake. We have corrected it.

**Suggested change to manuscript:**

Changed 'leaf potential' to 'leaf water potential'

Lines 22-23, p.9: maybe specify what other factors can affect leaf water potential

In the new section ***What potential model developments could improve the diurnal cycle of JULES GPP at this site?*** we have added a more detailed description of leaf water potential, what affects it and how it can be modelled.

**Suggested change to manuscript:**

Added

leaf water potential is affected by both soil moisture (water supply) and VPD (atmospheric water demand)... Leaf water potential is not currently modelled explicitly within JULES. Typically, in plant hydraulic models, leaf water potential is calculated assuming a steady-state water balance, using the soil water potential, transpiration, and leaf-to-root and root-to-soil resistance terms (as in, e.g. Newman et al 1969). Adding this to the JULES code is technically non-trivial as water stress is currently applied to leaf-level processes before transpiration is calculated. Also, modelling the plant resistances would require additional input parameters, which would need to be constrained from observations.

Lines 1-2, p.15: is this observed during the dry period as well? Does this mean that it is constantly proportional through time?

Yes, approximately, according to the results from these two studies. The dataset we consider in this version of the manuscript also shows this approximate relationship (since the VPD dependence is small compared to the spread in ci)

**Suggested change to manuscript:**

Clarify this in the manuscript by comparing this result to the Lin et al dataset:

Both Knapp et al 1985 and Polley et al 1992 found that leaf stomatal conductance $g_s$ is proportional to the net leaf assimilation at this site. Their results are approximately consistent with the Lin et al 2015 observations, given the difference in ambient $CO_2$ levels and the weak dependence on VPD.

Lines 6-8, p.16: please add the missing words in the sentence

Thanks for noticing this.

**Suggested change to manuscript:**

We have changed the sentence

It demonstrates the importance of ensuring that the threshold for water stress is consistent with the 'unstressed' leaf observations we tuned against, since using p0 =0 with these new parameters would have resulted GPP that is much too low the early growth period that we were using for tuning.

to

It demonstrates the importance of ensuring that the threshold for water stress is consistent with the 'unstressed' leaf observations we calibrated against. Continuing to use $p_0$=0 with the newly-tuned unstressed parameters would have resulted in much too low GPP during the early growth period.

Line 3, p.17: missing 'of' before 'the humidity response'

Thanks for pointing this out. This is one of the sections of the manuscript which has been rewritten, so this sentence no longer exists in that form.

Lines 4-6, p.18: leaf rolling/folding implementation feasibility in a global model should at least be discussed in view of the existing literature and considering the author's previous statement that 'this behaviour cannot be modelled in the current version of JULES' (line 28, p.11); the same goes for including leaf nitrogen

We do not know of any land-surface schemes which currently implement this process - since they couple to global climate and Earth System Models, they need to strike a balance between representing key processes, while avoiding the introduction of
5    parameters that would be difficult to define across the globe. As a result, global land-surface schemes often simplify or neglect local processes in order to be more generally applicable in less well-measured locations. We believe leaf folding is in the group of processes that are currently neglected.

To our knowledge, no publications of JULES have attempted to change the PFT parameters governing the leaf nitrogen concentration over the course of the season, although one (Williams et al 2017) noted that this was a source of uncertainty
10    when modelling maize, particularly during the senescence period.

(n.b. JULES does include a nitrogen cycle, which we do not switch on in this study. This uses nitrogen availability in the soil to limit carbon uptake.)

**Suggested change to manuscript:**

15    We have changed the statement 'this behaviour cannot be modelled in the current version of JULES' to be more explicit: 'this behaviour is not implemented in the current version of JULES'.

and added this paragraphs to the ***What potential model developments could improve the diurnal cycle of JULES GPP at this site?*** section:

Finally, another way to improve the diurnal cycle of GPP in the dry period would be to incorporate a parameterisation of leaf
20    rolling. For example, effective leaf area available to the radiation scheme could be decreased during hot, dry weather. Kim and Verma 1991 attribute the residual overestimation of net canopy carbon assimilation on days during the dry period of their leaf water potential-based model to this effect. It would therefore be interesting to investigate the contribution that leaf rolling makes to the overall plant water use strategy. However, while the occurrence of leaf rolling/folding at the FIFE site has been recorded, the effect has not been quantified. This would be a necessary first step for modelling this process at this site.

Line 9, p.25: the authors should specify "for C4 grasses" or something equivalent

As part of the changes described above, the conclusions section has been rewritten, and this sentence no longer exists. However, we have added in comments elsewhere to stress that FIFE is a useful for looking at water stress in JULES only as one of a very large number of other sites covering lots of vegetation types e.g.

> FIFE can play a role in JULES evaluation and development only as one small component of a comprehensive range of datasets, covering different climate regimes, time scales, spatial scales and vegetation types.

Line 14, p.26: I believe this is the first time senescence is mentioned. Do the authors propose to do this via the leaf water potential parameterisation? Or do they envision it might somehow relate to a LAI phenology?

As described above, we have rewritten the conclusions section. The new version no longer contains a mention of senescence. It would be interesting in a future study to improve the way that the JULES phenology scheme models LAI at this site (at the moment, it gives a poor annual cycle, which we avoid in this study by forcing with LAI observations), and this could include drought-induced senescence.

Line 20, p.32: the approximation that soil evaporation can be neglected for days without rainfall seems rather big to me; have the authors considered including soil evaporation (though it isn't the focus of the study) to reduce the uncertainty?

All our model runs include evaporation from the soil moisture store, surface store and canopy. The latent heat observations also include both evaporation and transpiration. However, we wanted to minimise the influence of evaporation on our results and focus the discussion on transpiration, which is why we used the same procedure as Cox et al 1998 to pick timesteps.

**Suggested change to manuscript:**

Have added plots dividing the model evapotranspiration diagnostic into the transpiration and evaporation (from the soil moisture store, surface store and canopy) components.

Figure A4: the depth unit should appear somewhere in the plot

**Suggested change to manuscript:**

We have added the depth unit to this plot.

**3   Response to Reviewer 2**

This manuscript describes an evaluation of the JULES land model using field data from the FIFE dataset. Three simulations are presented: a replication of a simulation from an earlier model version, a standard simulation using the current version, and a tuned simulation using site specific parameters rather than global parameters. Numerous model intercomparison projects (MIPs) have been published in recent years, and one of the criticisms of these studies is the lack of adequate control in the experimental design. For example, models participating in a study may differ in forcing data, structure, and parameters, which makes attribution of differences in the results to these model characteristics difficult if not impossible. This study can be thought of as a three model MIP, and it features the same difficulty. Although the three models are all versions of JULES, they differ in multiple ways, making the interpretation of the results unclear. For example, simulations 2 and 3 differ due to the tuning of simulation 3 of multiple parameters for stomatal conductance and C4 photosynthesis, such as: SLA, leaf nitrogen, light response parameters, Vcmax temperature dependence, A-ci response at low ci, ci-ca relationship, dark leaf respiration - Vcmax ratio, p0 (water stress onset parameter), and canopy structure (uniformity). Figures 8-10 show that the site specific tuning exercise leads to large differences between simulations 2 and 3; in fact, these differences are larger under *unstressed* conditions than under stressed conditions. Moreover, while the tuned parameters should by design lead to better agreement with the observations against which they were calibrated (figures 3-7), they do not in general improve the simulation of GPP or An under unstressed conditions. I recommend to the authors to give the reader a better understanding of the individual impacts of this tuning exercise in the context of their effect on GPP and An. Which parameter changes improved the comparison, and which degraded the comparison? This is important to understand, more so because the agreement differs for the different days presented in the analysis.

As discussed in the response to reviewer 1 above, the work for this study involved hundreds of runs, to get a deep understanding of the model behaviour. One of the most difficult challenges we faced when writing this manuscript was how to present our key findings from this large number of runs in a coherent way. Therefore, we had to pick a subset of runs, choosing each carefully to illustrate a particular point.

5 These were based on three main configurations. The first, repro-cox-1998, was used to demonstrate that a previous study was able to fit this data very well (when tuned to it), but that part of this success relied on a process (temperature response of Vcmax) which is tuned in a way that makes it a poor fit to the physical process it is designed to represent. The second configuration represents a 'standard' configuration that would be used to represent this PFT in a global run. We show that this shows an unreasonable degree of water stress during a period where the vegetation should not be experiencing severe stress

10 and that the shape of the diurnal cycle during the dry season is not well-captured. This could be because the generic C4 grass tuning parameter values are not suitable for the vegetation at the FIFE site or because the underlying parameterisations within the model are not adequate. In order to investigate this, we compare individual processes against FIFE observations and use the resulting tuned parameters in the tune-leaf configuration.

We then vary the tune-leaf configuration to see the impact of different plant species, varying canopy structure (or LAI

15 uncertainty), soil moisture for onset of stress (which can be considered a combination of vegetation response p0 and soil properties theta_crit), to show that our conclusions are robust to what we have identified to be the key observational and

parameter uncertainties. We were therefore able to show that the conclusion that the current version of JULES cannot capture the diurnal GPP cycle during the dry spell is robust to these uncertainties.

In the previous version of the manuscript, we then underlined that the model has an inadequate VPD response by inflating the VPD sensitivity to unrealistic proportions and demonstrating that even with these extreme parameters, the temporary drop in the dry season GPP diurnal cycle can not be modelled. We have replaced this section with a sensitivity test to systematically varying the VPD response, while maintaining consistency with unstressed observations (using a new observational dataset). We think that this is a clearer way to demonstrate the same point.

As a result of reading the reviewers' comments we have realised that the manuscript lacks an explicit illustration of the finding that the repro-cox-1998 gets the shape of the dry-season diurnal season correct due to its inaccurate Vcmax temperature dependence - we assert this in the manuscript, but did not include our evidence. We have therefore added extra plots, based on the repro-cox-1998 configuration, showing the effect of varying the Vcmax temperature dependence.

**Suggested change to manuscript:**

- Extensively edited the manuscript to make the motivation for each parameter combination clearer

- New sensitivity tests to the parameters involved in calculating the internal CO2 concentration - figure 14 (new numbering).

- added plots showing 3 runs that vary the Vcmax temperature dependence and keep other parameters the same as the repro-cox-1998 configuration

- Also added a plot showing how the effect of allowing the parameters involved in calculating the internal CO2 concentration to vary with soil moisture (this functionality is not included in the current version of JULES) - figure 15 (new numbering).

While it is subjective, I did not agree with the authors' statement that the model was "...able to successfully reproduce the net canopy assimilation and latent heat energy flux reasonably well through the season". Given this perhaps unsatisfactory starting point for unstressed conditions, I recommend to the authors to focus first on obtaining more credible results under unstressed conditions before addressing the model's response to drought.

We agree that representing the early season correctly is important and have tried to take full advantage of the relevant observations available at FIFE to investigate the model performance in this season. This focus here is whether both the repro-cox-1998 and global-c4-grass configurations are over-representing stress during this period, and after a careful consideration of performance of individual processes against observations and untangling compensations within the model, we are able to conclude that the default value of p0=0 does result in the model simulating a much too high degree of soil moisture stress during this period.

**Suggested change to manuscript:**

We have clarified the motivation for the tune-leaf configuration in the text.

The title indicates that the goal of the manuscript is the evaluation of water stress in the current version of JULES, thus my expectation was that the experimental design would isolate the behaviour of this parameterisation; however, this is not really the case.

We have changed the title to emphasize that this site can be used as part of a full evaluation of water stress in JULES in the
5   future, and to make sure that the title does not give the impression this paper itself contains a full (multi-site) evaluation of this process in the model.

**Suggested change to manuscript:**

Have changed the title from ***Revisiting the First ISLSCP Field Experiment to evaluate water stress in JULESv5.0*** to  ***How***
10   ***can the First ISLSCP Field Experiment contribute to present-day efforts to evaluate water stress in JULESv5.0?*** .

Only figures 11-13 show single factor analyses, and of those, only figure 11 directly examines the water stress parameterisation used in the model.

It is not possible to isolate the behaviour of soil moisture stress in *in-situ* observations for a clean comparison with the model without investigating factors such as the A-ci curve, Vcmax temperature dependence, LAI, canopy structure, soil properties, response to air VPD etc. Therefore we have to consider all these factors, and not just the shape of the soil moisture stress function.

15   **Suggested change to manuscript:**

We have clarified the motivation for each of parameter combination and explicitly showed how each one feeds in to our conclusions.

Figure 11 shows the results of varying the p0 parameter that determines the initial soil moisture value at which vegetation experiences stress. I recommend that the authors show a few actual lines rather than the spread of p0 = [0,0.4] to enable the reader to easily see whether increasing p0 increases or decreases GPP.

The p0 band is designed to demonstrate the uncertainty in GPP arising from the uncertainty in p0. The upper edge corre-
20   sponds to the maximum value of p0 and the lower edge to the minimum. This is because increasing p0 increases the stress factor beta (fig 1), and therefore increases GPP (since GPP is proportional to beta). We use a band rather than a set of lines to

show that this part of the analysis is looking at a key uncertainty, and keep it distinct from the sensitivity tests in figure 14 and 15 (new numbering), which plotted with lines.

**Suggested change to manuscript:**

We have added the information 'lower limit corresponds to p0=0, upper limit to p0=0.4' to the figure caption.

One might expect that water stress based on soil moisture would not exhibit large diurnal variation, and this is confirmed by these plots. Figures 8-10 show that the diurnal variation in GPP and An can be simulated (in this case by simulation 1) if a predictor having stronger diurnal variation (such as temperature) is used. However, the authors note that this type of parameterisation is not well supported ("The repro-cox-1998 simulation is more successful, but this response is mediated by a temperature dependence in leaf carbon assimilation which is not supported by observations") as shown by figure 4. At this point the analysis is basically complete, with no improvement in the diurnal cycle of carbon flux. This seems not to support the authors' conclusions regarding outcome of the study (e.g. "FIFE provides an ideal case study for improving the model representation of water stress on carbon and water fluxes on a tallgrass prairie site") as no significant improvement was made aside from tuning the p0 parameter based on unstressed soil moisture conditions.

5 We intend the manuscript to show that FIFE would be an ideal test site (along with a wide range of other sites) for future work on new parameterisations that could be added to the model. We did not intend to claim that we have added a new way of calculating soil moisture stress to JULES ourselves. We have realised that we failed to make this clear in the previous version of the manuscript and have made major changes to address this.

**Suggested change to manuscript:**

As described above, we have edited the manuscript to make this more explicit, including changing the title, major editing to the second half of introduction, rewriting the conclusions and adding a new section ***Can the FIFE dataset make a useful contribution to current-day JULES evaluation and development work?*** .

The authors note that leaf water potential was used by authors of previous studies to simulate the diurnal cycle of GPP under dry conditions, leading them to conclude "JULES is not currently able to capture the diurnal cycle of net canopy photosynthesis at this C4 grass site, due to the lack of a strong dependence on the canopy vapour pressure deficit (indirectly or directly)", but this is largely conjecture and not actually tested by the authors of this paper.

We have written the new section ***What potential model developments could improve the diurnal cycle of JULES GPP***
15 ***at this site?*** to discuss methods that could be investigated for improving the diurnal cycle of GPP at this site during dry spells (requested by reviewer 1). As part of this, we now demonstrate explicitly that changing the parameters in governing

the relationship between internal CO2 and VPD away from the unstressed values does help capture the dry spell diurnal GPP cycle.

**Suggested change to manuscript:**

Explicitly discussed which changes might help JULES capture the diurnal cycle of GPP during the dry season at this site and why.

Added runs which mimic a reduction in g1 during the dry season, to show that this could improve the fit to the observations.

Written more explicitly that we can not recommend changes to JULES based on one site alone, we can only discuss how these changes might impact this particular site e.g.

A global land-surface model such as JULES needs to perform well for a wide range of climate regimes, time scales, spatial scales and vegetation types. Model evaluation or development work needs to represent this variety... When adding a new process to a global land-surface model, it is important to tune new parameters to a comprehensive range of datasets. For example, as mentioned in Section 3.3, Lin et al. (2015) use data for 314 species from 56 sites across the world to tune the new g1 parameter introduced in the Medlyn model of stomatal conductance for key plant functional types. This breadth of sites and vegetation types is essential

In summary, this manuscripts describes many of the issues that one encounters when attempting to constrain a model to field observations, such as uncertainty in measure- ments and spatial heterogenaeity. It highlights the fact that greater model complexity does not guarantee greater model fidelity. It also shows that site-specific model parameters may give significantly different results relative to global parameters. This is valuable information and worth presenting. However, that is not the stated focus of the manuscript, which is water stress and its improvement in JULES. To that end, I recommend to the authors to revise their title and to shift the focus of their discussion towards the actual content of the paper.

**Suggested change to manuscript:**

As discussed, we have changed the title, redrafted the parts of the text dealing with the motivation and conclusions of this study and added a new section to explicitly discuss the value of including this site in large multi-site comparisons, to address these concerns.

**4   Response to Reviewer 3**

The authors present three simulations of FIFE Site 4439 (Konza Praire, Kansas) utilizing three configurations of JULES,

from Cox et al. 1998, Harper et al. 2016, and a third developed for this manuscript. The parameterizations are compared for sensitivity to changes in root-zone soil water, light availability, leaf temperature as well as for the relationship between intercellular CO2 and net assimilation. The results show that the two literature-based configurations may not capture observed relationships in site-level data (e.g. Polley et al. 1992, Knapp 1985). The simulations are next confronted with observation-based data of GPP, net assimilation, and latent heat flux from 8 days during 1987. Generally speaking, the Cox et al. simulation output tends to provide the best fit to these data. Lastly the authors present parameter sensitivity results (of GPP) for four "tune-leaf" parameters.

Strengths: (1) The authors very clearly explain the process by which they developed the tune-leaf parameterization and its differences with the Harper and Cox parameterizations. This includes Figures 3-5,7, which effectively demonstrate the fit provided to leaf-level data by the three parameterizations. (2) The authors show, across a small subset of observational data, that utilizing field-derived paramater values tend to degrade model performance (Figs8-10). This is an important (and not unexpected) finding that will merit further study.

We thank the reviewer for these comments, and agree that incorporating improved parameter values from field observations can often have surprising effects due to compensating errors (e.g. in Huntingford et al 2017, when they trialled a more comprehensive better leaf respiration dataset in JULES). This is a very interesting area of study, which is very important for trusting the underlying processes when we run the model in new regimes e.g. climate change studies.

Weaknesses: (1) Conclusions are not strongly based in the specific work conducted within this study. See comments.

5    As discussed at the beginning of this document, we realise that we have failed to clearly describe the motivation and conclusions of this study in the previous version of the manuscript.

**Suggested change to manuscript:**

We have edited the title and redrafted these discussions in the paper to be more specific.

(2) The scope of the study does not match the ambitions of the introduction. Page 2, Line 31 states: "This effort requires a large amount of data to evaluate against, covering a wide variety of climate and vegetation conditions". Yet within this study only one site is analyzed. The authors might consider better contextualizing the current status of progress towards such a dataset.

10   We intended this to sentence to make it clear that we did not think that what we present in this paper is a comprehensive evaluation, as it just considers one site. However, we feel that our work can make a worthwhile contribution to a comprehensive evaluation.

**Suggested change to manuscript:**

We have added a new section ***Can the FIFE dataset make a useful contribution to current-day JULES evaluation and development work?*** to be more explicit about this. This section stresses that lots of datasets need to be considered in land-surface model evaluation and development e.g.

> A global land-surface model such as JULES needs to perform well for a wide range of climate regimes, time scales, spatial scales and vegetation types. Model evaluation or development work needs to represent this variety... When adding a new process to a global land-surface model, it is important to tune new parameters to a comprehensive range of datasets. For example, as mentioned in Section 3.3, Lin et al. (2015) use data for 314 species from 56 sites across the world to tune the new $g_1$ parameter introduced in the Medlyn model of stomatal conductance for key plant functional types. This breadth of sites and vegetation types is essential.

and discusses the contribution that FIFE as a small component of larger evaluation efforts. It also discusses the new concept of 'JULES golden sites', which hopes to facilitate the use of detailed site data in the JULES community:

> although the data is easily downloadable, well documented and in common file formats, is still needs to be manipulated into a format that can be used in JULES runs. We aim to address this issue by providing a suite that can be used to pre-process the FIFE data and run JULES with the configurations described in this manuscript (see the 'code and data availability' section). This aim is central to the provision of this manuscript. FIFE is the first 'JULES golden site', a concept was launched at the annual JULES meeting 2018. A JULES golden site is a site targeted by the JULES community because it can help address one of the key science questions facing JULES and has high-quality observational data that can be used to drive JULES and evaluate the output. It creates a network of researchers within the JULES community with experience of how this site can be exploited for JULES development, with input from site investigators. A key component is the provision of shared runs and evaluation datasets, which can be gradually expanded and improved.

(3) The authors do not acknowledge the mismatch between the leaf level scale and the eddy covariance / ESM gridcells. A short discussion of the challenges of reconciling leaf level parameters with the effective large scale parameters would be merited.

As we mention in the manuscript, the Harper et al 2016 PFT parameters (which we base the global-c4-grass configuration on) have been evaluated both at the site-scale and for global gridded (n96) runs (Harper et al 2016, Harper et al 2018). So these parameters should give relatively good performance at both scales for C4 grass. Since we are our driving our runs with meteorological observations from FIFE and flux tower observations from FIFE, i.e. no gridded datasets, we feel that going into detail about site-scale versus ESM scale is beyond the scope of this study. We do, however, include a brief discussion of the difference between the site-averaged FIFE met product and the met observations at site 4439 in section A1. We discuss the effect of tuning to leaf-level observations on flux-tower scale carbon and water fluxes in section 3.2, where we present the results from the tune-leaf similations.

**Suggested change to manuscript:**

The new section *Can the FIFE dataset make a useful contribution to current-day JULES evaluation and development work?* explicitly says that multiple spatial scales need to be considered in model evaluation:

5        A global land-surface model such as JULES needs to perform well for a wide range of climate regimes, time scales, spatial scales and vegetation types.

(4) Observational data record is extremely short (8 days). Are there other data available (remote sensing or flux tower) that could be used to evaluate the different model configurations? Why is only one year of the FIFE data used?

The actual JULES runs were continuous from 1987-05-29 to 1987-10-12 (figure 2). While some datasets were available for longer than the duration of our model runs (e.g. the met forcing was available 1987-9), we were restricted to this length of run by the availability of the LAI observations. Flux measurements were also not taken continuously during the entire the period

10 of our run - just during the Intensive Field Campaigns (IFC), which targeted a critical phase of vegetative development (IFC-1 "greenup", IFC-2 "peak greenness", IFC-3 "dry-down", and IFC-4 "senescence"). We picked the eight days to be representative of each period, and also to coincide with the days analysed in Kim and Verma 1991a, which provides observations.

**Suggested change to manuscript:**

15 As part of the new discussion section *Can the FIFE dataset make a useful contribution to current-day JULES evaluation and development work?* , we have added:

       There are two main disadvantages to the use of FIFE in evaluation and model development studies. The first is the limited time period: observations are available for a period of up to three years, with some key measurements only undertaken during the intensive field campaigns. Where long term effects are being studied, alternative datasets would need to be used.

(5) Many statements based on literature review are presented without citation in the results and conclusions sections. The relevant citations should be repeated, or the reader should be referred to the specific section where the citations can be found.

20 **Suggested change to manuscript:**

Where the new version of the conclusions section refers specifically to a particular study in the literature, we have added in the citation.

**Suggested change to manuscript:**

We have moved this figure as suggested and updated the text accordingly.

Figures 8-10: would be interesing to see summary statistics on the goodness of fit for the various parameterizations.

We feel that most of the simpler measures of goodness of fit could be misleading for these particular plots given the shape of the diurnal cycle of GPP, net canopy assimilation and latent heat flux - it would over-emphasize a slight timing offset at the beginning or end of the day, and also influenced by differences between the times of day when observations are available (e.g. observations start at midday on 17th August, and only one observation is available in the crucial 9:30-13:30 period on 11th August).

We have, however calculated $R^2$ for the fit of the JULES ci:ca ratio to observations, since the model is a straight line, and there are no obvious features of the data which would detract from this interpretation. We use this $R^2$ to back up our assertion that the two JULES parameters dqcrit and f0 are poorly confined by this dataset (this can also be seen by eye in figure 6, new numbering).

**Suggested change to manuscript:**

Included $R^2$ in figure 6 (new numbering) and in the discussion of this figure in the text.

P2.L31: There is a significant disconnect between the first five sentences of this paragraph and the single-site nature of this study.

**Suggested change to manuscript:**

We have edited the introduction to clarify that we mean this study to contribute towards a larger (multi-site) evaluation effort and added a new section ***Can the FIFE dataset make a useful contribution to current-day JULES evaluation and development work?*** to talk about this issue explicitly.

P17.L24: "Other studies have argued that the dry period diurnal cycle at this site can by captured via an explicit dependence on leaf water potential". I understand this is covered in Section 2.3.2, but I think you should either repeat the citations or refer back to Section 2.3.2.

**Suggested change to manuscript:**

We have rewritten the conclusion section and this sentence no longer appears. We have looked through the new conclusions for other places where we reference studies in the literature and added the references in (2 occurrences).

P26.L2-3: "JULES is not currently able to capture the diurnal cycle of net canopy photosynthesis at this C4 grass site", unclear which model configuration you are referring to.

5     **Suggested change to manuscript:**

This sentence no longer appears in the new version of the conclusions section. We have checked that the new version is explicit about when it is talking about the results from one configuration in particular, and when it is talking about JULES in general.

P26.L2-3: "JULES is not currently able to capture the diurnal cycle of net canopy photosynthesis at this C4 grass site, due to the lack of a strong dependence on the canopy vapour pressure deficit (indirectly or directly)." Second clause needs citation. Not shown within this study.

10     **Suggested change to manuscript:**

We have rewritten the conclusions section and this sentence no longer appears in this form. However, there is a sentence in the new conclusions section makes a similar point.

In this new sentence, we say 'could be addressed' to emphasise that this is a possible solution (rather than the more definite
15 'due to lack of', which implies it is the only solution):

> This deficiency could be addressed by allowing the effect of soil moisture availability and vapour pressure deficit on stomatal conductance to interact, for example, via leaf water potential.

and we go in to this argument in detail in the new section *What potential model developments could improve the diurnal cycle of JULES GPP at this site?* to support this claim.

P26.L4-5: "The temperature response of Vcmax can be tuned to compensate for this, but it is more desirable for the model to respond to high temperature stress and high water stress individually." This is not very well contextualized, perhaps instead you could more specifically say that repro-cox still manages a reasonable diurnal cycle, but likely due to compensating errors, and then refer back to the section where you discuss that in greater detail.

**Suggested change to manuscript:**

The new conclusions section makes this point with improved clarity and with more context. We have described why it is important for the model to respond to temperature and VPD separately in the new section ***What potential model developments could improve the diurnal cycle of JULES GPP at this site?*** . We have also added plots which show explicitly the effect of using different Vcmax with the repro-cox-1998 runs.

P26.L5-6: "These runs also showed how the default water stress parameterisation can result in large reductions in photosynthesis during periods that are not considered water-limiting at the site." I am not sure which figure you are referring to or which time period specifically.

This sentence referred to section ***Onset of water stress and relationship between water stress and leaf potential*** (from the description of how the tune-leaf configuration is calibrated) but this sentence is no longer in the new version of the conclusions. Instead, the conclusions section makes the general point that you should not be

calibrating the unstressed parameters without also checking the time period during which the model considers the vegetation to be unstressed

P26.L10-11: "These have been extensively studied at FIFE in independent investigations and yet still show a wide spread, leading to large modelling uncertainties" needs citation.

This is first described in the ***Experimental set-up*** section but the appendix discusses this in detail, and plots the available observations. The result of uncertainties in LAI, canopy structure, plant-dependent threshold for water stress and critical soil moisture on the canopy water and carbon fluxes is demonstrated in the ***tune-leaf simulations*** part of the results section.

P26.L12: "The FIFE data also indicates ..." How exactly does the FIFE data indicate that JULES should represent leaf rolling and senescence?

We no longer refer to leaf rolling in the new version of the conclusions section.

---

## Referee Report (RR1)

The authors' response and revisions have satisfactorily addressed my comments on the earlier version of the manuscript. This paper is an informative description of the observations available in the FIFE dataset, and the role these data can play in evaluating LSMs. I especially valued inclusion of the new section: "Can the FIFE dataset make a useful contribution to current-day JULES evaluation and development work?".  I support publication of this version and look forward to reading the follow-up studies that utilize this work.

---

## Author Response (AR2)

**Response to reviewers: gmd-2018-210**

We thank reviewer #2 for their comments, which we repeat in the shaded boxes below. We address each comment in turn.

> This manuscript revision uses field data from the FIFE dataset to evaluate the JULES land model. It does not describe new model development, but instead compares and evaluates three configurations of JULES. The stated aim of the paper is to use the FIFE data to show that none of the model versions can capture the response of the vegetation to water stress (at least not for the "correct" reasons). The authors propose to use these results as a benchmark for future model development. It is still not clear to me why the authors feel the need to publish this study at this stage. An evaluation of a model is clearly a necessary step in the model development process. But after identifying a poorly represented process, why not continue the development process and attempt to improve the process representation? As a modeller, I would have liked to see some resolution of some of the identified deficiencies, in particular, the use of the Jacobs ci equation. In my opinion, the paper would be more valuable to the land modelling community if some of the identified biases were resolved.

We are proposing the manuscript to be published as a GMD model evaluation paper, rather than a GMD model development paper. As we discuss in section 3.4, model developments for a global land surface model need to work well for a wide variety of vegetation types and climate regimes. Therefore, we would not recommend any model developments based on this site alone. However, a detailed understanding of this site is important for diagnosing what parts of JULES perform well against tallgrass praire observations and which parts need improving, and this can inform future model developments when used alongside similarly well understood datasets. Studies based on 100+ datasets have limited space to go into the details and caveats for the interpretation of each site separately, and therefore need to be able to refer to other literature for these specifics.

> Comment on calibration of Jacobs' ci equation The authors make the following statement in regards to their attempt to

calibrate the Jacobs' equation for ci: "We found that the slope of the ci -ca relationship changed as ca increased... Therefore, we were unable to calibrate the JULES ci-ca relationship to this data."

They note that the ci/ca relationship shown in figure S8 is nonlinear, while the Jacobs' equation (eqn 1) is linear. Rather than address this apparent model shortcoming, the authors instead attempt to calibrate eqn 1 with the data in figure 6. Why is the data in figure 6 any better than that shown in figure S8? If anything it looks less useful; nothing about the data shown in figure indicates that it could be fit with a linear function. Later the authors state: "This implies that the Jacobs parameterisation used in JULES, where the relationship between ci/ca and specific humidity deficit does not vary over the course of the run, does not have the flexibility needed to capture the behaviour of GPP at this site." Why should one expect the slope to vary with time? Could not the behavior also be explained with a nonlinear ci function?

Our aim is to demonstrate that the current version of JULES, with the current representation of ci (i.e. Jacobs) does not give a good fit to GPP at this site during the dry spell. We are not attempting to propose a new functional form of ci versus ca (at constant dq) based on this dataset - as discussed above, new parameterisations should be motivated by comprehensive analyses of multiple datasets. In addition, our model runs are performed at constant atmospheric CO2 levels. Therefore the

5  non-linearity of ci versus ca observations (at constant dq) shown in figure S8 is not a primary concern - we are interested in ci at ambient ca for this year at this site. Of course, the response of vegetation to CO2 levels is very important in general for land-surface models, but is better explored using specifically designed experiments (e.g. FACE experiments). We therefore keep to the existing JULES parameterisation. We agree that the data in figure 6 does not constrain parameters governing the JULES response of ci to dq (at constant ca) very well, which is why we later perform a sensitivity test to variations in dqcrit.

10  Allowing the slope of ci/ca versus dq to vary as soil moisture varies over the course of the run is effectively what happens in Zhou et al. (2013) (figure 6, right panel). This is a published model, which has been calibrated with data from a variety of sites, and therefore it is interesting to try JULES runs that mimic the changes that water stress introduces under this model.

Comment on stress function It would be more intuitive to recast the stress function shown in figure 1 in tems of theta_crit and theta_wilt only. Given that the stress function is linear, p0 can be expressed in terms of theta_wilt and theta_crit. Varying p0 can then be described in terms of varying theta_crit, which is physically interpretable and can be compared to other soil properties such as porosity and field capacity. The authors allude to this fact later in the paper, but I suggest making this observation up front.

We describe the p0, theta_crit and theta_wilt parameters as they are implemented in JULES, to avoid confusion. In JULES, theta_crit and theta_wilt are parameters that depend solely on soil type (and are typically calculated from sand/silt/clay fractions

15  where they are not available directly from site observations, as described in section A4). In contrast, the parameter p0 is set separately for each PFT and does not depend on soil type. It is designed to allow different vegetation types to respond to the soil in different ways. In our study, we use one PFT and one soil type only, which is why one can act as a proxy for the other. However, in general (gridded) runs, these parameters have very different purposes.

Comment on assumed soil moisture stress The authors state that Kim and Verna show decreasing leaf water potential, but Polley et al. show no decrease in assimilation. Might this indicate that the observed changes in leaf water potential are not due to soil moisture stress, but simply increasing vapor pressure deficit? Figure 8 shows that pre-dawn water potential values are similar for all days, indicating limited water stress. Given the previously noted inadequacy of eqn 1, is it possible that the authors may be effectively compensating for the nonlinearity of the ci(dq) response through their water stress function?

The reason why Polley et al do not show a decrease in net leaf assimilation in the dry spell in their 1987 dataset is not known, but does not appear to be due to VPD, since observations were taken at ambient conditions throughout the day (Figure R1), including times with high VPD (according to timestamps in file, the dataset documentation, and personal communication with one of the authors). We therefore felt that, as we could not adequately explain this behaviour, it was better to calibrate JULES to the other available datasets (Knapp (1985) and Lin et al. (2015)). However, we still wanted to include the behaviour of the Polley et al dataset in the discussion for completeness, so that subsequent studies have access to all the available information.

Comment on proposed implementation of leaf water potential In section 3.3 the authors note that earlier studies used leaf water potential to modify photosynthetic parameters. Is there reason to believe that these parameterization are more physically based than the temperature dependence used in Cox et al.? The common aspect of temperature and leaf water potential appears to be similar diurnal variations. Is leaf water potential actually the key process, or does it simply have the "correct" variability required to empirically model the diurnal cycle of GPP etc.?

The Knapp (1985) study isolates the temperature response by physically changing leaf temperature (by altering the air temperature in the cuvette via the temperature of the water circulating through the water jacket). This does not allow the leaves to acclimatise, but that would tend to make the temperature response over the range during the dry spell more stable, not less stable. Given that the temperature diurnal cycle is therefore not driving the diurnal cycle of GPP during the dry spell, it is natural to next consider a leaf water potential dependence as a possible way to implement this behaviour, especially as it has been used successfully in Kim and Verma (1991a) and Kim and Verma (1991b) to model this site and forms a key part of more advanced water stress models (e.g. Tuzet et al. (2003); Sperry et al. (2017); Eller et al. (2018); Tardieu and Davies (1992); Dewar (2002); Huntingford et al. (2015), as discussed in section 3.3). We therefore describe adding a leaf water potential dependence as a possible way that JULES could be extended to improve the fit to FIFE data (alongside other possibilities) but we do not claim that this is the 'best' way. To determine this, many other datasets would need to be considered.

p. 3 remove: "We proceed as follows."

We have removed this sentence.

[Figure]

**Figure R1.** Net leaf assimilation against incident PAR from FIFE_PHO_LEAF_46. Observations were plotted only for $CO_2$ concentration inside the leaf greater than 150 $\mu$mol mol$^{-1}$ (Polley et al (1992) found that net leaf assimilation in *A. gerardii*, *S. nutans* and *P. virgatum* was saturated at 114±7 $mu$mol mol$^{-1}$, 88±10 $\mu$mol mol$^{-1}$ and 70±6 $\mu$mol mol$^{-1}$ respectively in this dataset). The box-and-whiskers are plotted using the derived values of net leaf assimilation (corrected to a common absorbed PAR and internal $CO_2$) in Polley et al (1992) Table 1. Also plotted is the band predicted by the tuned model given the range of leaf temperatures from the observations in each plot, assuming $\beta = 1$ and $c_i$=200 $\mu$mol $CO_2$ (mol air)$^{-1}$

p. 9 dq looks like a differential, consider renaming Delta q or Dq ?

We are aware of this confusion and have tried other notations, but run across the problem that we need to make a clear distinction between specific humidity deficit and vapour pressure deficit in Pa (which is typically denoted $D$). $\Delta q$ has the issue

that it looks like a finite difference and $Dq$ looks like a total derivative. We settled on $dq$ and $dqcrit$, but with italic $d$ rather than the non-italic d used in differentials i.e. $\frac{dq}{dq_{crit}}$ rather than $\frac{\mathrm{d}q}{\mathrm{d}q_{crit}}$. For JULES users, this has the added advantage that they will be familiar with this notation already, as this is how the variables are named in the JULES fortran code.

**How can the First ISLSCP Field Experiment contribute to present-day efforts to evaluate water stress in JULESv5.0?**

Karina E Williams[1], Anna B Harper[2], Chris Huntingford[3], Lina M Mercado[3,4], Camilla T Mathison[1], Pete D Falloon[1], Peter M Cox[2], and Joon Kim[5,6]

[1]Met Office, FitzRoy Road, Exeter, Devon, EX1 3PB, UK
[2]College of Engineering, Mathematics and Physical Sciences, University of Exeter, Exeter, EX4 4QF, UK
[3]Center for Ecology and Hydrology, Wallingford, OX10 8BB, UK
[4]Geography Department, College of Life and Environmental Sciences, University of Exeter, Exeter, UK
[5]Dept. of Landscape Architecture & Rural Systems Engineering, Interdisciplinary Program in Agricultural & Forest Meteorology, Research Institute for Agriculture and Life Sciences, Seoul National University, Seoul 08826, Republic of Korea
[6]Institute of GreenBio Science & Technology, Seoul National University, Pyeongchang, 25354 Republic of Korea

**Abstract.** The First ISLSCP Field Experiment (FIFE), Kansas, US, 1987-1989, made important contributions to the understanding of energy and $CO_2$ exchanges between the land-surface and the atmosphere, which heavily influenced the development of numerical land-surface modelling. Thirty years on, we demonstrate how the wealth of data collected at FIFE and its subsequent in-depth analysis in the literature continues to be a valuable resource for the current generation of land-surface models. To illustrate, we use the FIFE dataset to evaluate the representation of water stress on tallgrass prairie vegetation in the Joint UK Land Environment Simulator (JULES) and highlight areas for future development. We show that, while JULES is able to simulate a decrease in net carbon assimilation and evapotranspiration during a dry spell, the shape of the diurnal cycle is not well captured. Evaluating the model parameters and results against this dataset provides a case study on the assumptions in calibrating 'unstressed' vegetation parameters and thresholds for water stress. In particular, the response to low water availability and high temperatures are calibrated separately. We also illustrate the effect of inherent uncertainties in key observables, such as leaf area index, soil moisture and soil properties. Given these valuable lessons, simulations for this site will be a key addition to a compilation of simulations covering a wide range of vegetation types and climate regimes, which will be used to improve the way that water stress is represented within JULES.

*Copyright statement.* The works published in this journal are distributed under the Creative Commons Attribution 4.0 License. This licence does not affect the Crown copyright work, which is re-usable under the Open Government Licence (OGL). The Creative Commons Attribution 4.0 License and the OGL are interoperable and do not conflict with, reduce or limit each other.

©Crown copyright 2018

[revised manuscript text omitted]